# "Noisier" Noise Contrastive Estimation is (Almost) Maximum Likelihood

**Peiyu Yu**[1*]     **Dinghuai Zhang**[2*]     **Hengzhi He**[1*]     **Xiaojian Ma**[4]     **Sirui Xie**[1]

**Ruiyao Miao**[1]   **Yifan Lu**[1]   **Yasi Zhang**[1]   **Deqian Kong**[1]   **Ruiqi Gao**[3]   **Jianwen Xie**[5]

**Guang Cheng**[1]     **Ying Nian Wu**[1]

[1]University of California, Los Angeles   [2]Mila - Quebec AI Institute   [3]Google DeepMind
[4]Beijing Institute for General Artificial Intelligence (BIGAI)   [5]Akool Research

## ABSTRACT

Noise Contrastive Estimation (NCE) has fueled major breakthroughs in representation learning and generative modeling. Yet a long-standing challenge remains: accurately estimating ratios between distributions that differ substantially, which significantly limits the applicability of NCE on modern high-dimensional and multimodal datasets. We revisit this problem from a less explored perspective: the magnitude of the noise distribution. Specifically, we show that with a virtually scaled (*i.e.*, artificially increased) noise magnitude, the gradient of the NCE objective can closely align with that of Maximum Likelihood, enabling a trajectory-wise approximation from NCE to MLE, and faster convergence both theoretically and empirically. Building on this insight, we introduce "Noisier" NCE, a simple drop-in modification to vanilla NCE that incurs little to no extra computational cost, while effectively handling density-ratio estimation in challenging regimes where traditional MLE and NCE struggle. Beyond improving classical density-ratio learning, "Noisier" NCE proves broadly applicable: it achieves strong results across image modeling, anomaly detection, and offline black-box optimization. On CIFAR-10 and ImageNet64×64 datasets, it yields 10-step and even 1-step samplers that match or surpass state-of-the-art methods, while cutting training iterations by up to half.

## 1 INTRODUCTION

Generative modeling of data distributions has made impressive progress in recent years, driven by the development of a rich family of deep generative models (Ho et al., 2020; Song et al., 2020; Rombach et al., 2022; Karras et al., 2022; Lipman et al., 2022; Liu et al., 2022). Among existing paradigms, Noise Contrastive Estimation (NCE) (Gutmann & Hyvärinen, 2010; 2012), also referred to as density–ratio estimation in some literature (Sugiyama et al., 2012), is a powerful and fundamental framework that unifies generative modeling with discriminative representation learning (Karras et al., 2019; Chen et al., 2020; Radford et al., 2021; Huang et al., 2025). NCE reduces density estimation to a classification task: learning the ratio $r(\mathbf{x}) = q_*(\mathbf{x})/q_0(\mathbf{x})$ between a target distribution $q_*$ and a noise distribution $q_0$ from samples of each, thereby enabling density estimation without explicitly modeling the target density itself (Gutmann & Hyvärinen, 2010; Sugiyama et al., 2012).

Despite its wide adoption, most existing NCE-based objectives suffer from a fundamental drawback: when the target and noise distributions differ substantially, the neural classifier can achieve near-perfect discrimination accuracy while still providing a poor estimate of the density ratio. This issue, often referred to as the *density-chasm* (Rhodes et al., 2020), arises when the gap between the two distributions is large, *e.g.*, when the KL divergence between $q_*$ and $q_0$ exceeds tens of nats. Such situations are common in modern high-dimensional or highly multimodal datasets. Although NCE

---

*Equal Contribution. Code and data available at https://github.com/yuPeiyu98/Noisier-NCE.

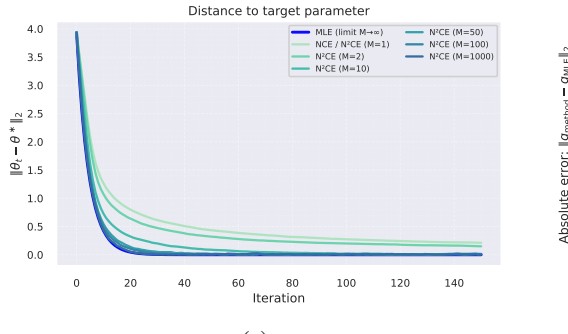 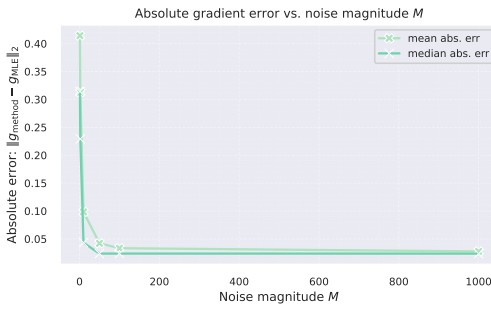

(a)                                                                                     (b)

Figure 1: **"Noisier" NCE gradients approach the MLE gradients.** As a sanity check, we simulate the results using 2d Gaussian distributions; *true* MLE gradients can be analytically computed. In Fig. 1a, we can see that $M \to \infty$ leads to a trajectory-wise convergence from NCE to MLE. In Fig. 1b, as the noise magnitude $M$ increases, "noisier" NCE gradients $\nabla_{\boldsymbol{\alpha}} \mathcal{L}_M^{\mathrm{NCE}}$ approach MLE gradients $\nabla_{\boldsymbol{\alpha}} \mathcal{J}^{\mathrm{MLE}}$; bias decaying in the order of $O(\frac{1}{M^2})$, which is consistent with Proposition 3.3. Further details can be found in Appx. D.1.

estimators are asymptotically consistent, meaning they recover the true density ratio in the infinite-sample limit, the convergence rate is provably slow: even an exponential increase in sample size yields only a linear decrease in estimation error (Poole et al., 2019; McAllester & Stratos, 2020), and the issue persists even with infinite data (Liu et al., 2021).

In this paper, we revisit NCE from a less explored perspective: the magnitude of the noise distribution used for ratio estimation. We show that, under mild conditions, introducing a virtually scaled noise magnitude allows the gradient of the NCE objective to align with that of Maximum Likelihood Estimation (MLE). This observation has two key implications. First, it establishes a principled *gradient-level* connection between NCE and MLE, where the NCE gradient serves as a controlled approximation to the MLE gradient. This perspective highlights that NCE can be understood as approximating the MLE in terms of the optimization trajectory rather than merely matching asymptotic error rates established by Gutmann & Hyvärinen (2012). Second, it offers a new lens on the density-chasm problem, revealing that a "noisier" NCE naturally mitigates convergence issues. Building on this insight, we propose "**N**oisier" **N**oise **C**ontrastive **E**stimation (N²CE), a simple drop-in modification to vanilla NCE that requires little to no additional computational cost. As we show in Sec. 4, this framework proves broadly applicable, demonstrating strong empirical performance in various downstream tasks (see Fig. 1 for an illustrative example).

Our main **contributions** are as follows: i) *Theoretically*, we show that, under mild conditions, a "noisier" NCE objective yields gradients that align with those of maximum likelihood, thus bridging the gap between NCE and MLE at the optimization trajectory level. Further, our analysis reveals that a virtually scaled noise magnitude naturally alleviates the convergence issues that arise when the target and noise distributions differ substantially. ii) *Practically*, we propose N²CE, a drop-in modification to vanilla NCE with negligible computational overhead. N²CE achieves strong empirical performance across diverse tasks including image modeling, anomaly detection, and offline black-box optimization. It produces 10-step and even 1-step samplers that match or surpass state-of-the-art methods on the CIFAR-10 and ImageNet64×64 datasets, with up to half the training cost.

## 2 BACKGROUND

**Noise Contrastive Estimation (NCE) and Nguyen-Wainwright-Jordan (NWJ)** Noise Contrastive Estimation (NCE) (Gutmann & Hyvärinen, 2010; 2012) offers a discriminative route to training unnormalized statistical models, sidestepping the need for direct likelihood maximization. Let $q_*$ denote the target distribution and $q_0$ a known noise distribution. Rather than directly modeling $p_{\boldsymbol{\alpha}}$ to approximate $q_*$, NCE estimates the ratio $r_{\boldsymbol{\alpha}}(\mathbf{x}) = \frac{p_{\boldsymbol{\alpha}}(\mathbf{x})}{q_0(\mathbf{x})}$ by training a classifier to distinguish between samples from $q_*$ and $q_0$. When the model is correctly specified, the optimal ratio satisfies $r_{\boldsymbol{\alpha}^*} \approx q_*/q_0$, which in turn implies $p_{\boldsymbol{\alpha}^*} \approx q_*$. In practice, $r_{\boldsymbol{\alpha}}$ is often parameterized as the exponential of a scalar-output neural network $\tilde{f}_{\boldsymbol{\alpha}}$. This parameterization naturally aligns with the energy-based view of $p_{\boldsymbol{\alpha}}$ (LeCun et al., 2006; Xie et al., 2016; Du & Mordatch, 2019), where $\tilde{f}_{\boldsymbol{\alpha}}$

incorporates the partition function. A common choice of classification loss is the logistic loss:

$$\mathcal{L}(\boldsymbol{\alpha}) = \mathbb{E}_{\mathbf{x} \sim q_*}\left[\log \frac{r_{\boldsymbol{\alpha}}(\mathbf{x})}{1 + r_{\boldsymbol{\alpha}}(\mathbf{x})}\right] + \mathbb{E}_{\mathbf{x} \sim q_0}\left[\log \frac{1}{1 + r_{\boldsymbol{\alpha}}(\mathbf{x})}\right]. \tag{1}$$

Numerous extensions of NCE have been developed, including generalized loss functions (Pihlaja et al., 2010; Sugiyama et al., 2012; Menon & Ong, 2016; Poole et al., 2019; Liu et al., 2021) and multi-stage ratio estimation strategies (Rhodes et al., 2020; Xiao & Han, 2022), which greatly enrich the paradigm. In this work, we revisit NCE from the perspective of the noise distribution's magnitude, showing that virtually scaling the noise magnitude offers a simple, effective remedy to long-standing convergence issues with little to no computational overhead.

Being equally fundamental and seminal, Nguyen-Wainwright-Jordan (NWJ), addresses the ratio–estimation problem through a variational characterization of $f$-divergences (Nguyen et al., 2010). This variational view reveals that likelihood–ratio estimation can be performed by maximizing a convex risk functional, yielding M-estimators that are statistically consistent and minimax-optimal under suitable regularity assumptions. Algorithmically, NWJ induces a tractable objective of the form

$$\mathcal{L}^{\mathrm{NWJ}}(\boldsymbol{\alpha}) = \mathbb{E}_{\mathbf{x} \sim q_*}[T_{\boldsymbol{\alpha}}(\mathbf{x})] + \mathbb{E}_{\mathbf{x} \sim q_0}[\exp T_{\boldsymbol{\alpha}}(\mathbf{x})], \tag{2}$$

where $T_{\boldsymbol{\alpha}} = \log r_{\boldsymbol{\alpha}}$. Provably, the optimum critic obtained by maximizing Eq. (2) retrieves the optimal ratio estimator. In contrast to NCE, which originates from a binary classification surrogate, NWJ is derived directly from convex duality and makes no explicit reference to a discriminative viewpoint. Intriguingly, however, as we demonstrate in Sec. 3.4, the noise-scaled NCE objectives studied in this work establish an implicit connection between these ostensibly distinct paradigms. This perspective positions our approach as a unifying family of likelihood-ratio estimators.

**Maximum Likelihood Estimation (MLE)**   Maximum Likelihood Estimation (MLE) remains the generative route: it fits probabilistic models by maximizing $\mathbb{E}_{q_*}[\log p_{\boldsymbol{\alpha}}(\mathbf{x})]$ over data samples. A common parameterization of $p_{\boldsymbol{\alpha}}$ (LeCun et al., 2006; Xie et al., 2016; Du & Mordatch, 2019) is

$$p_{\boldsymbol{\alpha}}(\mathbf{x}) := \frac{1}{Z_{\boldsymbol{\alpha}}} \exp\left(f_{\boldsymbol{\alpha}}(\mathbf{x})\right) q_0(\mathbf{x}), \tag{3}$$

where $f_{\boldsymbol{\alpha}}$ denotes the unnormalized log-density, $Z_{\boldsymbol{\alpha}}$ is the partition function, and $q_0(\mathbf{x})$ is the base noise distribution. The model in Eq. (3) can be interpreted as an energy-based correction or exponential tilting of $q_0$ (Xie et al., 2016). The corresponding gradient takes the general form

$$\nabla_{\boldsymbol{\alpha}} \mathcal{J}^{\mathrm{MLE}}(\boldsymbol{\alpha}) = \mathbb{E}_{q_*}[\nabla_{\boldsymbol{\alpha}} f_{\boldsymbol{\alpha}}(\mathbf{x})] - \mathbb{E}_{p_{\boldsymbol{\alpha}}}[\nabla_{\boldsymbol{\alpha}} f_{\boldsymbol{\alpha}}(\mathbf{x})]. \tag{4}$$

The first expectation over the data distribution is straightforward to approximate, but the second requires sampling from $p_{\boldsymbol{\alpha}}$. This is often intractable when the model involves an unknown partition function. In such cases, Markov Chain Monte Carlo (MCMC) methods such as Langevin dynamics (Welling & Teh, 2011) are commonly employed. While theoretically valid, MCMC sampling can converge slowly in high-dimensional or multimodal settings, making MLE difficult to apply in practice (Nijkamp et al., 2020). This sampling bottleneck is a key motivation for alternatives such as NCE, which avoids direct sampling from $p_{\boldsymbol{\alpha}}$ by reframing density estimation as a classification problem — an idea we revisit from a less-explored perspective in the next section.

## 3   "NOISIER" NOISE CONTRASTIVE ESTIMATION

### 3.1   NOISE MAGNITUDE

We now make precise what we mean by the *magnitude* of the noise distribution in N$^2$CE. Recall the standard objective in Eq. (1), where samples from the base distribution $q_0$ serve as negatives. More generally, we may scale the contribution of $q_0$ by a positive factor $M > 1$, which we refer to as the *noise magnitude*. This leads to the following "noisier" NCE objective (Gutmann & Hyvärinen, 2012):

$$\mathcal{L}_M(\boldsymbol{\alpha}) = \mathbb{E}_{q_*(\mathbf{x})}\left[\log \frac{r_{\boldsymbol{\alpha}}(\mathbf{x})}{M + r_{\boldsymbol{\alpha}}(\mathbf{x})}\right] + M\mathbb{E}_{q_0(\mathbf{x})}\left[\log \frac{M}{M + r_{\boldsymbol{\alpha}}(\mathbf{x})}\right]. \tag{5}$$

Here, $M = 1$ recovers the standard NCE objective, while larger $M$ corresponds to amplifying the effective weight of the noise distribution. Intuitively, this adjustment can be seen as replacing $q_0$ with a *virtually scaled* mixture of $M$ independent copies of $q_0$. As we show below, increasing $M$ has a striking effect: under mild conditions, the gradient of $\mathcal{L}_M(\boldsymbol{\alpha})$ approaches that of MLE, *i.e.*, Eq. (4). This not only establishes a principled bridge between NCE and MLE, but also provides a simple and effective remedy for the convergence issues of standard NCE.

## 3.2 SCALING EFFECT OF NOISE MAGNITUDE

> **Proposition 3.1** (Gradient approximation). *Under mild regularity conditions,*
> $$\lim_{M\to\infty} \nabla_{\boldsymbol{\alpha}} \mathcal{L}_M(\boldsymbol{\alpha}) = \mathbb{E}_{q_*}[\nabla_{\boldsymbol{\alpha}} f_{\boldsymbol{\alpha}}(\mathbf{x})] - \mathbb{E}_{p_{\boldsymbol{\alpha}}}[\nabla_{\boldsymbol{\alpha}} f_{\boldsymbol{\alpha}}(\mathbf{x})].$$

*Sketch of Proof.* The gradient of $\mathcal{L}_M$ can be expressed as

$$\nabla_{\boldsymbol{\alpha}} \mathcal{L}_M(\boldsymbol{\alpha}) = \int \frac{M}{M + r_{\boldsymbol{\alpha}}(\mathbf{x})} \left( q_*(\mathbf{x}) - p_{\boldsymbol{\alpha}}(\mathbf{x}) \right) \nabla_{\boldsymbol{\alpha}} f_{\boldsymbol{\alpha}}(\mathbf{x}) \, d\mathbf{x}.$$

This makes clear that, with smooth densities and ratio functions, as $M$ grows, the weight converges to 1 and the gradient approaches the MLE form. Additional derivation details appear in the Appx. B.1.

**Remark.** Our result (Proposition 3.1) establishes a gradient-level approximation to MLE. Prior works have not analyzed this: Gutmann & Hyvärinen (2012) focused only on asymptotic consistency without optimization dynamics, while Mnih & Teh (2012) treated discrete embeddings without convergence analysis. In contrast, our formulation applies to continuous settings and, crucially, enables the first explicit convergence guarantees for "noisier" NCE-type objectives. We demonstrate this in the exponential-family case summarized in Theorem 3.2.

> **Theorem 3.2** (Convergence in exponential families (informal)). *Under standard regularity assumptions for exponential families (details in Appx. B.3), letting $\lambda_{\min}, \lambda_{\max}$ denote the extremal eigenvalues of the Fisher information matrix, then for sufficiently large $M$, normalized gradient ascent on Eq. (5) finds an iterate within distance $\delta$ of the true parameter $\boldsymbol{\alpha}^*$ in at most*
> $$T \le C \left( \frac{\lambda_{\max}}{\lambda_{\min}} \right)^3 \frac{\|\boldsymbol{\alpha}^0 - \boldsymbol{\alpha}^*\|_2^2}{\delta^2}$$
> *iterations for a universal constant $C > 0$, i.e., there exists $t \le T$ with $\|\boldsymbol{\alpha}^t - \boldsymbol{\alpha}^*\|_2 \le \delta$.*

**Remark.** Theorem 3.2 gives a *polynomial* iteration complexity in the condition number $\kappa = \lambda_{\max}/\lambda_{\min}$ for exponential families when $M$ is sufficiently large. In contrast, for standard NCE the effective Hessian can be ill-conditioned unless $q_*$ and $q_0$ are already close. This can lead to much worse, often effectively exponential, dependence on the gap. Our result shows that virtually scaling the noise magnitude $M$ acts as a form of landscape regularization: the Hessian condition number of Eq. (5) remains uniformly bounded under standard assumptions (Liu et al., 2021), *without* requiring distributional closeness. We validate this intuition empirically: in 2D Gaussian simulations (Fig. 1) the trajectories indeed converge as predicted, and similar behavior persists even in high-dimensional, multimodal neural settings (Sec. 4). A full formal statement and proof are given in Appx. B.3.

## 3.3 PRACTICAL ERROR ANALYSIS, N²CE FAMILY AND REGULARIZATION

The population results above assume exact expectations and arbitrarily large $M$. In practice, both $M$ and the sample sizes are finite. We therefore ask: how well does the empirical "noisier" NCE gradient approximate the MLE gradient under these practical constraints? Proposition 3.3 provides an error decomposition that makes this trade-off explicit:

> **Proposition 3.3** (Finite-$M$, finite-sample error (informal)). *Let $\widehat{\mathcal{L}}_M$ be the empirical objective built from $n$ i.i.d. samples from $q_*$ and $q_0$, and write $D_M(\mathbf{x}) = \frac{r_{\boldsymbol{\alpha}}(\mathbf{x})}{M + r_{\boldsymbol{\alpha}}(\mathbf{x})}$. Under standard regularity assumptions (see Appx. B.2), the mean-squared approximation error satisfies*
> $$\mathbb{E} \left\| \nabla_{\boldsymbol{\alpha}} \mathcal{J}^{\mathrm{MLE}}(\boldsymbol{\alpha}) - \nabla_{\boldsymbol{\alpha}} \widehat{\mathcal{L}}_M(\boldsymbol{\alpha}) \right\|_2^2 \le V_u + B_u, \text{ where}$$
> $$B_u = O\left( \frac{1}{M^2} \right), V_u = \frac{C}{n} \left( \mathbb{E}_{q_*} \|\nabla_{\boldsymbol{\alpha}} \log r_{\boldsymbol{\alpha}}\|_2^2 + \min\left\{ M^2 \, \mathbb{E}_{q_0} \|\nabla_{\boldsymbol{\alpha}} \log r_{\boldsymbol{\alpha}}\|_2^2, \, \mathbb{E}_{q_0} \|\nabla_{\boldsymbol{\alpha}} r_{\boldsymbol{\alpha}}\|_2^2 \right\} \right)$$
> *and $C > 0$ hides benign constants.*

**Remark.** $B_u$ is the finite-$M$ *bias*; $V_u$ is the sampling *variance*. The finite-$M$ bias decays as $O(1/M^2)$, while the variance can grow as $O(M^2/n)$ unless the ratio (or its log) is sufficiently smooth under $q_0$, in which case the variance term saturates at the $\mathbb{E}_{q_0}\|\nabla r_{\boldsymbol{\alpha}}\|_2^2$ level. This highlights a bias–variance trade-off shaped jointly by $M$ and the behavior of $r_{\boldsymbol{\alpha}}$. Importantly, it shows that N²CE naturally induces a continuum of empirical objectives parameterized by $M$, and that an optimal finite-$M$ estimator exists in this spectrum. Besides, controlling the *roughness* of the ratio can further help stabilize the variance term and obtain robust performance. We make the optimal-$M$ characterization precise below, and present two concrete regularizations that restrict $r_{\boldsymbol{\alpha}}$ accordingly. As shown in Sec. 4, coupling these practical choices with the N²CE objective yields strong performance across a diverse set of tasks. A formal statement for Proposition 3.3 with more details appears in Appx. B.2.

**The N²CE family and U-shaped patterns** Indeed, with a finite fixed sample size $n$, we consistently observe the U-shaped dependence on $M$ predicted by Proposition 3.3. This emerges clearly in controlled 5-dimensional Gaussian experiments across different regimes (Appx. D.1), and perhaps more impressively, reappears in high-dimensional neural settings (Tabs. 16, 17 and 26). Furthermore, Proposition 3.3 predicts that the optimal $M$ should scale no larger than $C\sqrt{n}$, for some $C$ expected to lie within $1 - 10$ determined by the actual behavior of $r_{\boldsymbol{\alpha}}$. This theoretical prediction matches our empirical findings with remarkable fidelity. Thus, the finite-sample analysis in Proposition 3.3 both explains and anticipates the observed U-shaped curves, offering a principled and practical guideline for selecting $M$ in real-world applications. We next introduce two regularization strategies that provide practical handles for controlling the behavior of $r_{\boldsymbol{\alpha}}$.

**Multi-stage ratio estimation** One way to stabilize the gradient approximation is to adopt the multi-stage estimation strategy of Rhodes et al. (2020). This is especially useful when (i) a convenient base noise distribution is available, (ii) we prefer to use the density model directly as a discriminator or for decision making rather than as a smooth reward, and (iii) the problem scale is low- or moderately high-dimensional. The key idea is to decompose the ratio between $q_*$ and $q_0$ into a telescoping product, $\frac{q_*}{q_0} = \frac{q_*}{q_K}\frac{q_K}{q_{K-1}}\cdots\frac{q_1}{q_0}$, where $\{q_k\}_{k=1}^K$ are *pre-specified* intermediate distributions. Each ratio now involves a pair $(q_{k+1}, q_k)$ with greater overlap by design, yielding smaller values and more controllable variance at each stage. The trade-off is computational: in high-dimensional settings many stages may be needed, increasing overhead. For this reason, we mainly apply this technique in lower-dimensional tasks such as latent-space modeling.

**Direct ratio regularization** A more general and convenient strategy is to add a penalty on the ratio itself, for example $\mathbb{E}\|\log r_{\boldsymbol{\alpha}}\|_2^2$, directly to the "noisier" NCE objective. This approach is broadly applicable to high-dimensional data and does not rely on auxiliary intermediate distributions. The trade-off is that the added penalty may bias the gradients. Nonetheless, as we show in Sec. 4, this regularization is highly effective in practice, particularly for training reward models or critics in high-dimensional settings such as on ImageNet64×64 datasets.

### 3.4 AN INFORMATION-THEORETIC PERSPECTIVE

Complementing the gradient-based and finite-sample analyses, an information-theoretic interpretation further reveals that the N²CE objectives trace a continuous path between variational representations of JS and KL divergences (Sec. 7.13 in Polyanskiy & Wu (2025)), thereby making explicit both the $M \to \infty$ limit of N²CE being NWJ and its consistency with maximum-likelihood estimation.

**Variational representations of $f$-divergences** Specifically, on one end of the spectrum, the KL divergence satisfies the Nguyen–Wainwright–Jordan (NWJ) representation (see Sec. 2), recovering the NWJ objective in Eq. (2) when $T = \log r$:

$$D_{\text{KL}}(q_*\|q_0) = 1 + \sup_T\left(\mathbb{E}_{q_*}[T(\mathbf{x})] - \mathbb{E}_{q_0}[e^{T(\mathbf{x})}]\right) = 1 + \sup_r(\mathbb{E}_{q_*}[\log r(\mathbf{x})] - \mathbb{E}_{q_0}[r(\mathbf{x})]). \quad (6)$$

On the other end, the Jensen–Shannon divergence can be written as

$$D_{\text{JS}}(q_*\|q_0) = \log 2 + \sup_r\left[\mathbb{E}_{q_*}\left(\log\frac{r}{1+r}\right) + \mathbb{E}_{q_0}\left(\log\frac{1}{1+r}\right)\right], \quad (7)$$

which matches the standard NCE objective in Eq. (1) up to an additive constant (Nowozin et al., 2016). Thus NWJ and NCE arise as variational lower bounds on KL and JS divergences, respectively, each attaining its optimum at the true ratio $r^* = q_*/q_0$.

**N²CE objectives trace a continuous interpolation path** The N²CE objective in Eq. (5) interpolates between these two extremes. Let $\alpha = M/(1+M) \in (0,1)$ and consider the divergence

$$D_\alpha(q_*\|q_0) = (1-\alpha)D_{\mathrm{KL}}(q_*\|\alpha q_0 + (1-\alpha)q_*) + \alpha\, D_{\mathrm{KL}}(q_0\|\alpha q_0 + (1-\alpha)p_*)\,, \qquad (8)$$

which satisfies $D_{1/2} = D_{\mathrm{JS}}$ and $D_\alpha \to D_{\mathrm{KL}}$ as $\alpha \to 1$. One can now show that

$$D_\alpha(q_*\|q_0) = h(\alpha) + \sup_r \left[ \mathbb{E}_{q_*}\left(\log \frac{r}{M+r}\right) + M\,\mathbb{E}_{q_0}\left(\log \frac{M}{M+r}\right) \right]\,, \qquad (9)$$

which is identical (up to $h(\alpha)$, the binary entropy function) to the N²CE objective: (i) at $M=1$ ($\alpha = 1/2$), N²CE reduces to the usual NCE/JS bound; (ii) as $M \to \infty$ ($\alpha \to 1$), $\log \frac{Mr}{M+r} \to \log r$, $M \log \frac{M}{M+r} \to -r$, so the Eq. (5) converges to $\mathbb{E}_{q_*}[\log r] - \mathbb{E}_{q_0}[r] + \mathrm{const}$, the NWJ form of KL; maximizing this variational bound yields the same population optimum as maximum likelihood. We further provide empirical evidence in Appx. D.1 that N²CE indeed approaches NWJ when $M$ is sufficiently large ($M = 1e9$). With Sec. 3.2, these results jointly explain why letting $M \to \infty$ recovers maximum-likelihood learning, at both the divergence level and the gradient-dynamics level.

## 4 EXPERIMENTS

We organize our empirical study around three main questions: (i) How does the proposed objective compare with standard baselines such as pure MLE and vanilla NCE? (ii) Do the advantages of "noisier" NCE transfer to downstream tasks? (iii) How do key hyperparameters, in particular the noise magnitude $M$, affect performance?

To answer these, we conduct experiments across a broad range of datasets and benchmarks, spanning diverse model families. The results consistently show that our method outperforms existing approaches, while providing new insights into the role of noise scaling. Full experimental settings, implementation details and baseline descriptions are provided in Appx. C and D.

### 4.1 LEARNING LATENT-SPACE ENERGY-BASED MODEL (LEBM) WITH N²CE

Table 1: **FID(↓) on different datasets**. We highlight our model, the **1ˢᵗ** and 2ⁿᵈ performances; tables henceforth follow this format. Numbers from the first six rows are from Yu et al. (2024). nz denotes the latent dimension. M and K denote the noise magnitude and num. of stages for ratio estimation, respectively.

| Model | SVHN nz=100 | CelebA nz=100 | CIFAR10 nz=128 | CelebAHQ nz=512 |
|---|---|---|---|---|
| ABP | 49.71 | 51.50 | 90.30 | 160.21 |
| ABP-LEBM* | 29.44 | 37.87 | **70.15** | 133.07 |
| SRI | 44.86 | 61.03 | - | - |
| SRI (L=5) | 35.32 | 47.95 | - | - |
| 2s-VAE | 42.81 | 44.40 | 72.90 | - |
| RAE | 40.02 | 40.95 | 74.16 | - |
| VAE | 34.81 | 47.84 | 110.37 | 154.60 |
| w/ MLE-LEBM | 32.74 | 40.24 | 90.54 | 111.11 |
| w/ NCE-LEBM | 30.71 | 39.61 | 92.83 | 118.84 |
| w/ N²CE-LEBM | | | | |
| M=100,K=1 | 26.84 | 33.05 | 77.35 | 101.71 |
| M=100,K=3 | **25.63** | **31.09** | 77.05 | **95.66** |

Table 2: **AUPRC(↑) scores for unsupervised anomaly detection on MNIST**. Baseline numbers are taken from Yoon et al. (2023); Yu et al. (2024). Full results with variances in found in Appx. D.3.

| Heldout Digit | 1 | 4 | 5 | 7 | 9 |
|---|---|---|---|---|---|
| AE | 0.062 | 0.204 | 0.259 | 0.125 | 0.113 |
| VAE | 0.063 | 0.337 | 0.325 | 0.148 | 0.104 |
| ABP | 0.095 | 0.138 | 0.147 | 0.138 | 0.102 |
| IGEBM | 0.101 | 0.106 | 0.205 | 0.100 | 0.079 |
| MEG | 0.281 | 0.401 | 0.402 | 0.290 | 0.342 |
| BiGAN-$\sigma$ | 0.287 | 0.443 | 0.514 | 0.347 | 0.307 |
| ABP-LEBM | 0.336 | 0.630 | 0.619 | 0.463 | 0.413 |
| JVAEBM | 0.297 | 0.723 | 0.676 | 0.490 | 0.383 |
| Adaptive CE | 0.531 | 0.729 | 0.742 | 0.620 | 0.499 |
| NAE | 0.802 | 0.648 | 0.716 | 0.789 | 0.441 |
| MPDR-S | 0.764 | 0.823 | 0.741 | **0.857** | 0.478 |
| MPDR-R | 0.844 | 0.711 | 0.757 | 0.850 | 0.569 |
| DAMC | 0.684 | 0.911 | 0.939 | 0.801 | 0.705 |
| DAMC-NCE | 0.702 | 0.829 | 0.764 | 0.605 | 0.502 |
| DAMC-N²CE | | | | | |
| M=100,K=1 | 0.910 | 0.911 | 0.935 | 0.779 | 0.699 |
| M=100,K=3 | **0.959** | **0.935** | **0.959** | 0.845 | **0.854** |

As a proof of concept, we first conduct a set of lightweight experiments on latent energy-based models (LEBMs) across image datasets including CIFAR-10 (Krizhevsky et al., 2009), MNIST (LeCun, 1998), SVHN (Netzer et al., 2011), CelebA64 (Liu et al., 2015), and CelebAMask-HQ (Lee et al., 2020). Learning an LEBM (Pang et al., 2020a; Yu et al., 2022) can be viewed as fitting an unnormalized density model in the latent space of a VAE (Appx. C.1). Prior approaches largely rely on MCMC-based MLE-style training (Pang et al., 2020a; Du et al., 2021), where convergence and

Table 3: **CIFAR-10 (DDPM backbone) and ImageNet64×64 (EDM backbone) results** shown side-by-side. First six rows are from Yoon et al. (2024). $^\dagger$ highlights the starting point of DxMI fine-tuning.

| – Backbone: DDPM | | | | – Backbone: EDM | | | | |
|---|---|---|---|---|---|---|---|---|
| Method | NFE | FID ($\downarrow$) | Rec. ($\uparrow$) | Method | NFE | FID ($\downarrow$) | Prec. ($\uparrow$) | Rec. ($\uparrow$) |
| DDPM | 1000 | 3.21 | 0.57 | EDM (Heun) | 79 | 2.44 | 0.71 | 0.67 |
| FastDPM$^\dagger$ | 10 | 35.85 | 0.29 | EDM (Ancestral)$^\dagger$ | 10 | 50.27 | 0.37 | 0.35 |
| DDIM | 10 | 13.36 | – | Consistency Model | 2 | 4.70 | 0.69 | 0.64 |
| SFT-PG | 10 | 4.82 | 0.606 | Consistency Model | 1 | 6.20 | 0.68 | 0.63 |
| DxMI | 10 | 3.19 | 0.625 | DxMI | 10 | 2.68 | 0.777 | 0.574 |
| DxMI + Value Guidance | 10 | 3.17 | 0.623 | DxMI + Value Guidance | 10 | 2.67 | **0.780** | 0.574 |
| DxMI + NCE (M=1) | 10 | 3.93 | 0.623 | DxMI + NCE (M=1) | 10 | 2.69 | 0.756 | 0.585 |
| DxMI + N$^2$CE (M=100) | 10 | **2.99** | **0.638** | DxMI + N$^2$CE (M=100) | 10 | **2.23** | 0.757 | **0.599** |

stability are often problematic. This makes LEBMs a natural and tractable testbed for prototyping our method. In these experiments, we apply ratio decomposition as a regularization (Sec. 3.3) and follow the training and evaluation protocols in Pang et al. (2020a) (see Appx. C.2).

**Generative image modeling** We evaluate the quality of the learned LEBM with generated images measured by FID scores (Heusel et al., 2017) (see Tab. 1). For image generation with LEBM, we perform 100-step short-run Langevin Dynamics (LD) to draw latent vectors from the learned models, and map the latent vectors to the image space with the decoder network.

We observe that: first, LEBMs trained with N$^2$CE (*i.e.*, rows with M=100) consistently outperform those trained with vanilla NCE, with multi-stage estimation providing further gains that validate our analysis. Second, LEBMs trained with N$^2$CE show substantial improvements over those trained with MLE and short-run LD (denoted MLE-LEBM). These results confirm that N$^2$CE is a more reliable and effective objective that addresses the issues inherent to vanilla NCE and MCMC-based MLE. Finally, the improvement becomes more pronounced as the latent dimension nz increases, especially on the CelebAMask-HQ dataset with nz=512. This suggests that our proposal scales more gracefully to high-dimensional, multimodal targets.

**Anomaly Detection** To further highlight the practical advantages of our approach in highly multimodal settings, we evaluate anomaly detection on MNIST following the setup in Zenati et al. (2018). Specifically, we train models with one digit held out as anomalous, focusing on digits 1, 4, 5, 7, and 9—cases known to be especially challenging. We build on DAMC (Yu et al., 2024) for posterior inference, replacing its original prior with an LEBM learned via N$^2$CE. Since empirical posteriors from DAMC are sharp and highly multimodal, this serves as a demanding testbed. Table 2 reports AUPRC scores averaged over 10 trials. Our method yields consistent and often substantial improvements over baselines, achieving a strong overall performance across the hardest digits. Full experimental details appear in Appx. C.2 and D.3.

## 4.2 REWARD AND CRITIC LEARNING FOR DIFFUSION DISTILLATION

Building on the success of latent-space modeling, we next consider more challenging high-dimensional settings in the ambient image space. In particular, we study two representative scenarios for distilling diffusion samplers: (i) learning energy-based rewards for diffusion fine-tuning (DxMI) (Yoon et al., 2024), and (ii) learning critics for adversarial distillation (SiD$^2$A) (Zhou et al., 2024a). In both cases, we simply replace the original objectives with our "noisier" NCE combined with direct ratio regularization (Sec. 3.3), while using the same architectures and critical hyperparameters as baseline methods. Full implementation details appear in Appx. C.3.

In the DxMI setup (Yoon et al., 2024), N$^2$CE yields substantial improvements over vanilla NCE-like variants, previous methods and even the teacher models with full NFE (Tab. 3), underscoring its effectiveness in training rewards for diffusion distillation. In the SiD$^2$A setting (Zhou et al., 2024a), our method greatly outperforms vanilla NCE variants, and not only matches but can surpass strong baselines (Tabs. 4 and 5), notably with up to *half the training iterations*. Together, these results confirm that N$^2$CE scales to high-dimensional tasks while improving training efficiency across distinct distillation regimes. We provide additional ablation studies on $M$ on CIFAR-10 in Appx. D.4.

Table 4: **CIFAR-10 results**. "FID-U/C (iters)" shows uncond./cond. FID and, when provided, corresponding training iterations in parentheses. Baseline numbers from (Zhou et al., 2024a; Zheng & Yang, 2025).

| Method | NFE | FID-U ↓ | IS ↑ | FID-C ↓ |
|---|---|---|---|---|
| **Training From Scratch** | | | | |
| DDPM | 1000 | 3.17 | – | – |
| DDIM | 100 | 4.16 | – | – |
| Score SDE | 2000 | – | – | 2.20 |
| DPM-Solve-3 | 48 | – | – | 2.65 |
| EDM | 35 | 1.98 | – | 1.79 |
| BigGAN | 1 | – | – | 14.73 |
| StyleGAN2-ADA | 1 | 2.92 | 9.82 | 2.42 |
| SAN | 1 | **1.36** | – | – |
| iCT | 1 | 2.83 | 9.54 | – |
| iCT | 2 | 2.46 | 9.80 | – |
| iCT-deep | 1 | 2.51 | 9.76 | – |
| iCT-deep | 2 | 2.24 | 9.89 | – |
| **Post-training** | | | | |
| PD | 1 | 9.12 | – | – |
| DFNO | 1 | 3.78 | – | – |
| CD | 1 | 3.55 | – | – |
| CD | 2 | 2.93 | – | – |
| CTM | 1 | 1.98 | – | 1.73 |
| CTM | 2 | 1.87 | – | 1.63 |
| DMD | 1 | 2.62 | – | – |
| D2O-F | 1 | 1.54 | 10.10 | 1.44 |
| SiD | 1 | 1.92 | 9.98 | 1.71 |
| SiD$^2$A | 1 | 1.50 *(30K)* | 10.19 | 1.40 *(50K)* |
| SiD + NCE (M=1) | 1 | 1.53 *(30K)* | 10.20 | 1.46 *(30K)* |
| SiD + N$^2$CE (M=50) | 1 | **1.45** *(20K)* | **10.23** | **1.39** *(20K)* |

Table 5: **Conditional ImageNet 64×64 results.**

| Method | NFE | FID ↓ (iters) | Prec. ↑ | Rec. ↑ |
|---|---|---|---|---|
| **Training From Scratch** | | | | |
| RIN | 1000 | 1.23 | – | – |
| DDPM | 250 | 11.00 | 0.67 | 0.58 |
| ADM | 250 | 2.07 | 0.74 | 0.63 |
| EDM | 79 | 2.64 | – | – |
| iCT | 1 | 4.02 | 0.70 | 0.63 |
| iCT | 2 | 3.20 | 0.73 | 0.63 |
| iCT-deep | 1 | 3.25 | 0.72 | 0.63 |
| iCT-deep | 2 | 2.77 | 0.74 | 0.62 |
| BigGAN-deep | 1 | 4.06 | **0.79** | 0.48 |
| StyleGAN2-XL | 1 | 1.51 | – | – |
| **Post-training** | | | | |
| PD | 1 | 15.39 | – | – |
| BOOT | 1 | 16.3 | 0.68 | 0.36 |
| DFNO | 1 | 7.83 | – | 0.61 |
| CD | 1 | 6.20 | 0.68 | 0.63 |
| CD | 2 | 4.79 | 0.69 | **0.64** |
| CTM | 1 | 1.92 | 0.70 | 0.57 |
| CTM | 2 | 1.73 | 0.64 | 0.57 |
| DMD | 1 | 2.62 | – | – |
| sCD-S | 1 | 2.97 | – | – |
| sCD-S | 2 | 2.07 | – | – |
| DMD2 | 1 | 1.51 | – | – |
| MSD (DM) | 1 | 2.37 | – | – |
| MSD (ADM) | 1 | 1.20 | – | – |
| D2O-F | 1 | 1.16 | 0.75 | 0.60 |
| SiD | 1 | 1.52 | 0.74 | 0.63 |
| SiD$^2$A | 1 | **1.11** *(20K)* | 0.75 | 0.62 |
| SiD + NCE (M=1) | 1 | 1.28 *(15K)* | 0.75 | 0.62 |
| SiD + N$^2$CE (M=50) | 1 | **1.11** *(10K)* | 0.75 | 0.63 |

## 4.3 OFFLINE BLACK-BOX OPTIMIZATION

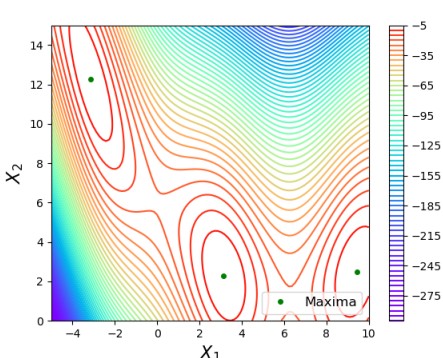

Figure 2: **Branin function level sets.**

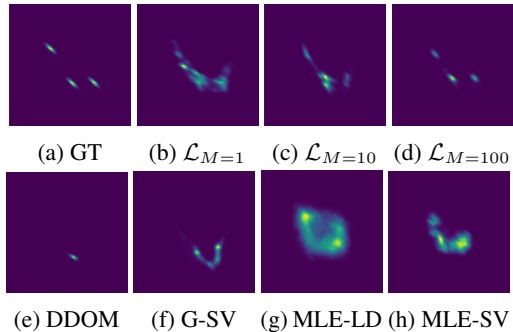

(a) GT    (b) $\mathcal{L}_{M=1}$    (c) $\mathcal{L}_{M=10}$    (d) $\mathcal{L}_{M=100}$

(e) DDOM    (f) G-SV    (g) MLE-LD   (h) MLE-SV

Figure 3: **Viz. of Branin optimal samples.** (b–d) are results of our method. G-SV denotes the Gaussian prior model sampled with SVGD. MLE-LD and MLE-SV denote the model trained by MLE sampled with LD and Stein Variational Gradient Descent (SVGD), respectively.

Beyond image modeling, we also explore the broader impact of our proposed technique through the lens of offline Black-Box Optimization (BBO). This task evaluates not only how well a model captures the training distribution, but also its ability to internalize structural regularities and generalize to unseen queries. In the *offline* BBO setting (Trabucco et al., 2022), one is given a pre-collected dataset $\mathcal{D} = \{(\mathbf{x}_i, y_i)\}_{i=1}^n$ of inputs and their black-box function values $h(\mathbf{x}_i) = y_i$. At test time, the optimizer may access a limited budget $Q$ of evaluations of the unknown function $h$ and must return candidates with high observed values. This setup is particularly well-suited for assessing generative approaches to optimization, where the function value $y$ acts as a conditioning signal (Brookes et al., 2019; Kumar & Levine, 2020; Krishnamoorthy et al., 2023).

We instantiate this view by parameterizing a conditional model $p_{\boldsymbol{\theta}}(\mathbf{x} \mid y)$ via latent variables $\mathbf{z}$, using

$$p_{\boldsymbol{\theta}}(\mathbf{x} \mid y) \; \propto \; \mathbb{E}_{p(\mathbf{z}|y)}[p_{\boldsymbol{\beta},\mathbf{x}}(\mathbf{x} \mid \mathbf{z})], \quad p(\mathbf{z} \mid y) \propto p_{\boldsymbol{\beta},y}(y \mid \mathbf{z}) \, p_{\boldsymbol{\alpha}}(\mathbf{z}),$$

and employing stochastic samplers such as Langevin Dynamics (LD) or Stein Variational Gradient Descent (SVGD) (Liu & Wang, 2016) for conditional sampling. To train this model, we use a VAE with an LEBM prior that jointly models $(\mathbf{x}, y)$. Full problem statement, implementation details, and extended results are deferred to Appx. D.5.

### 4.3.1 2D BRANIN FUNCTION

We begin with the 2D Branin function (Fig. 2) as a proof of concept. This task serves as a sanity check to validate that our method can both faithfully capture high-value modes and generalize beyond the best points observed in the offline dataset (see Appx. D.6 for details). Following DDOM (Krishnamoorthy et al., 2023), we perform a held-out offline optimization experiment. Specifically, we uniformly sample $N = 5000$ points from the domain $[-5, 10] \times [0, 15]$ to construct the offline dataset, then remove the top 10% by function value. This ensures that high-value contours are retained, but the exact global optima are excluded during training. For evaluation, we adopt a query budget $Q = 128$ and use SVGD for sampling. Table 6 shows that our method

Table 6: **Results on the top-10%-tile-removed Branin task (avg. over 5 runs).** OPT denotes the global optimum.

| $\mathcal{D}_{\max}$/Opt. | GA | BONET | DDOM | Ours |
|---|---|---|---|---|
| -6.1/-0.4 | $-4.0_{\pm 4.3}$ | $-1.8_{\pm 0.8}$ | $\underline{-1.6_{\pm 0.1}}$ | $\mathbf{-0.4_{\pm 0.1}}$ |

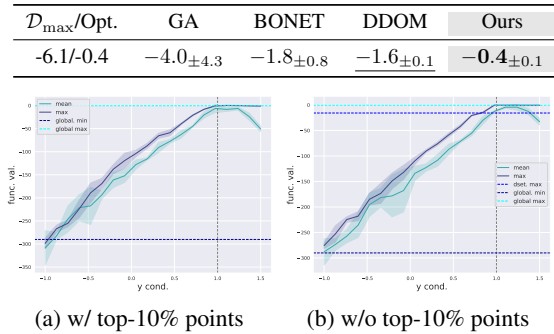

(a) w/ top-10% points     (b) w/o top-10% points

Figure 4: **Results on uniformly sampled Branin w/ and w/o top-10% points.** Zoom in for details.

generalizes well beyond the maximum value in the dataset $\mathcal{D}_{\max}$, approaching the true optimum and outperforming gradient-based GA as well as modern generative baselines BONET (Mashkaria et al., 2023) and DDOM (Krishnamoorthy et al., 2023).

We further visualize conditional samples corresponding to high function values (Fig. 3), and examine the correlation between target conditions (desired function values) and achieved function values (Fig. 4). For visualization, we draw 2560 samples from each model. In Fig. 3a, the distribution of true optima is emulated with a 3-component GMM centered at the global minima. As shown in the first row of Fig. 3, increasing $M$ in Eq. (5) produces samples that better approximate the ground-truth distribution, while models trained with MCMC-based MLE or vanilla NCE (`M=1`) lag behind. This again supports our analysis that a sufficiently large $M$ is critical for accurate gradient approximation.

Table 7: **Normalized results on design-bench with $Q = 256$ (avg. over 5 runs).**

| BASELINE | TFBIND8 | TFBIND10 | CHEMBL | SUPERCON. | ANT | D'KITTY | MEAN SCORE↑ | MEAN RANK↓ |
|---|---|---|---|---|---|---|---|---|
| $\mathcal{D}$ (best) | 0.439 | 0.467 | 0.605 | 0.399 | 0.565 | 0.884 | - | - |
| GP-qEI | $0.824 \pm 0.086$ | $0.635 \pm 0.011$ | $0.633 \pm 0.000$ | $0.501 \pm 0.021$ | $0.887 \pm 0.000$ | $0.896 \pm 0.000$ | $0.729 \pm 0.019$ | 7.8 |
| CMA-ES | $0.933 \pm 0.035$ | $0.679 \pm 0.034$ | $0.636 \pm 0.004$ | $0.491 \pm 0.004$ | $\mathbf{1.436 \pm 0.928}$ | $0.725 \pm 0.002$ | $\underline{0.816 \pm 0.168}$ | 5.8 |
| REINFORCE | $0.959 \pm 0.013$ | $0.640 \pm 0.028$ | $0.636 \pm 0.023$ | $0.481 \pm 0.017$ | $0.261 \pm 0.042$ | $0.474 \pm 0.202$ | $0.575 \pm 0.054$ | 8.3 |
| Gradient Ascent | $\underline{0.981 \pm 0.015}$ | $0.659 \pm 0.039$ | $0.647 \pm 0.020$ | $0.504 \pm 0.005$ | $0.340 \pm 0.034$ | $0.906 \pm 0.017$ | $0.672 \pm 0.021$ | 5.0 |
| COMs | $0.964 \pm 0.020$ | $0.654 \pm 0.020$ | $0.648 \pm 0.005$ | $0.423 \pm 0.033$ | $0.949 \pm 0.021$ | $0.948 \pm 0.006$ | $0.764 \pm 0.018$ | 5.8 |
| BONET | $0.975 \pm 0.004$ | $0.681 \pm 0.035$ | $\underline{0.654 \pm 0.019}$ | $0.437 \pm 0.022$ | $0.976 \pm 0.012$ | $\underline{0.954 \pm 0.012}$ | $0.780 \pm 0.022$ | $\underline{3.7}$ |
| CbAS | $0.958 \pm 0.018$ | $0.657 \pm 0.017$ | $0.640 \pm 0.005$ | $0.450 \pm 0.083$ | $0.876 \pm 0.015$ | $0.896 \pm 0.016$ | $0.746 \pm 0.003$ | 7.3 |
| MINs | $0.938 \pm 0.047$ | $0.659 \pm 0.044$ | $0.653 \pm 0.002$ | $0.484 \pm 0.017$ | $0.942 \pm 0.018$ | $0.944 \pm 0.009$ | $0.770 \pm 0.023$ | 5.5 |
| DDOM | $0.971 \pm 0.005$ | $\underline{0.688 \pm 0.092}$ | $0.633 \pm 0.007$ | $\underline{0.560 \pm 0.044}$ | $0.957 \pm 0.012$ | $0.926 \pm 0.009$ | $0.787 \pm 0.034$ | 4.5 |
| Ours | $\mathbf{0.990 \pm 0.003}$ | $\mathbf{0.803 \pm 0.085}$ | $\mathbf{0.661 \pm 0.025}$ | $\mathbf{0.567 \pm 0.017}$ | $\underline{0.982 \pm 0.012}$ | $\mathbf{0.961 \pm 0.006}$ | $\mathbf{0.827 \pm 0.021}$ | **1.2** |

### 4.3.2 OFFLINE BBO ON DESIGN-BENCH

**Task setup and evaluation.** We next evaluate on higher-dimensional real-world tasks[1] from Trabucco et al. (2022), grouped into **discrete optimization** (TF-Bind-8, TF-Bind-10, ChEMBL) and **continuous optimization** (D'Kitty, Ant Morphology, Superconductor). We compare against three categories of baselines: **(i)** generative inverse models with different parameterizations, including CbAS (Brookes et al., 2019), Auto.CbAS (Fannjiang & Listgarten, 2020), MIN (Kumar & Levine,

---

[1]NAS and Hopper tasks are excluded following Mashkaria et al. (2023); see Appx. D.7.2.

Table 8: **Ablation studies on design-bench (avg. over 5 runs)**, results with a budget $Q = 256$.

| BASELINE | TFBIND8 | TFBIND10 | CHEMBL | SUPERCON. | ANT | D'KITTY | MEAN SCORE↑ |
|---|---|---|---|---|---|---|---|
| $\mathcal{D}$ (best) | 0.439 | 0.467 | 0.399 | 0.565 | 0.884 | 0.605 | - |
| LD sampler | | | | | | | |
| MLE-LEBM | $0.524 \pm 0.228$ | $0.667 \pm 0.000$ | $0.633 \pm 0.000$ | $0.362 \pm 0.006$ | $0.512 \pm 0.000$ | $0.672 \pm 0.049$ | $0.562 \pm 0.033$ |
| N²CE-LEBM (M=100,K=6) | $0.834 \pm 0.073$ | $\underline{0.739 \pm 0.095}$ | $0.639 \pm 0.013$ | $0.363 \pm 0.006$ | $\underline{0.957 \pm 0.023}$ | $0.955 \pm 0.004$ | $0.748 \pm 0.025$ |
| SVGD sampler | | | | | | | |
| MLE-LEBM | $0.941 \pm 0.012$ | $0.681 \pm 0.079$ | $0.634 \pm 0.002$ | $0.296 \pm 0.004$ | $0.926 \pm 0.016$ | $0.915 \pm 0.007$ | $0.732 \pm 0.014$ |
| NCE-LEBM (M=1,K=6) | $\underline{0.961 \pm 0.024}$ | $0.655 \pm 0.034$ | $0.640 \pm 0.011$ | $\underline{0.497 \pm 0.044}$ | $0.907 \pm 0.007$ | $\underline{0.959 \pm 0.002}$ | $0.770 \pm 0.007$ |
| N²CE-LEBM (M=100,K=1) | $0.945 \pm 0.025$ | $0.722 \pm 0.043$ | $0.648 \pm 0.003$ | $0.421 \pm 0.020$ | $0.893 \pm 0.008$ | $0.953 \pm 0.008$ | $0.764 \pm 0.005$ |
| N²CE-LEBM (M=100,K=6) | $\mathbf{0.990 \pm 0.003}$ | $\mathbf{0.803 \pm 0.085}$ | $\mathbf{0.661 \pm 0.025}$ | $\mathbf{0.567 \pm 0.017}$ | $\mathbf{0.982 \pm 0.012}$ | $\mathbf{0.961 \pm 0.006}$ | $\mathbf{0.827 \pm 0.021}$ |

2020), and DDOM (Krishnamoorthy et al., 2023); **(ii)** gradient(-like) updating from existing designs, such as gradient-ascent-based methods (Fu & Levine, 2020; Yu et al., 2021b; Trabucco et al., 2021; Chen et al., 2022; Qi et al., 2022; Yuan et al., 2024; Chen et al., 2024) and BONET (Mashkaria et al., 2023); **(iii)** additional baselines including REINFORCE and evolutionary algorithm reported in Trabucco et al. (2022). Further details are given in Appx. D.7.1 and D.7.3.

Following standard practice (Trabucco et al., 2022), we report normalized ground-truth function values $y_n = \frac{y - y^*_{\min}}{y^*_{\max} - y^*_{\min}}$, where $y^*_{\min}$ and $y^*_{\max}$ are the minimum and maximum values in the full dataset (unseen during training). We summarize performance using both mean scores and mean normalized ranks (MNR) across tasks. We provide two sets of results: (1) $Q = 256$ queries following Krishnamoorthy et al. (2023); (2) $Q = 128$ queries following Trabucco et al. (2021); Chen et al. (2024), excluding ChEMBL. In the main text we report 100-th percentile results with $Q = 256$, and defer $Q = 128$ results and additional percentiles (e.g., 50-th) to Appx. D.7.4.

**Main results.** Tables 7 and 23 show that our method attains the best average ranks of 1.2 ($Q = 256$) and 1.8 ($Q = 128$), compared to runner-ups BONET (Mashkaria et al., 2023) and Tri-mentoring (Chen et al., 2024), which achieve 3.7 and 2.8, respectively. Our approach consistently yields the highest normalized mean values across tasks, outperforming all baselines—particularly generative inverse models—and confirming the effectiveness of our design. With $Q = 256$, we obtain the best performance on five of six tasks, with the sole exception being **ANT**. Notably, the best-performing baseline on this task, CMA-ES, exhibits a much higher variance (std. 0.928) compared to ours (0.012), underscoring its instability to initialization. With $Q = 128$, we achieve the best results on three out of five tasks. While ExPT (Nguyen et al., 2024) achieves competitive results on the remaining tasks, it heavily relies on pretraining from larger datasets and is not directly comparable to other baselines. Nevertheless, our method delivers top-tier performance across all tasks.

In Table 8, we further compare our method with MLE-LEBM and NCE (with ratio decomposition) under both LD and SVGD sampling. We find that the vanilla MLE paradigm, and even NCE, perform poorly in the offline BBO setting—likely due to the complex, highly multimodal joint latent space of $(\mathbf{x}, y)$. By contrast, our N²CE-based model produces strong results even with basic LD sampling, in line with our analysis on the importance of noise magnitude. Extended results, including ablations over M, K (Tab. 26) and robustness to query budget $Q$ (Tab. 27), are deferred to Appx. D.7.4.

## 5 CONCLUSION

In this work, we revisited NCE from the perspective of noise magnitude, an often-overlooked factor in ratio estimation. Building on this view, we introduced the N²CE framework, interpreting a "noisier" NCE objective as a controlled approximation to MLE and providing both theoretical and practical insights into its behavior. Our analysis shows that, under mild conditions, the N²CE gradient aligns with the MLE gradient when the noise magnitude is sufficiently large, and we further examined the finite-sample regime to motivate simple yet effective regularizations.

Empirically, N²CE delivers consistent improvements across image modeling, anomaly detection, and offline black-box optimization, outperforming strong baselines and demonstrating its versatility. More broadly, our results suggest that N²CE provides a practical and theoretically grounded framework for learning rewards and critics. Looking ahead, we see particular promise in extending this framework to discrete domains such as language modeling and to multimodal tasks like text-to-image and text-to-video generation, where traditional NCE remains limited.

## ACKNOWLEDGEMENT

Y. W. is partially supported by NSF DMS-2415226, DARPA W912CG25CA007 and research gift funds from Amazon and Qualcomm. We gratefully acknowledge Lambda, Inc. for providing the computational resources used in this project. We would also like to express our gratitude to the anonymous reviewer PhJz, for drawing our attention to the connection with NWJ and for contributing directly to Sec. 3.4. We also sincerely thank the anonymous AC FyXy for the careful handling of our submission and for the time and attention devoted to the review process.

## REPRODUCIBILITY STATEMENT

We provide complete statements and proofs of all theoretical results in Appx. B. Implementation details, including key hyperparameters, network architectures, and step-by-step training instructions, are described in Appx. C. Comprehensive descriptions of our experimental setups, datasets, and evaluation metrics are available in Appx. D. We will release code, data, and model checkpoints upon acceptance of this manuscript to ensure full reproducibility.

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

# A   RELATED WORK

**Noise contrastive estimation**   Noise Contrastive Estimation (NCE) (Hastie et al., 2009; Sugiyama et al., 2012; Gutmann & Hyvärinen, 2012) provides a handy interface bridging the gap between discriminative and generative learning. An interesting line of research has been devoted to estimating unnormalized density, *i.e.*, Energy-Based Models (EBMs) or LEBMs, with NCE-like objectives (Gutmann & Hyvärinen, 2012; Tu, 2007; Lazarow et al., 2017; Ceylan & Gutmann, 2018; Grover & Ermon, 2018; Gao et al., 2020a; Aneja et al., 2021). Particularly, Gao et al. (2020a) recruits a normalizing flow (Rezende et al., 2014; Kingma & Dhariwal, 2018) as the base distribution for density estimation. Aneja et al. (2021) refine the prior distribution of a pre-trained powerful VAE with noise contrastive learning. However, vanilla NCE, even with multi-stage estimation (Rhodes et al., 2020), may demonstrate less desirable results due to large gaps between target and noise distributions, dubbed as *density-chasm*. In this paper, we propose to study this fundamental problem from an often-neglected perspective: the magnitude of the noise distribution used for ratio estimation. Specifically, with an increased magnitude of the noise distribution, the gradient of the NCE objective can closely match the MLE gradient. We provide theoretical and empirical analysis showing the effectiveness of our proposed framework.

**Energy-based prior model**   As an interesting branch of EBMs (Xie et al., 2016; Nijkamp et al., 2019; 2020; Du & Mordatch, 2019; Du et al., 2021), Pang et al. (2020a) show that an EBM can serve as an informative prior model in the latent space, *i.e.*, LEBM; such a prior greatly improves the model expressivity over those with non-informative gaussian priors and brings strong performance on downstream tasks (Yu et al., 2021a; 2022; Pang et al., 2020b; 2021). However, learning LEBM requires MCMC sampling to estimate the learning gradients, which needs careful tuning and numerous iterations to converge when the target distributions are high-dimensional or highly multimodal. Commonly-used short-run MCMC (Nijkamp et al., 2019) in practice can lead to malformed energy landscapes and instability in training (Yu et al., 2022; 2024; Gao et al., 2020b; Nijkamp et al., 2020; Du & Mordatch, 2019; Du et al., 2021). In this work, we consider inducing the informative latent space with $N^2CE$ integrated into variational learning; the proposed method requires no MCMC for estimating the prior model, and shows reliable sampling quality in practice.

**Diffusion distillation**   Diffusion distillation has recently emerged as a key direction for accelerating sampling while retaining the strong generative performance of diffusion models. Early efforts such as consistency models (Song et al., 2023) and progressive distillation (Salimans & Ho, 2022) demonstrated that multi-step denoising trajectories could be compressed into a few steps, but these methods often faced trade-offs between speed and fidelity. More recent approaches refine this idea by explicitly distilling score information from pretrained diffusion models. For instance, Score Distillation Sampling (SDS) and its variants (Poole et al., 2022; Wang et al., 2023; Luo et al., 2023) leverage gradients of the diffused KL divergence to align student and teacher scores at different noise levels. Extensions such as Diffusion-GAN (Wang et al., 2022) and Distribution Matching Distillation (DMD) (Yin et al., 2024) incorporate adversarial or distribution-matching objectives to better handle mismatched supports in high-dimensional spaces. More recently, Zhou et al. (2024b) introduced Score identity Distillation (SiD), a data-free, Fisher-divergence-based method that enables one-step generation, later improved by SiDA (Zhou et al., 2024a), which integrates adversarial loss to surpass the teacher's performance. Complementary to these, reinforcement learning–based perspectives have been proposed: Yoon et al. (2024) formulated diffusion distillation as a maximum entropy inverse reinforcement learning problem, jointly training diffusion models and energy-based models, and further developed dynamic programming techniques to enable efficient updates. Collectively, these works highlight a growing trend toward principled frameworks that unify score matching, adversarial training, and control-theoretic formulations to accelerate and strengthen diffusion generation. Our work adds to this line by revisiting noise-contrastive estimation from the perspective of noise magnitude. In particular, we introduce $N^2CE$, a framework that connects "noisier" objectives to maximum likelihood, offering both theoretical insights and practical improvements across domains.

**Online BBO**   An important set-up for BBO problems is the online BBO, also referred to as active BBO in many literature (Kong et al., 2023a;b; Krishnamoorthy et al., 2023; Mashkaria et al., 2023). For online BBO, the proposed method can query the black-box function with limited times during training. A probably most well-known family is Bayesian Optimization (BO), with a large body of prior work in the area (Shahriari et al., 2015; Garnett et al., 2014; Moriconi et al., 2020a;b; Kandasamy et al., 2015; Rolland et al., 2018; de Freitas & Wang, 2013; Kusner et al., 2017; Lu et al., 2018; Gómez-

Bombarelli et al., 2018; Snoek et al., 2012; Srinivas et al., 2010; Nguyen & Osborne, 2020). Typically, BO employs surrogates such as Gaussian Processes to model the underlying function, where the surrogates are sequentially updated by querying the black-box function with newly proposed points from an uncertainty-aware acquisition function. Other branches include derivative free methods such as the cross-entropy method (Rubinstein & Kroese, 2004), methods derived from the REINFORCE trick (Williams, 1992; Rubinstein, 1997) and reward-weighted regression (Peters & Schaal, 2007), *etc*. Several practical approaches have combined these methodologies with Bayesian neural networks (Snoek et al., 2012; 2015), neural processes (Kim et al., 2018; Garnelo et al., 2018a;b), and ensembles of learned score models (Angermueller et al., 2019; 2020; Mirhoseini et al., 2020) with extensive efforts in developing advanced querying mechanisms or better approximation of the surrogates to address the online BBO problem.

**Offline BBO**   Recent works have shown significant progress in the offline BBO set-up (Brookes et al., 2019; Fannjiang & Listgarten, 2020; Kumar & Levine, 2020; Krishnamoorthy et al., 2023; Kim et al., 2024; Yu et al., 2021b; Trabucco et al., 2021; Chen et al., 2022; Qi et al., 2022; Yuan et al., 2024; Mashkaria et al., 2023; Trabucco et al., 2022; Chen et al., 2024; Fu & Levine, 2020). Among them, Kumar & Levine (2020) trains a stochastic inverse mapping $p_{\boldsymbol{\theta}}(\mathbf{x}|y) \propto \mathbb{E}_{p(\mathbf{z}|y)}[p(\mathbf{x}|\mathbf{z}, y)]$ with a conditional GAN-like model to instantiate $p(\mathbf{x}|\mathbf{z}, y)$ (Goodfellow et al., 2020; Mirza & Osindero, 2014). The method optimizes over $\mathbf{z}$ given the offline dataset maximum $y_{\max}$ and map $\mathbf{z}$ back to $\mathbf{x}$ for BBO solutions. The optimization process over $\mathbf{z}$ draw approximate samples from $p(\mathbf{z}|y)$. DDOM (Krishnamoorthy et al., 2023) consider directly parameterizing the inverse mapping $p_{\boldsymbol{\theta}}(\mathbf{x}|y)$ with a conditional diffusion model in the input design space utilizing the expressiveness of DDPMs (Ho et al., 2020). We show in this paper that our framework demonstrates significant improvements over previous parameterizations of inverse model. Forward methods (Fu & Levine, 2020; Trabucco et al., 2021; Chen et al., 2022; Qi et al., 2022; Yuan et al., 2024; Chen et al., 2024) employ gradient ascent to optimize the learned surrogates to propose candidate solutions for BBO. One critical issue, however, exists for the surrogate: the forward model can assign the out-of-training-distribution input designs $\mathbf{x}$ with erroneously large values or underestimate the true support of the distribution (Kumar & Levine, 2020; Mashkaria et al., 2023; Chen et al., 2024), especially when the set of valid inputs lies in a low-dimensional manifold in the high-dimensional input design space. Several inspiring works have focus extensively on addressing this issue (Yu et al., 2021b; Trabucco et al., 2021; Chen et al., 2022; Qi et al., 2022; Yuan et al., 2024; Chen et al., 2024). For example, Yu et al. (2021b); Trabucco et al. (2021) assign lower scores to identified outliers so that the approximated surrogate is expected to be robust. Mashkaria et al. (2023) instead mimic the optimization trajectories of black-box optimizers, and rolls out evaluation trajectories from the trained sequence model for optimization. Although the method circumvents the need of the surrogate, it assumes knowledge of the approximate value of the true optima during optimization, and can struggle relative to other approaches on certain domains. Our method learns an informative latent space suitable for BBO, and delivers strong results in comparison to prior works.

## B  DETAILED DERIVATIONS AND PROOFS

### B.1  PROOF OF PROPOSITION 3.1

**Assumption B.1** (Well-defined ratio). *The reference distribution $q_0$ does not depend on $\boldsymbol{\alpha}$ and its support covers that of $p_{\boldsymbol{\alpha}}$. Hence $r_{\boldsymbol{\alpha}}(\mathbf{x}) = p_{\boldsymbol{\alpha}}(\mathbf{x})/q_0(\mathbf{x})$ is well-defined for all $\mathbf{x}$.*

**Assumption B.2** (Differentiability and integrability). *For each $\mathbf{x}$, $p_{\boldsymbol{\alpha}}(\mathbf{x})$ is differentiable in $\boldsymbol{\alpha}$, and the score function $\nabla_{\boldsymbol{\alpha}} \log p_{\boldsymbol{\alpha}}(\mathbf{x})$ is integrable with respect to both $q_*$ and $p_{\boldsymbol{\alpha}}$.*

*Proof.*  Recall the objective

$$\mathcal{L}_M(\boldsymbol{\alpha}) = \mathbb{E}_{q_*}\left[\log \frac{r_{\boldsymbol{\alpha}}}{M + r_{\boldsymbol{\alpha}}}\right] + M\,\mathbb{E}_{q_0}\left[\log \frac{M}{M + r_{\boldsymbol{\alpha}}}\right].$$

By Assumption B.2, we may interchange $\nabla_{\boldsymbol{\alpha}}$ and $\mathbb{E}[\cdot]$. For the first term,

$$\nabla_{\boldsymbol{\alpha}} \log \frac{r_{\boldsymbol{\alpha}}}{M + r_{\boldsymbol{\alpha}}} = -\nabla_{\boldsymbol{\alpha}} \log\left(1 + \frac{M}{r_{\boldsymbol{\alpha}}}\right) = \frac{M}{M + r_{\boldsymbol{\alpha}}}\,\nabla_{\boldsymbol{\alpha}} \log r_{\boldsymbol{\alpha}}.$$

Since $q_0$ does not depend on $\boldsymbol{\alpha}$, $\nabla_{\boldsymbol{\alpha}} \log r_{\boldsymbol{\alpha}} = \nabla_{\boldsymbol{\alpha}} \log p_{\boldsymbol{\alpha}}$. Hence

$$\nabla_{\boldsymbol{\alpha}}\,\mathbb{E}_{q_*}\left[\log \frac{r_{\boldsymbol{\alpha}}}{M + r_{\boldsymbol{\alpha}}}\right] = \mathbb{E}_{q_*}\left[\frac{M}{M + r_{\boldsymbol{\alpha}}}\,\nabla_{\boldsymbol{\alpha}} \log p_{\boldsymbol{\alpha}}\right].$$

For the second term,

$$\nabla_{\boldsymbol{\alpha}} \log \frac{M}{M + r_{\boldsymbol{\alpha}}} = -\frac{1}{M + r_{\boldsymbol{\alpha}}}\nabla_{\boldsymbol{\alpha}} r_{\boldsymbol{\alpha}} = -\frac{r_{\boldsymbol{\alpha}}}{M + r_{\boldsymbol{\alpha}}}\,\nabla_{\boldsymbol{\alpha}} \log p_{\boldsymbol{\alpha}},$$

so

$$M\,\nabla_{\boldsymbol{\alpha}}\,\mathbb{E}_{q_0}\left[\log \frac{M}{M + r_{\boldsymbol{\alpha}}}\right] = -M\,\mathbb{E}_{q_0}\left[\frac{r_{\boldsymbol{\alpha}}}{M + r_{\boldsymbol{\alpha}}}\,\nabla_{\boldsymbol{\alpha}} \log p_{\boldsymbol{\alpha}}\right].$$

Using the change of measure $\mathbb{E}_{q_0}[r_{\boldsymbol{\alpha}} g] = \mathbb{E}_{p_{\boldsymbol{\alpha}}}[g]$ from Assumption B.1, we get

$$M\,\mathbb{E}_{q_0}\left[\frac{r_{\boldsymbol{\alpha}}}{M + r_{\boldsymbol{\alpha}}}\,g\right] = \mathbb{E}_{p_{\boldsymbol{\alpha}}}\left[\frac{M}{M + r_{\boldsymbol{\alpha}}}\,g\right].$$

Applying this with $g = \nabla_{\boldsymbol{\alpha}} \log p_{\boldsymbol{\alpha}}$ yields the stated gradient identity.

For the limit, note that $\frac{M}{M+r_{\boldsymbol{\alpha}}} \to 1$ pointwise as $M \to \infty$ and $0 \le \frac{M}{M+r_{\boldsymbol{\alpha}}} \le 1$. By Assumption B.2, dominated convergence applies, giving

$$\lim_{M \to \infty} \mathbb{E}_{q_*}\left[\frac{M}{M + r_{\boldsymbol{\alpha}}}\,\nabla_{\boldsymbol{\alpha}} \log p_{\boldsymbol{\alpha}}\right] = \mathbb{E}_{q_*}[\nabla_{\boldsymbol{\alpha}} \log p_{\boldsymbol{\alpha}}],$$

and similarly for the expectation under $p_{\boldsymbol{\alpha}}$. Combining finishes the proof.  $\square$

## B.2 Proof of Proposition 3.3

*Proof.* Since $\nabla_{\boldsymbol{\alpha}} \widehat{\mathcal{L}}_M(\boldsymbol{\alpha})$ is an unbiased estimator of $\nabla_{\boldsymbol{\alpha}} \mathcal{L}_M(\boldsymbol{\alpha})$, we have the standard bias–variance decomposition:

$$
\mathbb{E}\left\| \nabla_{\boldsymbol{\alpha}} \widehat{\mathcal{L}}_M(\boldsymbol{\alpha}) - \left( \mathbb{E}_{q_*}[\nabla_{\boldsymbol{\alpha}} f_{\boldsymbol{\alpha}}] - \mathbb{E}_{p_{\boldsymbol{\alpha}}}[\nabla_{\boldsymbol{\alpha}} f_{\boldsymbol{\alpha}}] \right) \right\|^2
$$
$$
= \mathbf{Var}\left( \nabla_{\boldsymbol{\alpha}} \widehat{\mathcal{L}}_M(\boldsymbol{\alpha}) \right) + \left\| \nabla_{\boldsymbol{\alpha}} \mathcal{L}_M(\boldsymbol{\alpha}) - \left( \mathbb{E}_{q_*}[\nabla_{\boldsymbol{\alpha}} f_{\boldsymbol{\alpha}}] - \mathbb{E}_{p_{\boldsymbol{\alpha}}}[\nabla_{\boldsymbol{\alpha}} f_{\boldsymbol{\alpha}}] \right) \right\|^2. \tag{10}
$$

**Variance term.** Since $\{(\mathbf{x}_0^i, \mathbf{x}_*^i)\}_{i=1}^n \sim (q_0, q_*)$ are i.i.d.,

$$
\mathbf{Var}\left( \nabla_{\boldsymbol{\alpha}} \widehat{\mathcal{L}}_M(\boldsymbol{\alpha}) \right) = \frac{1}{n} \mathbf{Var}\left( \nabla_{\boldsymbol{\alpha}} \log \frac{r_{\boldsymbol{\alpha}}(\mathbf{x}_*^i)}{M + r_{\boldsymbol{\alpha}}(\mathbf{x}_*^i)} + M \nabla_{\boldsymbol{\alpha}} \log \frac{M}{M + r_{\boldsymbol{\alpha}}(\mathbf{x}_0^i)} \right)
$$
$$
= \frac{1}{n} \mathbf{Var}\left( \frac{M}{M + r_{\boldsymbol{\alpha}}(\mathbf{x}_*^i)} \nabla_{\boldsymbol{\alpha}} \log r_{\boldsymbol{\alpha}}(\mathbf{x}_*^i) - \frac{M}{M + r_{\boldsymbol{\alpha}}(\mathbf{x}_0^i)} \nabla_{\boldsymbol{\alpha}} r_{\boldsymbol{\alpha}}(\mathbf{x}_0^i) \right). \tag{11}
$$

Using $(a - b)^2 \le 2a^2 + 2b^2$, we obtain

$$
\mathbf{Var}\left( \nabla_{\boldsymbol{\alpha}} \widehat{\mathcal{L}}_M(\boldsymbol{\alpha}) \right) \le \frac{2}{n} \mathbb{E}\left[ \frac{M}{M + r_{\boldsymbol{\alpha}}(\mathbf{x}_*^i)} \nabla_{\boldsymbol{\alpha}} \log r_{\boldsymbol{\alpha}}(\mathbf{x}_*^i) \right]^2 + \frac{2}{n} \mathbb{E}\left[ \frac{M}{M + r_{\boldsymbol{\alpha}}(\mathbf{x}_0^i)} \nabla_{\boldsymbol{\alpha}} r_{\boldsymbol{\alpha}}(\mathbf{x}_0^i) \right]^2. \tag{12}
$$

Since $\frac{M}{M+r} \le 1$ and $\frac{M}{M+r} \le \frac{M}{r}$ for any $M, r > 0$, we further bound

$$
\mathbf{Var}\left( \nabla_{\boldsymbol{\alpha}} \widehat{\mathcal{L}}_M(\boldsymbol{\alpha}) \right) \le \frac{2}{n} \mathbb{E}\left[ \left\| \nabla_{\boldsymbol{\alpha}} \log r_{\boldsymbol{\alpha}}(\mathbf{x}_*^i) \right\|^2 \right] \tag{13}
$$
$$
+ \frac{2}{n} \min\left\{ M^2 \mathbb{E}\left[ \left\| \nabla_{\boldsymbol{\alpha}} \log r_{\boldsymbol{\alpha}}(\mathbf{x}_0^i) \right\|^2 \right], \ \mathbb{E}\left[ \left\| \nabla_{\boldsymbol{\alpha}} r_{\boldsymbol{\alpha}}(\mathbf{x}_0^i) \right\|^2 \right] \right\}.
$$

Thus the variance is controlled by the second moments of the score (or equivalently $\nabla_{\boldsymbol{\alpha}} r_{\boldsymbol{\alpha}}$), and in the typical regime where the first branch dominates, it decays as $O(M^2/n)$.

**Bias term.** From the proof of Proposition 3.1,

$$
\left\| \nabla_{\boldsymbol{\alpha}} \mathcal{L}_M(\boldsymbol{\alpha}) - \left( \mathbb{E}_{q_*}[\nabla_{\boldsymbol{\alpha}} f_{\boldsymbol{\alpha}}] - \mathbb{E}_{p_{\boldsymbol{\alpha}}}[\nabla_{\boldsymbol{\alpha}} f_{\boldsymbol{\alpha}}] \right) \right\|^2 = \left\| \int_{\mathbf{x}} \frac{r_{\boldsymbol{\alpha}}(\mathbf{x})}{M + r_{\boldsymbol{\alpha}}(\mathbf{x})} \left( q_*(\mathbf{x}) - p_{\boldsymbol{\alpha}}(\mathbf{x}) \right) \nabla_{\boldsymbol{\alpha}} f_{\boldsymbol{\alpha}}(\mathbf{x}) \, d\mathbf{x} \right\|^2
$$
$$
\le \frac{1}{M^2} \left\| \int_{\mathbf{x}} r_{\boldsymbol{\alpha}}(\mathbf{x}) \left( q_*(\mathbf{x}) - p_{\boldsymbol{\alpha}}(\mathbf{x}) \right) \nabla_{\boldsymbol{\alpha}} f_{\boldsymbol{\alpha}}(\mathbf{x}) \, d\mathbf{x} \right\|^2. \tag{14}
$$

Hence the squared bias vanishes at rate $O(1/M^2)$.

**Conclusion.** Combining equation 13 and equation 14, the mean squared error of $\nabla_{\boldsymbol{\alpha}} \widehat{\mathcal{L}}_M(\boldsymbol{\alpha})$ satisfies

$$
\mathbb{E}\left\| \nabla_{\boldsymbol{\alpha}} \widehat{\mathcal{L}}_M(\boldsymbol{\alpha}) - \left( \mathbb{E}_{q_*}[\nabla_{\boldsymbol{\alpha}} f_{\boldsymbol{\alpha}}] - \mathbb{E}_{p_{\boldsymbol{\alpha}}}[\nabla_{\boldsymbol{\alpha}} f_{\boldsymbol{\alpha}}] \right) \right\|^2 = O\left( \frac{M^2}{n} \right) + O\left( \frac{1}{M^2} \right),
$$

which completes the proof. $\qquad \square$

### B.3 Proof of Theorem 3.2

As in Liu et al. (2021), we work with the exponential family

$$p_{\boldsymbol{\alpha}}(\mathbf{x}) = \exp\Big(\boldsymbol{\alpha}^\top \tilde{T}(\mathbf{x}) - \log Z(\boldsymbol{\alpha})\Big),$$

where $\boldsymbol{\alpha} \in \mathbb{R}^m$ is the natural parameter, $\tilde{T}(\mathbf{x}) \in \mathbb{R}^m$ are sufficient statistics, and $Z(\boldsymbol{\alpha})$ is the partition function. For convenience in derivatives, we also use the extended parameterization

$$\boldsymbol{\tau} := [\boldsymbol{\alpha}, \alpha_Z], \quad \alpha_Z := \log Z(\boldsymbol{\alpha}), \qquad T(\mathbf{x}) := [\tilde{T}(\mathbf{x}), -1],$$

so that $p_{\boldsymbol{\tau}}(\mathbf{x}) = \exp(\boldsymbol{\tau}^\top T(\mathbf{x}))$. Expectations $\mathbb{E}_{\boldsymbol{\tau}}[\cdot]$ are taken w.r.t. $p_{\boldsymbol{\tau}}$.

**Assumption B.3** (Exponential-family regularity). *The map $\boldsymbol{\alpha} \mapsto \log Z(\boldsymbol{\alpha})$ is twice continuously differentiable on the parameter set $\mathcal{A}$. Equivalently, $p_{\boldsymbol{\alpha}}(\mathbf{x})$ is $C^2$ in $\boldsymbol{\alpha}$ and differentiation under the integral is valid for the quantities considered.*

**Assumption B.4** (Reference coverage and Fisher regularity). *The reference $q_0$ covers the support of $p_{\boldsymbol{\alpha}}$, so $r_{\boldsymbol{\alpha}}(\mathbf{x}) = p_{\boldsymbol{\alpha}}(\mathbf{x})/q_0(\mathbf{x})$ is well-defined. Moreover, there exist constants $0 < \lambda_{\min} \le \lambda_{\max} < \infty$ such that, for all extended parameters $\boldsymbol{\tau}$,*

$$\lambda_{\min} I \preceq \mathbb{E}_{\boldsymbol{\tau}}\big[T(\mathbf{x})T(\mathbf{x})^\top\big] \preceq \lambda_{\max} I.$$

**Assumption B.5** (Bounded parameter set). $\|\boldsymbol{\alpha}\|_2 \le \omega$ *for all* $\boldsymbol{\alpha} \in \mathcal{A}$.

---

**Lemma B.6** (Convexity). *Let $r_{\boldsymbol{\alpha}} = p_{\boldsymbol{\alpha}}/q_0$ with $p_{\boldsymbol{\alpha}}$ in the exponential family. Under Assumptions B.3 and B.4, the negative $N^2CE$ objective in Eq. (5) is convex in $\boldsymbol{\alpha}$.*

---

*Proof.* Use the extended parameterization $p_{\boldsymbol{\tau}}(\mathbf{x}) = \exp(\boldsymbol{\tau}^\top T(\mathbf{x}))$. By Assumption B.3 and finite second moments implied by Assumption B.4, we may compute the derivatives inside the integral.

The gradient can be written as

$$\nabla_{\boldsymbol{\tau}} \mathcal{L}_M(\boldsymbol{\tau}) = \int \frac{M \, q_0(\mathbf{x})}{p_{\boldsymbol{\tau}}(\mathbf{x}) + M q_0(\mathbf{x})} \big(q_*(\mathbf{x}) - p_{\boldsymbol{\tau}}(\mathbf{x})\big) T(\mathbf{x}) \, d\mathbf{x}. \tag{15}$$

Differentiating once more gives the Hessian

$$\nabla_{\boldsymbol{\tau}}^2 \mathcal{L}_M(\boldsymbol{\tau}) = -\int \frac{M \, p_{\boldsymbol{\tau}}(\mathbf{x}) \, q_0(\mathbf{x})}{\big(p_{\boldsymbol{\tau}}(\mathbf{x}) + M q_0(\mathbf{x})\big)^2} \big(q_*(\mathbf{x}) + M q_0(\mathbf{x})\big) T(\mathbf{x})T(\mathbf{x})^\top \, d\mathbf{x}. \tag{16}$$

For each $\mathbf{x}$, the scalar weight

$$\frac{M \, p_{\boldsymbol{\tau}}(\mathbf{x}) \, q_0(\mathbf{x})}{\big(p_{\boldsymbol{\tau}}(\mathbf{x}) + M q_0(\mathbf{x})\big)^2} \big(q_*(\mathbf{x}) + M q_0(\mathbf{x})\big) \ge 0,$$

and $T(\mathbf{x})T(\mathbf{x})^\top \succeq 0$. Hence the integrand in equation 16 is positive semidefinite; with the leading minus sign, the Hessian is negative semidefinite for all $\boldsymbol{\tau}$. Therefore $-\mathcal{L}_M$ is convex in $\boldsymbol{\alpha}$. $\qquad \square$

---

**Lemma B.7** (Condition number). *Under Assumptions B.3 and B.4,*

$$\lim_{M \to \infty} \big(-\nabla_{\boldsymbol{\tau}}^2 \mathcal{L}_M(\boldsymbol{\tau}^*)\big) = \mathbb{E}_{\boldsymbol{\tau}^*}\big[T(\mathbf{x})T(\mathbf{x})^\top\big], \tag{17}$$

*and hence*

$$\kappa^* = \kappa\Big(-\lim_{M \to \infty} \nabla_{\boldsymbol{\tau}}^2 \mathcal{L}_M(\boldsymbol{\tau}^*)\Big) \le \frac{\lambda_{\max}}{\lambda_{\min}}. \tag{18}$$

*Moreover, for large but finite $M$,*

$$\kappa\big(-\nabla_{\boldsymbol{\tau}}^2 \mathcal{L}_M(\boldsymbol{\tau}^*)\big) \le \frac{\lambda_{\max}}{\lambda_{\min}} \big(1 + O(1/M)\big). \tag{19}$$

---

*Proof.* From equation 16, for any $\boldsymbol{\tau}$ we can write

$$\nabla_{\boldsymbol{\tau}}^2 \mathcal{L}_M(\boldsymbol{\tau}) = -\mathbb{E}_{p_{\boldsymbol{\tau}}}\big[\phi_M(\mathbf{x}; \boldsymbol{\tau})\, T(\mathbf{x})T(\mathbf{x})^\top\big], \qquad \phi_M(\mathbf{x}; \boldsymbol{\tau}) = \frac{1 + \frac{q_*(\mathbf{x})}{Mq_0(\mathbf{x})}}{\big(1 + \frac{p_{\boldsymbol{\tau}}(\mathbf{x})}{Mq_0(\mathbf{x})}\big)^2}.$$

Pointwise, $\phi_M(\mathbf{x}; \boldsymbol{\tau}) \to 1$ as $M \to \infty$, and $0 < \phi_M(\mathbf{x}; \boldsymbol{\tau}) \le 1 + O(1/M)$ uniformly. By Assumption B.4, $\mathbb{E}_{p_{\boldsymbol{\tau}}}\big[\|T(\mathbf{x})\|^2\big] < \infty$, so dominated convergence applies. Assumption B.3 ensures differentiation and expectations are well-defined. Hence

$$\lim_{M \to \infty} \nabla_{\boldsymbol{\tau}}^2 \mathcal{L}_M(\boldsymbol{\tau}) = -\mathbb{E}_{p_{\boldsymbol{\tau}}}[T(\mathbf{x})T(\mathbf{x})^\top].$$

Evaluating at the optimum $\boldsymbol{\tau}^*$ yields equation 17. By Assumption B.4, the eigenvalues of $\mathbb{E}_{\boldsymbol{\tau}^*}[T(\mathbf{x})T(\mathbf{x})^\top]$ lie in $[\lambda_{\min}, \lambda_{\max}]$, which gives the condition-number bound equation 18.

For finite $M$, write

$$-\nabla_{\boldsymbol{\tau}}^2 \mathcal{L}_M(\boldsymbol{\tau}^*) = \mathbb{E}_{\boldsymbol{\tau}^*}[T(\mathbf{x})T(\mathbf{x})^\top] + R_M, \qquad R_M = \mathbb{E}_{p_{\boldsymbol{\tau}^*}}\big[(1 - \phi_M(\mathbf{x}; \boldsymbol{\tau}^*))\, T(\mathbf{x})T(\mathbf{x})^\top\big].$$

Since $1 - \phi_M(\mathbf{x}; \boldsymbol{\tau}^*) = O(1/M)$ and $\mathbb{E}_{\boldsymbol{\tau}^*}[\|T(\mathbf{x})\|^2] < \infty$, we have $\|R_M\| = O(1/M)$. Standard eigenvalue perturbation bounds then give equation 19. $\qquad\square$

---

**Theorem B.8** (Convergence rate). *Let $\mathcal{L}_M$ denote the $N^2CE$ objective. Under Assumptions B.3 and B.4, the Hessian of $-\mathcal{L}_M$ satisfies the local smoothness and curvature bounds required in Theorem 5.1 of Liu et al. (2021). Consequently, normalized gradient descent on $\mathcal{L}_M$ with $M \to \infty$ converges to the optimum $\boldsymbol{\tau}^*$ at a rate polynomial in the condition number. In particular, for any tolerance $\delta > 0$,*

$$T \le \left(\frac{\lambda_{\max}}{\lambda_{\min}}\right)^3 \frac{\|\boldsymbol{\tau}_0 - \boldsymbol{\tau}^*\|_2^2}{\delta^2}$$

*iterations suffice to obtain some iterate $\boldsymbol{\tau}_t$ with $\|\boldsymbol{\tau}_t - \boldsymbol{\tau}^*\|_2 \le \delta$.*

---

*Proof.* This follows from Theorem 5.1 of Liu et al. (2021), together with convexity (Lemma B.6) and the condition number bound (Lemma B.7), which verify the smoothness and curvature assumptions in that theorem. $\qquad\square$

**Remark.** The $N^2CE$ loss inherits favorable convexity and well-conditioned curvature from the exponential family, ensuring that normalized gradient methods converge efficiently to the optimum.

## C  NETWORK ARCHITECTURE AND IMPLEMENTATION DETAILS

**Code**  We promise to release code, data and model checkpoints upon acceptance of this manuscript.

### C.1  HOW TO LEARN A LATENT SPACE MODEL WITH N$^2$CE

**ELBO**  We show how the proposed form can be integrated into the variational learning scheme to learn an accurate and explicit density $p_{\boldsymbol{\alpha}}$. We can first see that the target distribution for $p_{\boldsymbol{\alpha}}$ is the aggregated posterior $q_{K+1}(\mathbf{z}) = \int_{\mathbf{x}} q_{\boldsymbol{\phi}}(\mathbf{z}|\mathbf{x})p(\mathbf{x})d\mathbf{x}$ (Tomczak & Welling, 2018), where $q_{\boldsymbol{\phi}}$ denotes the posterior distribution parameterized by the encoder network. When considering ratio decomposition, we can form a corresponding KL divergence which leads to the following ELBO:

$$\text{ELBO}_{\boldsymbol{\theta},\boldsymbol{\phi}} = \mathbb{E}_{q_{\boldsymbol{\phi}}}[\log p_{\boldsymbol{\beta}}(\mathbf{x}|\mathbf{z})] - D_{\text{KL}}(q_{\boldsymbol{\phi}}\|r_{\boldsymbol{\alpha}_K}q_K) - \sum_{k=0}^{K-1} D_{\text{KL}}(q_{k+1}\|r_{\boldsymbol{\alpha}_k}q_k), \tag{20}$$

which is a valid ELBO by the non-negativity of the KL Divergence. $\boldsymbol{\beta}$ denotes the decoder network.

**Training algorithms**  We directly learn the log-ratio estimator $\tilde{f}_{\boldsymbol{\alpha}_k} = \log r_{\boldsymbol{\alpha}_k}$ for implementation. Let $\sigma(\cdot)$ denote the sigmoid function, the gradient for learning each ratio estimator $\{p_{\boldsymbol{\alpha}_k}\}_{k=0}^{K}$ can be derived from Eq. (5) as:

$$\nabla_{\boldsymbol{\alpha}_k} \text{ELBO} \approx \nabla_{\boldsymbol{\alpha}_k}\mathbb{E}_{q_{k+1}}\left[\log \sigma\left(\tilde{f}_{\boldsymbol{\alpha}_k} - \log M\right)\right] + \nabla_{\boldsymbol{\alpha}_k} M\mathbb{E}_{q_k}\left[\log\left(1 - \sigma\left(\tilde{f}_{\boldsymbol{\alpha}_k} - \log M\right)\right)\right]. \tag{21}$$

Let $\boldsymbol{\psi} = \{\boldsymbol{\beta}, \boldsymbol{\phi}\}$ collect the parameters of the decoder and encoder networks. For $q_{\boldsymbol{\phi}}(\mathbf{z}|\mathbf{x})$ and $p_{\boldsymbol{\beta}}(\mathbf{x}|\mathbf{z})$, we can calculate the learning gradient as follows:

$$\nabla_{\boldsymbol{\psi}} \text{ELBO} = \nabla_{\boldsymbol{\psi}}\mathbb{E}_{q_{\boldsymbol{\phi}}}[\log p_{\boldsymbol{\beta}}] - \nabla_{\boldsymbol{\phi}}D_{\text{KL}}(q_{\boldsymbol{\phi}}\|p_0) - \nabla_{\boldsymbol{\phi}}\mathbb{E}_{q_{\boldsymbol{\phi}}}\left[\sum_{k=0}^{K} \tilde{f}_{\boldsymbol{\alpha}_k}\right]. \tag{22}$$

Alg. 1 summarizes the training procedure.

**Implementation**  When optimizing the ELBO in Eq. (20), we randomly choose one intermediate stage to optimize at each training iteration as in Ho et al. (2020). To construct these intermediate distributions, we follow Yu et al. (2022) to spherically interpolates between pairs of samples $\mathbf{z}_{K+1} \sim q_{K+1}$ and $\mathbf{z}_0 \sim q_0$; samples from the $k$-th intermediate distribution $q_k$ are: $\mathbf{z}_k = \sqrt{1 - \sigma_k^2}\mathbf{z}_0 + \sigma_k\mathbf{z}_{K+1}$, where the $\{\sigma_k\}_{k=1}^{K}$ form an increasing sequence from 0 to 1 controlling the distance between these implicitly defined distributions $\{q_k\}_{k=1}^{K}$. We use a linear schedule to specify $\sigma_k^2$ and construct the intermediate distributions $\{q_k\}_{k=0}^{K}$.

---

**Algorithm 1** Variational Learning with N$^2$CE

---

**Input:** dataset $\mathcal{D} : \{\mathbf{x}^{(i)}\}_{i=1}^{N}$, params. $\left(\{\boldsymbol{\alpha}_k\}_{k=0}^{K}, \boldsymbol{\beta}, \boldsymbol{\phi}\right)$,
**Output:** updated params. $\left(\{\boldsymbol{\alpha}_k^*\}_{k=0}^{K}, \boldsymbol{\beta}^*, \boldsymbol{\phi}^*\right)$

**repeat**
  **posterior sampling:** For each pair of $\{\mathbf{x}^{(i)}\}$, sample $\mathbf{z}_{K+1}^{(i)} \sim q_{\boldsymbol{\phi}}(\mathbf{z}|\mathbf{x}^{(i)})$ using inference network.
  **forming** $\{q_k\}_{k=1}^{K}$**:** For each sample $\mathbf{z}_{K+1}^{(i)}$, sample $k \sim \text{Unif}(\{0, ..., K\})$ and obtain $\left(\mathbf{z}_k^{(i)}, \mathbf{z}_{k+1}^{(i)}\right) \sim (q_k, q_{k+1})$ using $\mathbf{z}_k = \sqrt{1 - \sigma_k^2}\mathbf{z}_0 + \sigma_k\mathbf{z}_{K+1}$.
  **learning** $\{\boldsymbol{\alpha}_k\}_{k=0}^{K}$**:** Update $\{\boldsymbol{\alpha}_k\}_{k=0}^{K}$ with Eq. (21).
  **learning** $\boldsymbol{\psi} = (\boldsymbol{\beta}, \boldsymbol{\phi})$**:** Update $\boldsymbol{\psi}$ with Eq. (22).
**until** converged.

---

### C.2  IMAGE MODELING AND ANOMALY DETECTION

**Architecture**  We provide detailed network architecture for the log-ratio estimator for each stage $\tilde{f}_{\boldsymbol{\alpha}_k}(\mathbf{z}, k)$ in Tab. 11. For the VAE model, the decoder network has a simple deconvolution structure similar to DCGAN (Radford et al., 2015) shown in Tab. 9. The encoder network to embed the observed images has a fully convolutional structure, as shown in Tab. 10.

Table 9: **Network structures of the generator networks** used for the SVHN, CelebA, CIFAR-10, CelebA-HQ and MNIST (from top to bottom) datasets. ConvT($n$) indicates a transposed convolutional operation with $n$ output channels. We use `ngf=64` for the SVHN dataset and `ngf=128` for the rest. LReLU indicates the Leaky-ReLU activation function. The slope in Leaky ReLU is set to be 0.2.

| Layers | Out Size | Stride |
|---|---|---|
| Input: **z** | 1x1x100 | - |
| 4x4 ConvT(ngf x 8), LReLU | 4x4x(ngf x 8) | 1 |
| 4x4 ConvT(ngf x 4), LReLU | 8x8x(ngf x 4) | 2 |
| 4x4 ConvT(ngf x 2), LReLU | 16x16x(ngf x 2) | 2 |
| 4x4 ConvT(3), Tanh | 32x32x3 | 2 |
| **Layers** | **Out Size** | **Stride** |
| Input: **z** | 1x1x100 | - |
| 4x4 ConvT(ngf x 8), LReLU | 4x4x(ngf x 8) | 1 |
| 4x4 ConvT(ngf x 4), LReLU | 8x8x(ngf x 4) | 2 |
| 4x4 ConvT(ngf x 2), LReLU | 16x16x(ngf x 2) | 2 |
| 4x4 ConvT(ngf x 1), LReLU | 32x32x(ngf x 1) | 2 |
| 4x4 ConvT(3), Tanh | 64x64x3 | 2 |
| **Layers** | **Out Size** | **Stride** |
| Input: **z** | 1x1x128 | - |
| 8x8 ConvT(ngf x 8), LReLU | 8x8x(ngf x 8) | 1 |
| 4x4 ConvT(ngf x 4), LReLU | 16x16x(ngf x 4) | 2 |
| 4x4 ConvT(ngf x 2), LReLU | 32x32x(ngf x 2) | 2 |
| 3x3 ConvT(3), Tanh | 32x32x3 | 1 |
| **Layers** | **Out Size** | **Stride** |
| Input: **z** | 1x1x128 | - |
| 4x4 ConvT(ngf x 16), LReLU | 4x4x(ngf x 16) | 1 |
| 4x4 ConvT(ngf x 8), LReLU | 8x8x(ngf x 8) | 2 |
| 4x4 ConvT(ngf x 4), LReLU | 16x16x(ngf x 4) | 2 |
| 4x4 ConvT(ngf x 4), LReLU | 32x32x(ngf x 4) | 2 |
| 4x4 ConvT(ngf x 2), LReLU | 64x64x(ngf x 2) | 2 |
| 4x4 ConvT(ngf x 1), LReLU | 128x128x(ngfx1) | 2 |
| 4x4 ConvT(3), Tanh | 256x256x3 | 2 |
| **Layers** | **Out Size** | **Stride** |
| Input: **z** | 1x1x8 | - |
| 7x7 ConvT(ngf x 8), LReLU | 7x7x(ngf x 8) | 1 |
| 4x4 ConvT(ngf x 4), LReLU | 14x14x(ngf x 4) | 2 |
| 4x4 ConvT(ngf x 2), LReLU | 28x28x(ngf x 2) | 2 |
| 3x3 ConvT(1), Tanh | 28x28x1 | 1 |

**Hyperparameters and training details**  In the image modeling experiments, for the MLE-LEBM baseline, we run LD for $T = 60$ iterations for prior updates during training with a step size of $s = 0.4$. For test time sampling from LEBMs trained with different objectives, we consistently LD for $T = 100$ with the step size of $s = 0.4$ to draw samples from $p_{\alpha}$. In the anomaly detection experiments, we follow the same experiment configuration as used in Yu et al. (2024).

We use the linear schedule as in and Yu et al. (2022) to specify $\sigma_k^2$ and construct the intermediate distributions $\{q_k\}_{k=0}^K$. We set the number of stages to 3. Specifically, the sequence of $\{\sigma_k^2\}_{k=0}^3$ is: $\{\sigma_k^2\}_{k=0}^3 = [0.01, 0.69175489, 0.92238785, 0.99974058]$. To estimate the Eq. (5), we can use Monte-Carlo average: $\mathcal{L}_M(\boldsymbol{\alpha}_k) \approx \frac{1}{n_1}\sum_{i=1}^{n_1}\left[\log \frac{r_{\boldsymbol{\alpha}_k}(\mathbf{z}_i)}{M+r_{\boldsymbol{\alpha}_k}(\mathbf{z}_i)}\right] + \frac{M}{n_2}\sum_{j=1}^{n_2}\left[\log \frac{M}{M+r_{\boldsymbol{\alpha}_k}(\mathbf{z}_j)}\right]$, where $\mathbf{z}_i \sim q_{k+1}, \mathbf{z}_j \sim q_k$. We follow Mnih & Teh (2012) and set $n_2 = Mn_1$ in practice, where $M = 100$ for all experiments; $n_1$ is the batch size. When pre-training the VAE model, we set the weight for reconstruction to $[1, 1, 100, 1]$ on the four datasets and KL weights to 1; when training jointly with the LEBM, we set the weight for reconstructing $\mathbf{x}$ to $[5, 5, 2, 1]$ for the term $\mathbb{E}_{q_{\phi}}[\log p_{\boldsymbol{\beta}}(\mathbf{x}|\mathbf{z})]$ on SVHN, CelebA64, CIFAR10 and CelebAHQ datasets, respectively. We set the weight for controlling the KL term between $q_{\phi}$ and $p_{\alpha}$ to 1, *i.e.*, $D_{\mathrm{KL}}(q_{\phi}\|r_{\boldsymbol{\alpha}_K}q_K) - \sum_{k=0}^{K-1} D_{\mathrm{KL}}(q_{k+1}\|r_{\boldsymbol{\alpha}_k}q_k)$. Other hyperparameters are shared across all the experiments.

Table 10: **Network structures of the encoder networks** used for the SVHN, CelebA, CIFAR-10, CelebA-HQ and MNIST (from top to bottom) datasets. Conv($n$)Norm indicates a convolutional operation with $n$ output channels followed by the Instance Normalization (Ulyanov et al., 2016). We use `nif=64` for all the datasets. LReLU indicates the Leaky-ReLU activation function. The slope in Leaky ReLU is set to be 0.2.

| Layers | Out Size | Stride |
|---|---|---|
| Input: **x** | 32x32x3 | - |
| 3x3 Conv(nif x 1)Norm, LReLU | 32x32x(nif x 1) | 1 |
| 4x4 Conv(nif x 2)Norm, LReLU | 16x16x(nif x 2) | 2 |
| 4x4 Conv(nif x 4)Norm, LReLU | 8x8x(nif x 4) | 2 |
| 4x4 Conv(nif x 8)Norm, LReLU | 4x4x(nif x 8) | 2 |
| 4x4 Conv(nemb)Norm, LReLU | 1x1x(nz) | 1 |
| **Layers** | **Out Size** | **Stride** |
| Input: **x** | 64x64x3 | - |
| 3x3 Conv(nif x 1)Norm, LReLU | 64x64x(nif x 1) | 1 |
| 4x4 Conv(nif x 2)Norm, LReLU | 32x32x(nif x 2) | 2 |
| 4x4 Conv(nif x 4)Norm, LReLU | 16x16x(nif x 4) | 2 |
| 4x4 Conv(nif x 8)Norm, LReLU | 8x8x(nif x 8) | 2 |
| 4x4 Conv(nif x 8)Norm, LReLU | 4x4x(nif x 8) | 2 |
| 4x4 Conv(nemb)Norm, LReLU | 1x1x(nz) | 1 |
| **Layers** | **Out Size** | **Stride** |
| Input: **x** | 32x32x3 | - |
| 3x3 Conv(nif x 1)Norm, LReLU | 32x32x(nif x 1) | 1 |
| 4x4 Conv(nif x 2)Norm, LReLU | 16x16x(nif x 2) | 2 |
| 4x4 Conv(nif x 4)Norm, LReLU | 8x8x(nif x 4) | 2 |
| 4x4 Conv(nif x 8)Norm, LReLU | 4x4x(nif x 8) | 2 |
| 4x4 Conv(nz)Norm, LReLU | 1x1x(nz) | 1 |
| **Layers** | **Out Size** | **Stride** |
| Input: **x** | 256x256x3 | - |
| 3x3 Conv(nif x 1)Norm, LReLU | 256x256x(nif x 1) | 1 |
| 4x4 Conv(nif x 2)Norm, LReLU | 128x128x(nif x 2) | 2 |
| 4x4 Conv(nif x 4)Norm, LReLU | 64x64x(nif x 4) | 2 |
| 4x4 Conv(nif x 4)Norm, LReLU | 32x32x(nif x 4) | 2 |
| 4x4 Conv(nif x 8)Norm, LReLU | 16x16x(nif x 8) | 2 |
| 4x4 Conv(nif x 8)Norm, LReLU | 8x8x(nif x 8) | 2 |
| 4x4 Conv(nif x 8)Norm, LReLU | 4x4x(nif x 8) | 2 |
| 4x4 Conv(nz)Norm, LReLU | 1x1x(nz) | 1 |
| **Layers** | **Out Size** | **Stride** |
| Input: **x** | 28x28x3 | - |
| 3x3 Conv(nif x 1)Norm, LReLU | 28x28x(nif x 1) | 1 |
| 4x4 Conv(nif x 2)Norm, LReLU | 14x14x(nif x 2) | 2 |
| 4x4 Conv(nif x 4)Norm, LReLU | 7x7x(nif x 4) | 2 |
| 4x4 Conv(nif x 8)Norm, LReLU | 3x3x(nif x 8) | 2 |
| 3x3 Conv(nz)Norm, LReLU | 1x1x(nz) | 1 |

The parameters of all the networks are initialized with the default pytorch methods (Paszke et al., 2019). We use the Adam optimizer (Kingma & Ba, 2014) with $\beta_1 = 0.5$ and $\beta_2 = 0.999$ to train the decoder and encoder networks; the log-ratio estimator network is trained with default $(\beta_1, \beta_2)$. The initial learning rates of the generator, encoder and log-ratio networks are `2e-4`. We further perform gradient clipping by setting the maximal gradient norm as 100 for these networks. We run the experiments on a A6000 GPU with the batch size of 128. Training converges within 100K iterations on all the datasets.

### C.3 REWARD AND CRITIC LEARNING

We adapt the objectives in prior work as follows:

(i) For DxMI (Yoon et al., 2024), we replace the contrastive-divergence-style EBM update (Sec. 3.2 of Yoon et al. (2024)) with our objective in Eq. (5), treating the true data distribution as the target and diffusion samples as noise.

(ii) For SiD$^2$A (Zhou et al., 2024a), we substitute the discriminator loss in Eq. (10) in Zhou et al. (2024a) with our objective in Eq. (5), again using true data as the target and diffusion samples as noise.

All experiments retain the original network architectures and hyperparameters unless noted:

(i) DxMI: direct ratio regularization weight is $1.5$ on CIFAR-10 and $1.55$ on ImageNet64×64; on ImageNet64×64, the value function learning rate is `3e-5`.

(ii) SiD$^2$A: direct ratio regularization weight is $0.5$ for CIFAR-10 (conditional and unconditional) and $0.4$ for ImageNet64×64; we set $\alpha = 1.2$ throughout.

(iii) All remaining hyperparameters (e.g., sampler learning rates, batch sizes) follow the defaults, making training iterations directly comparable as a measure of computational cost, particularly for SiD$^2$A.

### C.4 OFFLINE BLACK-BOX OPTIMIZATION

**Architecture** We provide detailed network architecture for the log-ratio estimator $\tilde{f}_{\boldsymbol{\alpha}_k}(\mathbf{z}, k)$ in Tab. 11; we adopt the same architecture as in Appx. C.2 throughout the experiments. The generator networks for $\mathbf{x}$ and $y$ is a 5-layer MLP structure shown in Tab. 12. We employ a single network that directly outputs the $(\mathbf{x}, y)$ pair to implement $g_{\boldsymbol{\beta},\mathbf{x}}$, $g_{\boldsymbol{\beta},y}$. For discrete tasks, the $g_{\boldsymbol{\beta},\mathbf{x}}$ head outputs logits for reconstruction. The encoder network to embed the input is a 3-layer MLP, as shown in Tab. 12. For discrete tasks, we add an additional embedding layer to map discrete inputs to continuous vectors. The network architectures are shared across different task domains; we only adjust the input size for inputs from different tasks. For function values $y$ and continuous $\mathbf{x}$, we normalize these values for training using the offline dataset maxium $\mathbf{x}_{\max}, y_{\max}$ and minimum $\mathbf{x}_{\min}, y_{\min}$: $\mathbf{x}_{\text{train}} = \frac{\mathbf{x} - \mathbf{x}_{\min}}{\mathbf{x}_{\max} - \mathbf{x}_{\min}}$. $y_{\text{train}} = \frac{y - y_{\min}}{y_{\max} - y_{\min}}$.

**Hyperparameters and implementation details** We use the linear schedule as in Ho et al. (2020) and Yu et al. (2022) to specify $\sigma_k^2$ and construct the intermediate distributions $\{q_k\}_{k=0}^K$. We set the number of stages to 6. Specifically, the sequence of $\{\sigma_k^2\}_{k=0}^5$ is: $\{\sigma_k^2\}_{k=0}^5 = [0.01, 0.3237, 0.5165, 0.6322, 0.7132, 0.7734, 0.9997]$. We re-weight the objective for each $\boldsymbol{\alpha}_k$ with $w_k = \sqrt{\sigma_K / \prod_{i=k}^K \sigma_i}$ to further emphasize stages closer to the final target distribution $q_{K+1}$, as in Yu et al. (2022). To estimate the Eq. (5), we can use Monte-Carlo average: $\mathcal{L}_M(\boldsymbol{\alpha}_k) \approx \frac{1}{n_1} \sum_{i=1}^{n_1} \left[ \log \frac{r_{\boldsymbol{\alpha}_k}(\mathbf{z}_i)}{M + r_{\boldsymbol{\alpha}_k}(\mathbf{z}_i)} \right] + \frac{M}{n_2} \sum_{j=1}^{n_2} \left[ \log \frac{M}{M + r_{\boldsymbol{\alpha}_k}(\mathbf{z}_j)} \right]$, where $\mathbf{z}_i \sim q_{k+1}, \mathbf{z}_j \sim q_k$. We follow Mnih & Teh (2012) and set $n_2 = M n_1$ in practice, where $M = 100$ for all experiments; $n_1$ is the batch size. When training with LEBM, we set the weight for reconstructing $\mathbf{x}$ and predicting $y$ to 50 for the term $\mathbb{E}_{q_{\boldsymbol{\phi}}}[\log p_{\boldsymbol{\beta}}(\mathbf{x}, y | \mathbf{z})]$. We set the weight for KL term between $q_{\boldsymbol{\phi}}$ and $p_{\boldsymbol{\alpha}}$ to 0.75. We apply orthogonal regularization (Brock et al., 2016) to the parameters of decoder networks, with a weight of 0.001 to facilitate training. These hyperparameters are shared across all the experiments.

The parameters of all the networks are initialized with the default pytorch methods (Paszke et al., 2019). We use the Adam optimizer (Kingma & Ba, 2014) with $\beta_1 = 0.5$ and $\beta_2 = 0.999$ to train the generator and encoder networks; the log-ratio estimator network is trained with default $(\beta_1, \beta_2)$. The initial learning rates of the generator and encoder networks are `1e-4`, and `5e-5` for the log-ratio network. We further perform gradient clipping by setting the maximal gradient norm as 100 for these networks. We run the experiments on a A6000 GPU with the batch size of 128. Training typically converges within 30K iterations on all the datasets.

When sampling with SVGD for optimization, the number of SVGD iterations $T$ is set to 500. For all experiments, we use RBF kernel $k(\mathbf{z}, \mathbf{z}_0) = \exp\left(-\frac{1}{2h^2}\|\mathbf{z} - \mathbf{z}_0\|_2^2\right)$ for SVGD, and set the bandwidth to be $h^2 = \frac{\text{med}^2}{2\log(n+1)}$. med is the median of the pairwise distance between the current

Table 11: **Network architecture for the log-ratio estimator** $\tilde{f}_{\boldsymbol{\alpha}_k}(\mathbf{z}, k)$**.** $N$ is set to 3 for image modeling and anomaly detection experiments, and 16 for the offline BBO experiments. ReLU denotes the Leaky ReLU activation function. The slope in Leaky ReLU is set to 0.2. For SVHN and CelebA datasets, we use nz=100. For the CIFAR-10 and CelebA-HQ datasets, we use nz=128 and nz=512, respectively. We use nz=8 for anomaly detection on the MNIST dataset; nz=20 is the latent space dimension used for offline BBO experiments.

| Layers | Output size | Note |
|---|---|---|
| **Indx. Emb.** | | |
| Input: $k$ | 1 | stage indx. of N$^2$CE |
| Sin. emb. | 128 | |
| Linear, LReLU | 128 | neg_slope 0.2 |
| Linear | 128 | |
| **Input Emb.** | | |
| Input: $\mathbf{z}$ | $nz = 20$ | |
| Linear, LReLU | 128 | neg_slope 0.2 |
| Linear | 128 | |
| **$\tilde{\mathbf{f}}_{\boldsymbol{\alpha}_k}(\mathbf{z}, \mathbf{k})$** | | |
| Input: $\mathbf{z}, t$ | $1, nz = 20$ | |
| Input & Indx. Emb. | $128 \times 2$ | Emb. of each input |
| Concat. | 256 | |
| LReLU, Linear | 128 | neg_slope 0.2 |
| N ResBlocks | 128 | LReLU, Linear + Input Skip |
| LReLU, Linear | 1 | energy score |

Table 12: **Generator and encoder network architectures.** LReLU indicates the Leaky-ReLU activation function. $d_{\mathbf{x}}$ is the dimension of input $\mathbf{x}$ mentioned in Tab. 18. For discrete tasks, $d_{\mathbf{x}}$ specifies the length of input, and $L$ denotes number of possible discrete tokens.

| Generator | | |
|---|---|---|
| **Layers** | **Out Size** | **Note** |
| Input: $\mathbf{z}$ | $nz = 20$ | |
| Linear, LReLU | 128 | |
| Linear, LReLU | 64 | neg_slope 0.2 |
| Linear, LReLU | 32 | |
| Linear, LReLU | 32 | |
| **Continuous Output** | | |
| Linear | $d_{\mathbf{x}} + 1$ | output $(\mathbf{x}, y)$ |
| **Discrete Output** | | |
| Linear | $d_{\mathbf{x}} \times L + 1$ | num. tokens. $L$ |

| Encoder | | |
|---|---|---|
| **Layers** | **Out Size** | **Note** |
| **Embedding** | | |
| (for discrete $\mathbf{x}$ only) | | |
| Input: $\mathbf{x}$ | $32 \times d_{\mathbf{x}}$ | emb. $\mathbf{x}$ |
| concat $(\mathbf{x}, y)$ | $(32\times)d_{\mathbf{x}} + 1$ | input $(\mathbf{x}, y)$ |
| Linear, LReLU | 128 | neg_slope 0.2 |
| Linear, LReLU | 128 | |
| **Mean head** | | |
| Linear | 20 | |
| **Var. head** | | |
| Linear | 20 | |

samples $\{\mathbf{z}_i^t\}_{i=1}^Q$; this follows the same heuristic as in Liu & Wang (2016) that $\sum_j k(\mathbf{z}_i, \mathbf{z}_j) \approx n \exp(-\frac{1}{2h^2}\text{med}^2) = 1$. We can balance the contribution from its own gradient and the influence from the other points for each sample $\mathbf{z}_i$. Of note, the bandwidth changes adaptively across iterations with this heuristic. For stability, we use Adam (Kingma & Ba, 2014) with default hyperparemters instead of AdaGrad (Duchi et al., 2011) used in Liu & Wang (2016) to allow for adaptive step size $\epsilon_t$. The initial step size is within the range of $[0.3, 0.5]$ for different tasks.

# D ADDITIONAL EXPERIMENT DETAILS & SUPPLEMENTARY RESULTS

## D.1 GAUSSIAN SIMULATION

We validate our analysis in Proposition 3.3 and Theorem 3.2 on a Gaussian location family. Specifically, we consider $p_{\boldsymbol{\alpha}}(\mathbf{x}) = \mathcal{N}(\mathbf{x}; \boldsymbol{\alpha}, I_d)$, where $\boldsymbol{\alpha} \in \mathbb{R}^d$ denotes the mean parameter and $I_d$ is the identity covariance. The target distribution is set to $q_* = \mathcal{N}(\boldsymbol{\alpha}^*, I_d)$, and the noise distribution to $q_0 = \mathcal{N}(0, I_d)$. Unless otherwise noted, we use $d = 2$ for visualization.

The log-ratio and its gradient admit closed forms:

$$f_{\boldsymbol{\alpha}}(\mathbf{x}) = \boldsymbol{\alpha}^\top \mathbf{x} - \tfrac{1}{2}\|\boldsymbol{\alpha}\|^2, \qquad \nabla_{\boldsymbol{\alpha}} f_{\boldsymbol{\alpha}}(\mathbf{x}) = \mathbf{x} - \boldsymbol{\alpha}.$$

Hence, the noisier NCE gradient is

$$\nabla_{\boldsymbol{\alpha}} \mathcal{L}_M(\boldsymbol{\alpha}) = \mathbb{E}_{q_*}\left[\frac{M}{M + r_{\boldsymbol{\alpha}}(\mathbf{x})}\ (\mathbf{x} - \boldsymbol{\alpha})\right] - \mathbb{E}_{p_{\boldsymbol{\alpha}}}\left[\frac{M}{M + r_{\boldsymbol{\alpha}}(\mathbf{x})}\ (\mathbf{x} - \boldsymbol{\alpha})\right],$$

where $r_{\boldsymbol{\alpha}}(\mathbf{x}) = \frac{p_{\boldsymbol{\alpha}}(\mathbf{x})}{q_0(\mathbf{x})}$. As $M \to \infty$, this reduces to the exact MLE gradient

$$\nabla_{\boldsymbol{\alpha}} \mathcal{L}_{\mathrm{MLE}}(\boldsymbol{\alpha}) = \mathbb{E}_{q_*}[\mathbf{x}] - \mathbb{E}_{p_{\boldsymbol{\alpha}}}[\mathbf{x}] = \boldsymbol{\alpha}^* - \boldsymbol{\alpha}.$$

**Setup.** We set $\boldsymbol{\alpha}^* = (1.5, -0.8)$ and initialize at $\boldsymbol{\alpha}_0 = (-2.0, 1.0)$. Gradients are approximated by Monte Carlo with $n = 4000$ samples per iteration. Optimization is performed with step size $\eta = 0.2$ for $T = 150$ iterations. We track several metrics during training, including the distance to the optimum $\|\boldsymbol{\alpha}_t - \boldsymbol{\alpha}^*\|_2$ and mean and median of absolute gradient errors. We report results across different noise magnitudes $M \in \{1, 2, 10, 50, 100, 1000\}$, with $M = 1$ corresponding to vanilla NCE and large $M$ approximating MLE.

**Further comparison with Nguyen-Wainwright-Jordan (NWJ) and simple reweighting.** We also compare our method with the NWJ objective (Nguyen et al., 2010):

$$\mathcal{L}_{\mathrm{NWJ}}(\boldsymbol{\alpha}) = \mathbb{E}_{q_*}[\log r_{\boldsymbol{\alpha}}] - \mathbb{E}_{q_0}[r_{\boldsymbol{\alpha}}] \implies \nabla_{\boldsymbol{\alpha}} \mathcal{L}_{\mathrm{NWJ}}(\boldsymbol{\alpha}) = \mathbb{E}_{q_*}[\mathbf{x} - \boldsymbol{\alpha}] - \mathbb{E}_{q_0}[r_{\boldsymbol{\alpha}}(\mathbf{x})\,(\mathbf{x} - \boldsymbol{\alpha})],$$

as well as a variant of our N²CE objective, where we only use N²CE-style weighting for the negative part of the objective and use the positive part of the vanilla N²CE objective:

$$\mathcal{L}_{\mathrm{neg-reweight}}(\boldsymbol{\alpha}) = \mathbb{E}_{q_*(\mathbf{x})}\left[\log \frac{r_{\boldsymbol{\alpha}}(\mathbf{x})}{1 + r_{\boldsymbol{\alpha}}(\mathbf{x})}\right] + M\mathbb{E}_{q_0(\mathbf{x})}\left[\log \frac{M}{M + r_{\boldsymbol{\alpha}}(\mathbf{x})}\right].$$

Similar to our previous setting, we consider in this case 5-d gaussians. We set $\boldsymbol{\alpha}^* = ([-1.5, -0.75, 0., 0.75, 1.5])$ and initialize at $\boldsymbol{\alpha}_0 = -\boldsymbol{\alpha}^*$, *i.e.*, on the opposite side. We use different number of Monte Carlo samples to approximate different regimes and for better visual comparison: (i) Gradients are approximated by Monte Carlo with $n = 500$ samples per iteration, corresponding with low gradient estimation variance in Proposition 3.3, and (ii) with $n = 2$ samples per iteration, corresponding with high estimation variance. Optimization is performed with step size $\eta = 0.2$ for $T = 150$ iterations. We plot again the distance to the optimum $\|\boldsymbol{\alpha}_t - \boldsymbol{\alpha}^*\|_2$ to illustrate the training trajectories. We plot a single-run visualization of the optimization trajectory shown in Fig. 5. Further, for more systematic and scientific comparisons, we use the trajectory-wise average squared norm $\frac{1}{T}\sum_{t=0}^{T-1}\|\theta_t - \theta^*\|_2^2$ as a summary of one run. We repeat 100 independent experiments and report the means and stds in Tabs. 13 and 14. We can see that (i) with $n = 2/500$, the optimal-$M$ N²CEs are indeed much better than those of NWJs, demonstrated by both lower mean values and lower stds. (ii) When $M$ is sufficiently large, N²CE empirically approaches the NWJ objective with $T = \log r$.

Table 13: **MSE sweep for $n = 2$.** Optimal $M$ predicted by Prop. 3.3: $C\sqrt{2} \approx 1.414C$.

| $M$ | NWJ ($T = \log r$) | 1 | 1.5 | 2 | 5 |
|---|---|---|---|---|---|
| avg. runs MSE | $54.548 \pm 237.528$ | $0.904 \pm 0.084$ | $\mathbf{0.884 \pm 0.083}$ | $0.888 \pm 0.084$ | $0.983 \pm 0.107$ |

| $M$ | 10 | 100 | 1000 | $10^9$ |
|---|---|---|---|---|
| avg. runs MSE | $1.139 \pm 0.168$ | $3.158 \pm 2.683$ | $17.564 \pm 42.751$ | $61.291 \pm 277.336$ |

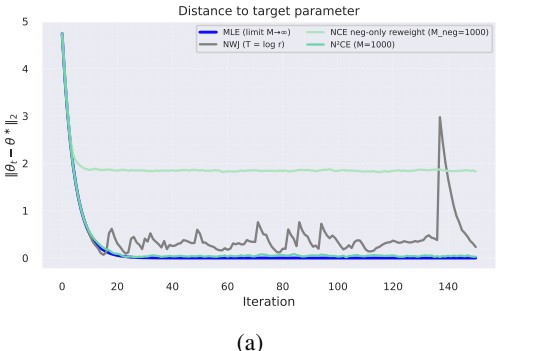
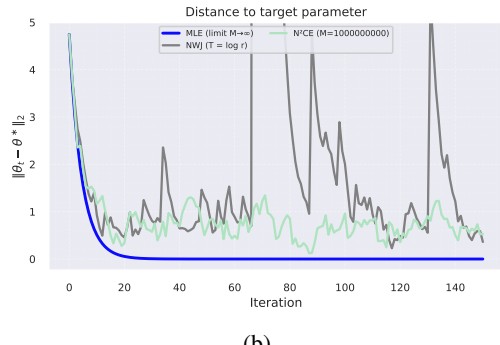

| (a) | (b) |

Figure 5: **$N^2$CE gradients can approach MLE gradients with appropriate $M$s, while NWJ and simple reweighting cannot.** We plot trajectories represented by $L_2$ norms between model and target parameters.

Table 14: **MSE sweep for** $n = 500$. Optimal $M$ predicted by Prop. 3.3: $C\sqrt{500} \approx 22.36C$.

| $M$ | NWJ ($T = \log r$) | 1 | 10 | 50 | 100 |
|---|---|---|---|---|---|
| avg. runs MSE | $1.359 \pm 5.454$ | $0.678 \pm 0.004$ | $0.489 \pm 0.004$ | $0.456 \pm 0.006$ | $\mathbf{0.453 \pm 0.007}$ |
| $M$ | 1000 | $10^4$ | $2 \times 10^4$ | $10^9$ | |
| avg. runs MSE | $0.489 \pm 0.020$ | $0.641 \pm 0.324$ | $0.750 \pm 0.889$ | $1.909 \pm 11.201$ | |

## D.2 IMAGE MODELING

For image modeling in Sec. 4.1, we include the following datasets to study our method: SVHN ($32 \times 32 \times 3$), CIFAR-10 ($32 \times 32 \times 3$), CelebA ($64 \times 64 \times 3$) and CeleAMask-HQ (256 x 256 x 3). Following Pang et al. (2020a), we use the full training set of SVHN (73,257) and CIFAR-10 (50,000), and take 40,000 samples of CelebA as the training data. We take 29,500 samples from the CelebAMask-HQ dataset as the training data. The images are scaled to $[-1, 1]$ and are randomly horizontally flipped with a prob. of .5 for training.

Table 15: **AUPRC($\uparrow$) scores for unsupervised anomaly detection on MNIST**. Baseline numbers are taken from Yoon et al. (2023); Yu et al. (2024). Results of our model are averaged over 10 trials.

| Heldout Digit | 1 | 4 | 5 | 7 | 9 |
|---|---|---|---|---|---|
| AE | $0.062 \pm 0.00$ | $0.204 \pm 0.00$ | $0.259 \pm 0.01$ | $0.125 \pm 0.00$ | $0.113 \pm 0.00$ |
| VAE | $0.063$ | $0.337$ | $0.325$ | $0.148$ | $0.104$ |
| ABP | $0.095 \pm 0.03$ | $0.138 \pm 0.04$ | $0.147 \pm 0.03$ | $0.138 \pm 0.02$ | $0.102 \pm 0.03$ |
| IGEBM | $0.101 \pm 0.02$ | $0.106 \pm 0.02$ | $0.205 \pm 0.11$ | $0.100 \pm 0.04$ | $0.079 \pm 0.02$ |
| MEG | $0.281 \pm 0.04$ | $0.401 \pm 0.06$ | $0.402 \pm 0.06$ | $0.290 \pm 0.04$ | $0.342 \pm 0.03$ |
| BiGAN-$\sigma$ | $0.287 \pm 0.02$ | $0.443 \pm 0.03$ | $0.514 \pm 0.03$ | $0.347 \pm 0.02$ | $0.307 \pm 0.03$ |
| ABP-LEBM | $0.336 \pm 0.01$ | $0.630 \pm 0.02$ | $0.619 \pm 0.01$ | $0.463 \pm 0.01$ | $0.413 \pm 0.01$ |
| JVAEBM | $0.297 \pm 0.03$ | $0.723 \pm 0.04$ | $0.676 \pm 0.04$ | $0.490 \pm 0.04$ | $0.383 \pm 0.03$ |
| Adaptive CE | $0.531 \pm 0.02$ | $0.729 \pm 0.02$ | $0.742 \pm 0.01$ | $0.620 \pm 0.02$ | $0.499 \pm 0.01$ |
| NAE | $0.802 \pm 0.08$ | $0.648 \pm 0.05$ | $0.716 \pm 0.03$ | $0.789 \pm 0.04$ | $0.441 \pm 0.07$ |
| MPDR-S | $0.764 \pm 0.05$ | $0.823 \pm 0.02$ | $0.741 \pm 0.04$ | $\mathbf{0.857} \pm 0.02$ | $0.478 \pm 0.05$ |
| MPDR-R | $0.844 \pm 0.03$ | $0.711 \pm 0.03$ | $0.757 \pm 0.02$ | $\underline{0.850} \pm 0.01$ | $0.569 \pm 0.04$ |
| DAMC | $0.684 \pm 0.02$ | $0.911 \pm 0.01$ | $0.939 \pm 0.02$ | $0.801 \pm 0.01$ | $\underline{0.705} \pm 0.01$ |
| DAMC-NCE | $0.702 \pm 0.01$ | $0.829 \pm 0.02$ | $0.764 \pm 0.01$ | $0.605 \pm 0.01$ | $0.502 \pm 0.02$ |
| DAMC-N²CE | | | | | |
| M=100,K=1 | $\underline{0.910} \pm 0.02$ | $\underline{0.911} \pm 0.01$ | $\underline{0.935} \pm 0.01$ | $0.779 \pm 0.01$ | $0.699 \pm 0.02$ |
| M=100,K=3 | $\mathbf{0.959} \pm 0.01$ | $\mathbf{0.935} \pm 0.01$ | $\mathbf{0.959} \pm 0.02$ | $0.845 \pm 0.01$ | $\mathbf{0.854} \pm 0.01$ |

## D.3 ANOMALY DETECTION

**Experiment settings** For anomaly detection in Sec. 4.1, we consider the MNIST (28 x 28 x 1) dataset; we follow the experimental settings in Zenati et al. (2018); Yu et al. (2024) and use 80% of the

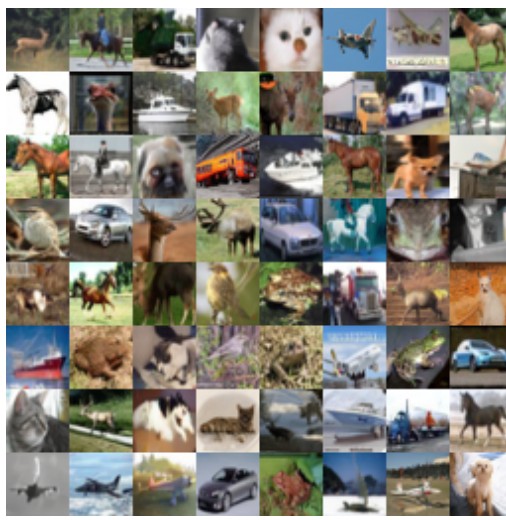 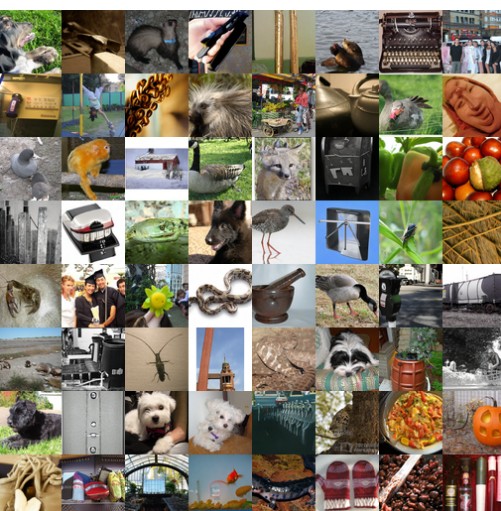

(a)                                                                 (b)

Figure 6: **Uncurated 1-step generation results from our distilled EDM models.** We present 1-step generation results from our distilled models on CIFAR-10 (left) and ImageNet64x64 (right) datasets.

in-domain data to train the model. We base our method upon the DAMC (Yu et al., 2024) model for posterior inference, where we use the LEBM learned by N$^2$CE as a plug-in replacement of the original prior model. With properly learned models, the posterior $p_{\boldsymbol{\theta},\boldsymbol{\phi}}(\mathbf{z}|\mathbf{x})$ could form a discriminative latent space that has separated probability densities for normal and anomalous data. Given the testing sample $\mathbf{x}$, we use log of unnormalized joint density $p_{\boldsymbol{\theta},\boldsymbol{\phi}}(\mathbf{z}|\mathbf{x}) \propto p_{\boldsymbol{\theta},\boldsymbol{\phi}}(\mathbf{x},\mathbf{z}) \approx p_{\boldsymbol{\beta}}(\mathbf{x}|\mathbf{z})p_{\boldsymbol{\alpha}}(\mathbf{z})|_{\mathbf{z}\sim q_{\boldsymbol{\phi}}(\mathbf{z}|\mathbf{x})}$ as our decision function; we draw samples from $q_{\boldsymbol{\phi}}(\mathbf{z}|\mathbf{x})$ and compare the corresponding reconstruction errors and energy scores. A higher value of log joint density indicates a higher probability of the test sample being a normal sample.

**Baselines**   We benchmark our method against three groups of strong counterparts: i) data-space energy-based models, including IGEBM (Du & Mordatch, 2019), MEG (Kumar et al., 2019), JVAEBM (Han et al., 2020) and MPDR-S/R (Yoon et al., 2023); ii) latent variable models, including AE, VAE (Kingma & Welling, 2013), ABP (Han et al., 2017), ABP-LEBM (Pang et al., 2020a), Adaptive CE (Xiao & Han, 2022), NAE (Yoon et al., 2021) and DAMC (Yu et al., 2024), and iii) GAN-Based method like BiGAN-$\sigma$ (Zenati et al., 2018).

## D.4   REWARD AND CRITIC LEARNING FOR DIFFUSION DISTILLATION

Table 16: **Ablative FID scores of $M$ on CIFAR-10 under Yoon et al. (2024) setting.**

| $M=1$ | $M=50$ | $M=100$ | $M=200$ |
|:-----:|:------:|:-------:|:-------:|
| 3.93  | 3.05   | **2.99** | 3.04   |

Table 17: **Ablative uncond. FID scores of $M$ on CIFAR-10 under Zhou et al. (2024a) setting.**

| $M=1$ | $M=10$ | $M=50$ | $M=100$ |
|:-----:|:------:|:------:|:-------:|
| 1.53  | 1.51   | **1.45** | 1.50   |

We further provide ablation of $M$ on CIFAR-10 in Tabs. 16 and 17. We can see that greater $M$, *e.g.*, $M=50$ or $M=100$ is significantly better than smaller $M$, especially $M=1$. However, larger $M$ may also incur larger approximation variance based on our analysis in Proposition 3.3. We therefore choose $M=100$ and $M=50$ respectively after the sweep for these experiments. These results further confirm our assumption about the noise magnitude.

### D.5 OFFLINE BLACK-BOX OPTIMIZATION

#### D.5.1 PROBLEM STATEMENT

Let $h : \mathcal{X} \to \mathbb{R}$ be a scalar function; the domain $\mathcal{X}$ is an arbitrary subset of $\mathbb{R}^d$. In Black-Box Optimization (BBO), $h$ is an unknown black-box function; we are interested in finding $\mathbf{x}^*$ that maximizes $h$:

$$\mathbf{x}^* \in \arg\max_{\mathbf{x} \in \mathcal{X}} h(\mathbf{x}). \tag{23}$$

Typically, $h$ is expensive to evaluate. In offline BBO, we assume no direct access to $h$ during training, and are thus **not allowed** to actively query the black-box function during optimization, unlike in online BBO where most approaches would employ iterative online solving schemes (Kong et al., 2023b;a; Shahriari et al., 2015; Snoek et al., 2012). Specifically, in the *offline* BBO setting (Trabucco et al., 2022), one has only access to a pre-collected dataset of observations, $\mathcal{D} = \{(\mathbf{x}_i, y_i)\}_{i=1}^n$, where $h(\mathbf{x}_i) = y_i$. During evaluation, one may query the function $h$ for a small budget of $Q$ queries to output candidates with the best function value obtained.

#### D.5.2 LEARNING LATENT VARIABLE MODELS FOR OFFLINE BLACK-BOX OPTIMIZATION

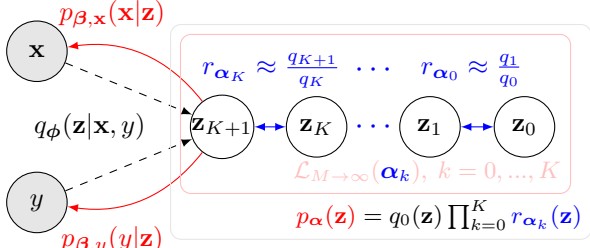

Figure 7: **Graphical illustration of our BBO framework.** We construct an energy-based latent space model $p_{\boldsymbol{\alpha}}$ for offline BBO via learning a series of ratio estimators $\{r_{\boldsymbol{\alpha}_k}\}_{k=0}^K$ with the N²CE objective $\mathcal{L}_{M \to \infty}$ to optimize the ELBO *without* MCMC. After training, we employ stochastic samplers like LD or SVGD to perform BBO by sampling from the implicit inverse model $p_{\boldsymbol{\theta}}(\mathbf{x}|y) \propto \mathbb{E}_{p_{\boldsymbol{\theta}}(\mathbf{z}|y)}[p_{\boldsymbol{\beta},\mathbf{x}}(\mathbf{x}|\mathbf{z})]$, where $p_{\boldsymbol{\theta}}(\mathbf{z}|y) \propto p_{\boldsymbol{\beta},y}(y|\mathbf{z})p_{\boldsymbol{\alpha}}(\mathbf{z})$ given $y$. Best viewed in color.

We consider again variational learning of the latent variable model. Specifically, the ELBO in this setting would be

$$\begin{aligned}\text{ELBO}_{\boldsymbol{\theta},\boldsymbol{\phi}} &= \log p_{\boldsymbol{\theta}}(\mathbf{x}, y) - D_{\text{KL}}(q_{\boldsymbol{\phi}}(\mathbf{z}|\mathbf{x}, y)\|p_{\boldsymbol{\theta}}(\mathbf{z}|\mathbf{x}, y)) \\ &= \mathbb{E}_{q_{\boldsymbol{\phi}}(\mathbf{z}|\mathbf{x},y)}[\log p_{\boldsymbol{\beta}}(\mathbf{x}, y|\mathbf{z})] - D_{\text{KL}}(q_{\boldsymbol{\phi}}(\mathbf{z}|\mathbf{x}, y)\|p_{\boldsymbol{\alpha}}(\mathbf{z})),\end{aligned} \tag{24}$$

where in $p_{\boldsymbol{\theta}}(\mathbf{x}, y)$, $p_{\boldsymbol{\beta},\mathbf{x}}(\mathbf{x}|\mathbf{z}) = \mathcal{N}(g_{\boldsymbol{\beta},\mathbf{x}}(\mathbf{z}), \sigma_x^2 \mathbf{I})$ for continuous $\mathbf{x}$ and multinomial for discrete $\mathbf{x}$, and $p_{\boldsymbol{\beta},y}(y|\mathbf{z}) = \mathcal{N}(g_{\boldsymbol{\beta},y}(\mathbf{z}), \sigma_y^2 \mathbf{I})$. $g_{\boldsymbol{\beta},\mathbf{x}}$ and $g_{\boldsymbol{\beta},y}$ is the generator network for $\mathbf{x}$ and prediction network for $y$, respectively. $\sigma_x^2, \sigma_y^2$ take pre-specified values. $q_{\boldsymbol{\phi}}(\mathbf{z}|\mathbf{x}, y) = \mathcal{N}(\boldsymbol{\mu}_{\boldsymbol{\phi}}, \boldsymbol{\sigma}_{\boldsymbol{\phi}}\mathbf{I})$ is the amortized posterior. See Fig. 7 for the graphical illustration.

**Optimization as gradient-based sampling** Once the latent variable model with $p_{\boldsymbol{\alpha}}$ is properly trained with N²CE, we can sample from the parameterized inverse model $p_{\boldsymbol{\theta}}(\mathbf{x}|y) \propto \mathbb{E}_{p_{\boldsymbol{\theta}}(\mathbf{z}|y)}[p_{\boldsymbol{\beta},\mathbf{x}}(\mathbf{x}|\mathbf{z})]$ to solve the BBO problem. Specifically, we **i)** sample $\mathbf{z}$ from $p_{\boldsymbol{\theta}}(\mathbf{z}|y) \propto p_{\boldsymbol{\beta},y}(y|\mathbf{z})p_{\boldsymbol{\alpha}}(\mathbf{z})$ with LD or SVGD, where $y$ is set to the offline dataset maximum $y_{\max}$ as in (Krishnamoorthy et al., 2023); and **ii)** feed them into the generator network $g_{\boldsymbol{\beta},\mathbf{x}}$ to obtain the proposed input designs $\mathbf{x}$. Intuitively, $p_{\boldsymbol{\theta}}(\mathbf{z}|y = y_{\max})$ exploits the offline high value input design modes, while sampling from $p_{\boldsymbol{\theta}}$ and mapping from $z$ to $x$ with $p_{\boldsymbol{\beta},\mathbf{x}}(\mathbf{x}|\mathbf{z})$ achieves expanded latent exploration guided by the latent space model $p_{\boldsymbol{\alpha}}$. We provide an interpretation of latent exploration in Appx. D.5.3.

**Stein Variational Gradient Descent (SVGD)** A well-established counterpart of MCMC for approximating the target distribution is SVGD, first proposed in the seminal work of Liu & Wang (2016). Specifically, it tackles the sampling problem with a set of particles for approximation, on which a form of (functional) gradient descent is performed to minimize the Kullback-Leibler Divergence (KLD) and drive the particles to fit the true target distribution. Given an unnormalized log-density function

$l(\mathbf{z})$, and an initial set of samples $\{\mathbf{z}_i^0\}_{i=1}^n$, we can iteratively update these samples as follows to approximate the distribution specified by $l$:

$$\mathbf{z}_i^{t+1} = \mathbf{z}_i^t + \epsilon_t \hat{\phi}^*(\mathbf{z}_i^t), \; t = 0, 1, ..., T - 1 \text{ where}$$

$$\hat{\phi}^*(\mathbf{z}) = \frac{1}{n} \sum_{j=1}^n \left[ k(\mathbf{z}_j^t, \mathbf{z}) \nabla_{\mathbf{z}_j^t} l(\mathbf{z}_j^t) + \nabla_{\mathbf{z}_j^t} k(\mathbf{z}_j^t, \mathbf{z}) \right], \tag{25}$$

where $k(\cdot, \cdot)$ is a positive definite kernel (*e.g.*, RBF kernel) and $\epsilon_t$ the step size at the $t$-th iteration. We formulate the optimization process as sampling from an unnormalized log-density and utilize the mode-exploration ability of SVGD to propose high-quality candidates for the BBO problem.

### D.5.3 INTERPRETATION OF OPTIMIZATION AS GRADIENT-BASED SAMPLING

In order to extrapolate as far as possible, while still staying on the actual manifold of high-value observations, we need to measure the validity of the generated samples $\mathbf{x}$ as in Trabucco et al. (2021). We identify the value of $y$ from $p_{\boldsymbol{\beta},y}(y|\mathbf{z})$ (forward model) serves as one good indicator as in Kumar & Levine (2020). The generated samples $\mathbf{x}$, associated with function values similar to existing $y$ values in the offline dataset, are likely in-distribution; those where the $p_{\boldsymbol{\beta},y}(y|\mathbf{z})$ predicts a very different score (often erroneously large, see Fig. 4 when $y > 1$) can be too far outside the training distribution (Kumar & Levine, 2020; Mashkaria et al., 2023; Krishnamoorthy et al., 2023). In addition, ideally, after training we have a well-learned and informative prior model $p_{\boldsymbol{\alpha}}$ that i) captures the multiple possible modes of the one-to-many inverse mapping $p_{\boldsymbol{\theta}}(\mathbf{x}|y)$, and ii) faithfully assigns the function values to observations through the joint latent space (Fig. 3 and 4). Taking together, we can optimize over $\mathbf{z}$ in the latent space to find the best, most trustworthy $\mathbf{x}$ subject to the following constraints: i) $\mathbf{z}$ has a high likelihood under the prior and ii) the predicted function value associated with $\mathbf{z}$ is not too different from the value $y_{\max}$. This can be formulated as the following optimization problem:

$$\underset{y(\mathbf{z})}{\arg\max} \, y = g_{\boldsymbol{\beta}}(\mathbf{z}) \; s.t. \; p_{\boldsymbol{\beta}}(y_{\max}|\mathbf{z}) > \epsilon_1, \; p_{\boldsymbol{\alpha}}(\mathbf{z}) > \epsilon_2. \tag{26}$$

The above constrained optimization problem is conceptually similar to the formulation mentioned in Kumar & Levine (2020) Sec. 3.2. The $p_{\boldsymbol{\beta}}(y_{\max}|\mathbf{z}) > \epsilon_1$ constraint enforces that $\mathbf{z}$ stays on the manifold of high-value observations as we discussed; the $p_{\boldsymbol{\alpha}}(\mathbf{z}) > \epsilon_2$ constraint encourages extrapolating beyond the best score in the offline dataset $\mathcal{D}$ by providing meaningful gradient during the optimization.

In our set-up, the unnormalized log-density of $p_{\boldsymbol{\beta},y}(y|\mathbf{z})p_{\boldsymbol{\alpha}}(\mathbf{z})$ given the best offline function value $y_{\max}$ can be written as $-\lambda_1 \|y_{\max} - g_{\boldsymbol{\beta},y}(\mathbf{z})\|_2^2 + \lambda_2 \left( \sum_{k=0}^K \tilde{f}_{\boldsymbol{\alpha}_k}(\mathbf{z}) - \frac{1}{2}\|\mathbf{z}\|_2^2 \right)$, where $\lambda_1$ is the variance of $p_{\boldsymbol{\beta},y}(y|\mathbf{z}) = \mathcal{N}(g_{\boldsymbol{\beta},y}(\mathbf{z}), \sigma_y^2 \mathbf{I})$ and $\lambda_2$ re-weights the prior term. This is equivalent to optimizing the Lagrangian of Eq. (26), with the density constraints modified to be log-densities. In practice, we employ LD or SVGD to fully utilize the informative latent space during optimization. Ideally, the dual variables, $\lambda_1$ and $\lambda_2$ are supposed to be optimized together with $y$ and $\mathbf{z}$, however, we find it convenient to choose them to be fixed as a constant throughout, at $\lambda_1 = 20.0$ and $\lambda_2 = 1.0$, analogous to penalty methods in constrained optimization. The same values of $\lambda_1$ and $\lambda_2$ are used for all domains in our experiments.

### D.6 2D BRANIN FUNCTION

Branin is a well-known function for benchmarking optimization methods. We consider the negative of the standard 2D Branin function in the range $x_1 \in [-5, 10], x_2 \in [0, 15]$:

$$f_{br}(x_1, x_2) = -a(x_2 - bx_1^2 + cx_1 - r)^2 - s(1 - t) \cos x_1 - s, \tag{27}$$

where $a = 1$, $b = \frac{5.1}{4\pi^2}$, $c = \frac{5}{\pi}$, $r = 6$, $s = 10$, and $t = \frac{1}{8\pi}$. In this square region, $f_{br}$ has three global maximas, $(-\pi, 12.275)$, $(\pi, 2.275)$, and $(9.425, 2.475)$; with the maximum value of $-0.398$ (Fig. 2). The Gradient Ascent (GA) baseline uses the offline dataset to train a forward prediction model parameterized by a 2-layer MLP mapping $\mathbf{x}$ to $y$ and then performs gradient ascent on $\mathbf{x}$ to infer its optima. DDOM and BONET follow their default implementation. For the typical forward method, GA in the data space, the problem of assigning an input $\mathbf{x}$ with erroneously large value is clear even in this 2D toy example; the proposed points often violates the square domain constraints. This can be partly seen in the largest std. in Tab. 6.

### D.7 DESIGN-BENCH

### D.7.1 TASK OVERVIEW

The goal of **TF-Bind-8** and **TF-Bind-10** is to optimize for a DNA sequence that has a maximum affinity to bind with a particular transcription factor. The sequences are of length 8 for **TF-Bind-8** and 10 for **TF-Bind-10**, where each element in the sequence is one of 4 bases. **ChEMBL** optimizes drugs for specific chemical properties. In **D'Kitty** and **Ant Morphology**, one need to optimize for the morphology of robots. In **Superconductor**, the goal is to find a material with a high critical temperature. These tasks contain neither personally identifiable nor offensive information. We present detailed information on the tasks we evaluate on in Design-Bench (Trabucco et al., 2022) as below in Tab. 18. **SIZE** denotes the training dataset size, and **NUM. TOKS** denotes the number of tokens for discrete tasks.

Table 18: **Design-Bench dataset statistics.**

| TASK | SIZE | INPUT DIM. | NUM. TOKS | TASK MAX |
|---|---|---|---|---|
| TF-Bind-8 | 32898 | 8 | 4 | 1.0 |
| TF-Bind-10 | 10000 | 10 | 4 | 2.128 |
| ChEMBL | 441 | 31 | 185 | 443000.0 |
| D'Kitty | 10004 | 56 | - | 340.0 |
| Ant | 10004 | 60 | - | 590.0 |
| Superconductor | 17014 | 86 | - | 185.0 |

### D.7.2 HOPPERCONTROLLER & NAS

As in Krishnamoorthy et al. (2023) and Mashkaria et al. (2023), we exclude the HopperController task in our results due to significant inconsistencies between the offline dataset values and the values obtained when running the oracle (see Fig. 8 and 9). This is a known bug in Design-bench (see details in the link). Therefore, we decided not to include Hopper in our evaluation and analysis. Following Krishnamoorthy et al. (2023), we exclude NAS due to excessive computation cost required beyond our budget for evaluating across multiple seeds.

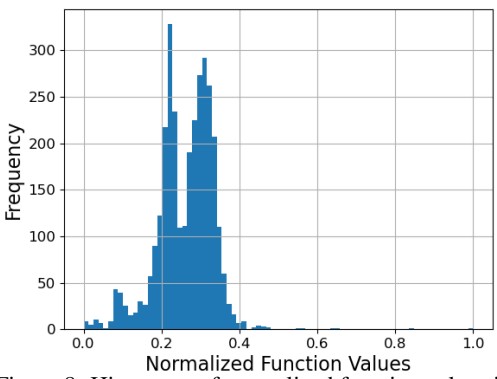
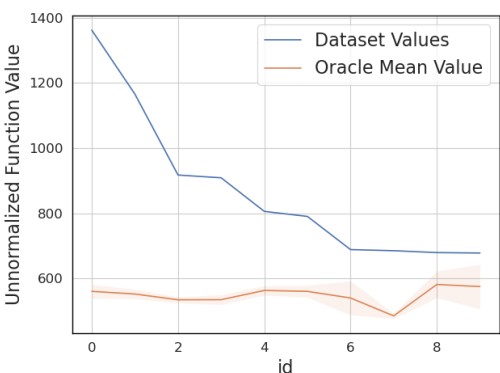

Figure 8: Histogram of normalized function values in the Hopper dataset. The distribution is highly skewed towards low function values (plot from Mashkaria et al. (2023)).

Figure 9: Dataset values vs Oracle values for top 10 points. We can see from mean and standard deviation over 20 runs that oracle is noisy (plot from Mashkaria et al. (2023)).

### D.7.3 BASELINES

As mentioned in Sec. 4.3.2, we compare our method with three groups of baselines: **i)** sampling via generative inverse models with different parameterizations, including CbAS (Brookes et al., 2019), Auto.CbAS (Fannjiang & Listgarten, 2020), MIN (Kumar & Levine, 2020) and DDOM to validate our model design; **ii)** gradient(-like) updating from existing designs, *e.g.*, GA-based (Fu

& Levine, 2020; Trabucco et al., 2021; Chen et al., 2022; Qi et al., 2022; Yuan et al., 2024; Chen et al., 2024) and BONET and **iii)** baselines mentioned in (Trabucco et al., 2022). The generative inverse models learn $p_{\boldsymbol{\theta}}(\mathbf{x}|y)$ with different parameterizations and then perform conditional sampling of the high-value designs for BBO. CbAS (Brookes et al., 2019) directly models $p_{\boldsymbol{\theta}}(\mathbf{x}|y)$, while introducing the joint distribution of $\mathbf{x}$ and $\mathbf{z}$ to facilitate importance sampling. It trains a VAE model to parameterize the distribution using identified high-quality inputs $\mathbf{x}$ (by thresholding); it gradually adapts the distribution to the high-value part by refining the threshold. MIN (Kumar & Levine, 2020) learns $p_{\boldsymbol{\theta}}(\mathbf{x}|y) \propto \mathbb{E}_{p(\mathbf{z}|y)}[p(\mathbf{x}|\mathbf{z}, y)]$ with a conditional GAN-like model to instantiate $p(\mathbf{x}|\mathbf{z}, y)$ (Goodfellow et al., 2020; Mirza & Osindero, 2014). The method optimizes over $\mathbf{z}$ given the offline dataset maximum $y_{\max}$ and map $\mathbf{z}$ back to $\mathbf{x}$ for BBO solutions. The optimization process over $\mathbf{z}$ draw approximate samples from $p(\mathbf{z}|y)$. DDOM (Krishnamoorthy et al., 2023) consider directly parameterizing the inverse mapping $p_{\boldsymbol{\theta}}(\mathbf{x}|y)$ with a conditional diffusion model in the input design space utilizing the expressiveness of DDPMs (Ho et al., 2020). The second group includes the gradient(-like) updating methods, and has the flavor of forward methods in general. The baseline methods include: (i) Gradient Ascent: operates directly on the learned surrogates to propose new input designs via simple gradient ascent; (ii) COMs (Trabucco et al., 2021): regularizes the NN-parameterized surrogate by assigning lower function values to designs obtained during the gradient ascent process; (iii) ROMA (Yu et al., 2021b): incorporates prior knowledge about function smoothness into the surrogate and optimizes the design against the proxy; (iv) NEMO (Fu & Levine, 2020): leverages normalized maximum likelihood to constrain the distance between the surrogate and the ground-truth; (v) BDI (Chen et al., 2022): proposes to distill the information from the static dataset into the high-scoring design; (vi) IOM (Qi et al., 2022): enforces the invariance between the representations of the static dataset and generated designs to achieve a natural trade-off; (vii) ICT (Yuan et al., 2024) explores using a pseudo-labeler to generate valuable data for fine-tuning the surrogate. BONET (Mashkaria et al., 2023) trains a transformer-based model to mimic the optimization trajectories of black-box optimizers, and rolls out evaluation trajectories from the trained model for optimization.

In addition to these recent works, we also compare with several baseline methods described in Trabucco et al. (2022): 1) GP-qEI (Wilson et al., 2017): instantiates BO using a Gaussian Process as the uncertainty quantifier and the quasi-Expected Improvement (q-EI) acquisition function, 2) CMA-ES (Hansen, 2006): an evolutionary algorithm that maintains a belief distribution over the optimal candidates; it gradually refines the distribution by adapting the covariance matrix towards optimal candidates using feedback from the learned surrogate, and 3) REINFORCE (Williams, 1992): first learns a proxy function and then optimizes the input space distribution by leveraging the proxy and the policy-gradient estimator. Since we are in the offline setting, for active methods like BO, we follow the procedure of Trabucco et al. (2022) and optimize on a surrogate model trained on the offline dataset.

For $Q = 128$ results in Tab. 23, we additionally include ExPT (Nguyen et al., 2024) and BOOTGEN (Kim et al., 2024) for rough reference. Of note, these two methods have slightly different set-ups, and are therefore not directly comparable to other baselines and our method. ExPT (Nguyen et al., 2024) focuses extensively on few-shot learning scenarios with models pre-trained on larger datasets. BOOTGEN (Kim et al., 2024) focuses specifcially on optimizing biological sequences.

### D.7.4 ADDITIONAL RESULTS & ANALYSIS

**Proof-of-concept results for data efficiency in BBO** First, we uniformly sample $N = 5000$ and $N = 50$ points from the branin function domain for training. We observe that our method i) performs consistently better than BONET (Mashkaria et al., 2023) and gradient ascent, and ii) demonstrates less performance drop compared with BONET when dataset size is significantly reduced. Further, we use the D'Kitty dataset and withhold an $x\%$ size subsection of data, *i.e.*, $0\%$ represents the full dataset. We have similar observation on these datasets summarized in Tabs. 19 and 20, where our method shows much less performance drop compared with the transformer-based method BONET.

### D.7.5 ADDITIONAL RESULTS FOR $Q = 256$

**Unnormalized results** For reference, we present the unnormalized results of Tab. 7 in Tab. 21.

Table 19: **Results on (reduced size) Branin function dataset.**

| N | $\mathcal{D}$ (best) | Grad. Ascent | BONET | Ours |
|---|---|---|---|---|
| 5000 | $-6.199$ | $-3.95 \pm 4.26$ | $-1.79 \pm 0.84$ | $\mathbf{-0.41 \pm 0.13}$ |
| 50 | $-6.231$ | $-4.64 \pm 3.17$ | $-2.13 \pm 0.15$ | $\mathbf{-0.43 \pm 0.12}$ |

Table 20: **Results on (reduced size) D'Kitty dataset.** $x\%$ indicates the proportion of training data withheld, *i.e.*, $0\%$ represents the full dataset.

| | 0% | 90% | 99% |
|---|---|---|---|
| BONET | $285.11 \pm 15.13$ | $274.11 \pm 7.57$ | $\mathbf{241.17 \pm 18.07}$ |
| Ours | $293.33 \pm 7.45$ | $289.67 \pm 7.86$ | $\mathbf{283.65 \pm 9.23}$ |

Table 21: **Results on design-bench. Unnormalized** results with a budget $Q = 256$.

| BASELINE | TFBIND8 | TFBIND10 | CHEMBL | SUPERCON. | ANT | D'KITTY |
|---|---|---|---|---|---|---|
| $\mathcal{D}$ (best) | 0.439 | 0.00532 | 383700.000 | 74.0 | 165.326 | 199.363 |
| GP-qEI | $0.824 \pm 0.086$ | $0.675 \pm 0.043$ | $387950.000 \pm 0.000$ | $92.686 \pm 3.944$ | $480.049 \pm 0.000$ | $213.816 \pm 0.000$ |
| CMA-ES | $0.933 \pm 0.035$ | $0.848 \pm 0.136$ | $388400.000 \pm 400.000$ | $90.821 \pm 0.661$ | $\mathbf{1016.409 \pm 906.407}$ | $4.700 \pm 2.230$ |
| REINFORCE | $0.959 \pm 0.013$ | $0.692 \pm 0.113$ | $388400.000 \pm 2100.000$ | $89.027 \pm 3.093$ | $-131.907 \pm 41.003$ | $-301.866 \pm 246.284$ |
| Gradient Ascent | $\underline{0.981 \pm 0.015}$ | $0.770 \pm 0.154$ | $390050.000 \pm 2000.000$ | $93.252 \pm 0.886$ | $-54.955 \pm 33.482$ | $226.491 \pm 21.120$ |
| COMs | $0.964 \pm 0.020$ | $0.750 \pm 0.078$ | $390200.000 \pm 500.000$ | $78.178 \pm 6.179$ | $540.603 \pm 20.205$ | $277.888 \pm 7.799$ |
| BONET | $0.975 \pm 0.004$ | $0.855 \pm 0.139$ | $\underline{391000.000 \pm 1900.000}$ | $80.845 \pm 4.087$ | $567.042 \pm 11.653$ | $\underline{285.110 \pm 15.130}$ |
| CbAS | $0.958 \pm 0.018$ | $0.761 \pm 0.067$ | $389000.000 \pm 500.000$ | $83.178 \pm 15.372$ | $468.711 \pm 14.593$ | $213.917 \pm 19.863$ |
| MINs | $0.938 \pm 0.047$ | $0.770 \pm 0.177$ | $390950.000 \pm 200.000$ | $89.469 \pm 3.227$ | $533.636 \pm 17.938$ | $272.675 \pm 11.069$ |
| DDOM | $0.971 \pm 0.005$ | $\underline{0.885 \pm 0.367}$ | $387950.000 \pm 1050.000$ | $\underline{103.600 \pm 8.139}$ | $548.227 \pm 11.725$ | $250.529 \pm 10.992$ |
| Ours | $\mathbf{0.990 \pm 0.003}$ | $\mathbf{1.344 \pm 0.340}$ | $\mathbf{392149.225 \pm 3821.853959}$ | $\mathbf{104.848 \pm 3.113}$ | $\underline{572.490 \pm 7.277}$ | $\mathbf{293.327 \pm 7.452}$ |

**50-th percentile results** We present the normalized results with $Q = 256$ at 50-th percentile to further demonstrate the effectiveness of our approach in Tab. 22.

Table 22: **Normalized results on design-bench.** 50-th percentile results with $Q = 256$. Baseline numbers from Mashkaria et al. (2023). DDOM (Krishnamoorthy et al., 2023) does not report 50-th percentile results; we are unable to find public model weights corresponding with the results in Tab. 7 for DDOM and therefore omit DDOM in this table.

| BASELINE | TFBIND8 | TFBIND10 | CHEMBL | SUPERCON. | ANT | D'KITTY | MEAN SCORE$^\uparrow$ | MNR$^\downarrow$ |
|---|---|---|---|---|---|---|---|---|
| GP-qEI | $0.443 \pm 0.004$ | $0.494 \pm 0.002$ | $0.299 \pm 0.002$ | $0.272 \pm 0.006$ | $0.754 \pm 0.004$ | $0.633 \pm 0.000$ | $0.491 \pm 0.016$ | 4.8 |
| CMA-ES | $0.543 \pm 0.007$ | $0.483 \pm 0.011$ | $0.376 \pm 0.004$ | $-0.051 \pm 0.004$ | $0.685 \pm 0.018$ | $0.633 \pm 0.000$ | $0.466 \pm 0.020$ | 5.2 |
| REINFORCE | $0.450 \pm 0.003$ | $0.472 \pm 0.000$ | $0.470 \pm 0.017$ | $0.146 \pm 0.009$ | $0.307 \pm 0.002$ | $0.633 \pm 0.000$ | $0.083 \pm 0.004$ | 5.7 |
| Gradient Ascent | $0.572 \pm 0.024$ | $0.470 \pm 0.004$ | $0.463 \pm 0.022$ | $0.141 \pm 0.010$ | $0.637 \pm 0.148$ | $0.633 \pm 0.000$ | $0.478 \pm 0.029$ | 5.1 |
| COMs | $0.492 \pm 0.009$ | $0.472 \pm 0.012$ | $0.365 \pm 0.026$ | $0.525 \pm 0.018$ | $0.885 \pm 0.002$ | $0.633 \pm 0.000$ | $0.522 \pm 0.034$ | 4.6 |
| BONET | $0.505 \pm 0.055$ | $0.496 \pm 0.037$ | $0.369 \pm 0.015$ | $\mathbf{0.819 \pm 0.032}$ | $\mathbf{0.907 \pm 0.020}$ | $0.630 \pm 0.000$ | $0.614 \pm 0.035$ | 3.4 |
| CbAS | $0.422 \pm 0.007$ | $0.458 \pm 0.001$ | $0.111 \pm 0.009$ | $0.384 \pm 0.010$ | $0.752 \pm 0.003$ | $0.633 \pm 0.000$ | $0.436 \pm 0.008$ | 6.5 |
| MINs | $0.425 \pm 0.011$ | $0.471 \pm 0.004$ | $0.330 \pm 0.011$ | $0.651 \pm 0.010$ | $0.890 \pm 0.003$ | $0.633 \pm 0.000$ | $0.547 \pm 0.005$ | 4.8 |
| Ours | $\mathbf{0.896 \pm 0.033}$ | $\mathbf{0.507 \pm 0.012}$ | $0.373 \pm 0.002$ | $0.715 \pm 0.004$ | $\mathbf{0.907 \pm 0.008}$ | $0.630 \pm 0.000$ | $\mathbf{0.672 \pm 0.009}$ | 1.2 |

### D.7.6 Additional results for $Q = 128$

**25-th, 50-th, 75-th and 100-th percentile results** We present the normalized results with $Q = 128$ at 100-th and 50-th percentile to further demonstrate the effectiveness of our approach in Tabs. 23 and 25. Additionally, we provide results at 25-th and 75-th percentiles for reference in Tab. 24.

### D.7.7 Extended discussion for ablation study

As mentioned in Sec. 4.3, we provide ablative results on noise magnitude $M$, number of intermediate stages $K$ and query budget $Q$ in Tabs. 8, 26 and 27. Of note, the $K = 1, M = 100$ row in Tab. 8 means learning a LEBM w/ Eq. (5) alone, w/o ratio decomposition. We can see that the variant delivers significantly better results than MLE-LEBM w/ SVGD sampler. This indicates the effectiveness of the noise-intensified objective. $K = 6, M = 1$ row means learning with the original NCE objective but with ratio decomposition, w/o intensifying the noise distributions. We can see that simply combining the NCE objective with variational learning of LEBM only achieves inferior performance compared with learning LEBM with the N$^2$CE objective.

We further provide detailed ablation of $M$ in Tab. 26. We can see that greater $M$, *e.g.*, $M = 100$ or $M = 1000$ is significantly better than smaller $M$, *e.g.*, $M = 1$ or $M = 10$. However, larger $M$ incurs larger memory consumption based on our implementation in Appx. C. It also has a pitfall of leading to larger approximation variance based on our analysis in Proposition 3.3. We therefore choose $M = 100$ for all experiments. These results further confirm our assumption about the noise magnitude. We can see from Tab. 26 that larger number of intermediate stages $K$ typically brings better performance. These results demonstrate the need for ratio decomposition, aligned with our

Table 23: **Normalized design-bench results with $Q = 128$ at 100-th percentile.** Baseline numbers from Kim et al. (2024); Yuan et al. (2024); Nguyen et al. (2024); Chen et al. (2024). **ChEMBL** dataset is excluded following baseline set-ups. * indicates that the baseline has slightly different set-ups; the numbers are for rough reference (see Appx. D.7.3 for more details).

| BASELINE | TFBIND8 | TFBIND10 | SUPERCON. | ANT | D'KITTY | MEAN SCORE↑ | MNR↓ |
|---|---|---|---|---|---|---|---|
| $\mathcal{D}$ (best) | 0.439 | 0.467 | 0.399 | 0.565 | 0.884 | - | - |
| GP-qEI | $0.798 \pm 0.083$ | $0.652 \pm 0.038$ | $0.402 \pm 0.034$ | $0.819 \pm 0.000$ | $0.896 \pm 0.000$ | 0.713 | 14.2 |
| CMA-ES | $0.953 \pm 0.022$ | $0.670 \pm 0.023$ | $0.465 \pm 0.024$ | $\mathbf{1.214 \pm 0.732}$ | $0.724 \pm 0.001$ | 0.805 | 8.6 |
| REINFORCE | $0.948 \pm 0.028$ | $0.663 \pm 0.034$ | $0.481 \pm 0.013$ | $0.266 \pm 0.032$ | $0.562 \pm 0.196$ | 0.584 | 12.4 |
| *BOOTGEN | $\underline{0.979 \pm 0.001}$ | - | - | - | - | - | - |
| *ExPT | $0.933 \pm 0.036$ | $0.677 \pm 0.048$ | - | $\underline{0.970 \pm 0.004}$ | $\mathbf{0.973 \pm 0.005}$ | - | - |
| Grad. Ascent | $0.886 \pm 0.035$ | $0.647 \pm 0.021$ | $0.495 \pm 0.011$ | $0.934 \pm 0.011$ | $0.944 \pm 0.017$ | 0.781 | 10.2 |
| COMS | $0.496 \pm 0.065$ | $0.622 \pm 0.003$ | $0.491 \pm 0.028$ | $0.856 \pm 0.040$ | $0.938 \pm 0.015$ | 0.681 | 13.4 |
| BONET | $0.911 \pm 0.005$ | $\underline{0.756 \pm 0.006}$ | $0.500 \pm 0.002$ | $0.927 \pm 0.002$ | $0.954 \pm 0.000$ | 0.810 | 6.8 |
| ROMA | $0.924 \pm 0.040$ | $0.666 \pm 0.035$ | $0.510 \pm 0.015$ | $0.917 \pm 0.030$ | $0.927 \pm 0.013$ | 0.789 | 8.2 |
| NEMO | $0.943 \pm 0.005$ | $0.711 \pm 0.021$ | $0.502 \pm 0.002$ | $0.958 \pm 0.011$ | $0.954 \pm 0.007$ | 0.814 | 5.2 |
| BDI | $0.870 \pm 0.000$ | $0.605 \pm 0.000$ | $0.513 \pm 0.000$ | $0.906 \pm 0.000$ | $0.919 \pm 0.000$ | 0.763 | 11.6 |
| IOM | $0.878 \pm 0.069$ | $0.648 \pm 0.023$ | $\underline{0.520 \pm 0.018}$ | $0.918 \pm 0.031$ | $0.945 \pm 0.012$ | 0.782 | 8.6 |
| ICT | $0.958 \pm 0.008$ | $0.691 \pm 0.023$ | $0.503 \pm 0.017$ | $0.961 \pm 0.007$ | $\underline{0.968 \pm 0.020}$ | 0.816 | 3.4 |
| Tri-mtring | $\underline{0.970 \pm 0.001}$ | $0.722 \pm 0.017$ | $0.514 \pm 0.018$ | $0.948 \pm 0.014$ | $0.966 \pm 0.010$ | $\underline{0.824}$ | $\underline{3}$ |
| CbAS | $0.927 \pm 0.051$ | $0.651 \pm 0.060$ | $0.503 \pm 0.069$ | $0.876 \pm 0.031$ | $0.892 \pm 0.008$ | 0.770 | 10.8 |
| Auto.CbAS | $0.910 \pm 0.044$ | $0.630 \pm 0.045$ | $0.421 \pm 0.045$ | $0.882 \pm 0.045$ | $0.906 \pm 0.006$ | 0.745 | 13 |
| MINs | $0.905 \pm 0.052$ | $0.616 \pm 0.021$ | $0.499 \pm 0.017$ | $0.445 \pm 0.080$ | $0.892 \pm 0.011$ | 0.671 | 13.6 |
| DDOM | $0.957 \pm 0.006$ | $0.657 \pm 0.006$ | $0.495 \pm 0.012$ | $0.940 \pm 0.004$ | $0.935 \pm 0.001$ | 0.797 | 8 |
| Ours | $\mathbf{0.983 \pm 0.011}$ | $\mathbf{0.771 \pm 0.096}$ | $\mathbf{0.559 \pm 0.024}$ | $0.954 \pm 0.008$ | $0.954 \pm 0.004$ | $\mathbf{0.844}$ | $\mathbf{2}$ |

Table 24: **Normalized design-bench 25-th and 75-th results with $Q = 128$.**

| Percentile | TFBIND8 | TFBIND10 | SUPERCON. | ANT | D'KITTY |
|---|---|---|---|---|---|
| 25-th | $0.635 \pm 0.020$ | $0.453 \pm 0.007$ | $0.351 \pm 0.019$ | $0.527 \pm 0.033$ | $0.878 \pm 0.002$ |
| 75-th | $0.873 \pm 0.006$ | $0.945 \pm 0.012$ | $0.409 \pm 0.013$ | $0.832 \pm 0.012$ | $0.910 \pm 0.002$ |

Table 25: **Normalized design-bench 50-th percentile results with $Q = 128$ at 100-th percentile.** Baseline numbers from Kim et al. (2024); Yuan et al. (2024); Nguyen et al. (2024); Chen et al. (2024). **ChEMBL** dataset is excluded following baseline set-ups. * indicates that the baseline has slightly different set-ups; the numbers are for rough reference (see Appx. D.7.3 for more details).

| BASELINE | TFBIND8 | TFBIND10 | SUPERCON. | ANT | D'KITTY | MEAN SCORE↑ | MNR↓ |
|---|---|---|---|---|---|---|---|
| $\mathcal{D}$ (best) | 0.439 | 0.467 | 0.399 | 0.565 | 0.884 | - | - |
| GP-qEI | $0.439 \pm 0.000$ | $0.467 \pm 0.000$ | $0.300 \pm 0.015$ | $0.567 \pm 0.000$ | $0.883 \pm 0.000$ | 0.531 | 11.2 |
| CMA-ES | $0.537 \pm 0.014$ | $0.484 \pm 0.014$ | $0.379 \pm 0.003$ | $-0.045 \pm 0.004$ | $0.684 \pm 0.016$ | 0.408 | 10.2 |
| REINFORCE | $0.462 \pm 0.021$ | $0.475 \pm 0.008$ | $0.463 \pm 0.016$ | $0.138 \pm 0.032$ | $0.356 \pm 0.131$ | 0.379 | 10.8 |
| *BOOTGEN | $0.833 \pm 0.007$ | - | - | - | - | - | - |
| *ExPT | $0.473 \pm 0.014$ | $0.477 \pm 0.014$ | - | $0.705 \pm 0.018$ | $0.902 \pm 0.006$ | - | - |
| Grad. Ascent | $0.532 \pm 0.017$ | $0.529 \pm 0.027$ | $0.339 \pm 0.015$ | $0.564 \pm 0.014$ | $0.877 \pm 0.005$ | 0.568 | 7.4 |
| COMS | $0.439 \pm 0.000$ | $0.466 \pm 0.002$ | $0.316 \pm 0.022$ | $0.568 \pm 0.002$ | $0.883 \pm 0.002$ | 0.534 | 10.6 |
| BONET | $0.505 \pm 0.004$ | $0.465 \pm 0.002$ | $\mathbf{0.470 \pm 0.004}$ | $0.620 \pm 0.003$ | $0.897 \pm 0.000$ | 0.591 | 5.4 |
| ROMA | $0.555 \pm 0.020$ | $0.512 \pm 0.020$ | $0.372 \pm 0.019$ | $0.479 \pm 0.041$ | $0.853 \pm 0.007$ | 0.554 | 7.6 |
| NEMO | $0.548 \pm 0.017$ | $0.516 \pm 0.020$ | $0.322 \pm 0.008$ | $0.593 \pm 0.000$ | $0.885 \pm 0.000$ | 0.539 | 10 |
| BDI | $0.439 \pm 0.000$ | $0.476 \pm 0.000$ | $0.412 \pm 0.000$ | $0.474 \pm 0.000$ | $0.855 \pm 0.000$ | 0.546 | 9.4 |
| IOM | $0.439 \pm 0.000$ | $0.477 \pm 0.010$ | $0.352 \pm 0.021$ | $0.509 \pm 0.033$ | $0.876 \pm 0.006$ | 0.531 | 9.8 |
| ICT | $0.551 \pm 0.013$ | $\mathbf{0.541 \pm 0.004}$ | $0.399 \pm 0.012$ | $0.592 \pm 0.025$ | $0.874 \pm 0.005$ | 0.591 | 5.2 |
| Tri-mtring | $0.609 \pm 0.021$ | $0.527 \pm 0.008$ | $0.355 \pm 0.003$ | $0.606 \pm 0.007$ | $0.886 \pm 0.001$ | 0.597 | 4.2 |
| CbAS | $0.428 \pm 0.010$ | $0.463 \pm 0.007$ | $0.111 \pm 0.017$ | $0.384 \pm 0.016$ | $0.753 \pm 0.008$ | 0.428 | 15 |
| Auto.CbAS | $0.419 \pm 0.007$ | $0.461 \pm 0.007$ | $0.131 \pm 0.010$ | $0.364 \pm 0.014$ | $0.736 \pm 0.025$ | 0.422 | 15.8 |
| MINs | $0.421 \pm 0.015$ | $0.468 \pm 0.006$ | $0.336 \pm 0.016$ | $0.618 \pm 0.040$ | $0.887 \pm 0.004$ | 0.546 | 8.8 |
| DDOM | $0.553 \pm 0.002$ | $0.488 \pm 0.001$ | $0.295 \pm 0.001$ | $0.590 \pm 0.003$ | $0.870 \pm 0.001$ | 0.559 | 8.6 |
| Ours | $\mathbf{0.873 \pm 0.006}$ | $0.499 \pm 0.002$ | $0.370 \pm 0.004$ | $\mathbf{0.704 \pm 0.013}$ | $\mathbf{0.897 \pm 0.003}$ | $\mathbf{0.669}$ | $\mathbf{3}$ |

analysis in Sec. 3.3. In addition, since we are using the same network architecture for ablating different $K$s, the network capacity is fixed. Larger number of intermediate stages may require a larger network to perform well (circumventing competition between stages), as we are learning the ensemble of ratio estimators using the shared network. SVGD in the data space typically requires a relatively large population of initial samples to perform well. In our set-up, we set the query budget $Q$ to the population size and run SVGD in the latent space. We can see from Tab. 27 that our method is robust to $Q$ by pulling SVGD sampling back to the latent space.

**Random baselines** One valid concern is that our framework solves the offline BBO problem by simply memorizing the best points in the offline dataset and proposing new points close to those best points during evaluation utilizing the randomness in the sampling process. To see whether this is the case, we follow Mashkaria et al. (2023) to perform a simple experiment on the **D'Kitty** task.

Table 26: **Ablation of $K, M$ on SPRCON. (avg. over 5 runs).**

| $M = 100$ | $K = 2$ | $K = 4$ | $K = 6$ | $K = 8$ |
|---|---|---|---|---|
| | $0.427_{\pm 0.013}$ | $0.557_{\pm 0.029}$ | $\mathbf{0.567}_{\pm 0.017}$ | $0.527_{\pm 0.049}$ |
| $K = 6$ | $M = 1$ | $M = 10$ | $M = 100$ | $M = 1000$ |
| | $0.497_{\pm 0.044}$ | $0.486_{\pm 0.023}$ | $\mathbf{0.567}_{\pm 0.017}$ | $0.548_{\pm 0.046}$ |

Table 27: **Ablation of $Q$ on D'Kitty (avg. over 5 runs).**

| $Q = 2$ | $Q = 8$ | $Q = 32$ | $Q = 128$ | $Q = 256$ |
|---|---|---|---|---|
| $0.906_{\pm 0.009}$ | $0.925_{\pm 0.013}$ | $0.937_{\pm 0.006}$ | $0.954_{\pm 0.004}$ | $\mathbf{0.961}_{\pm 0.006}$ |

We include random baselines by similarly choosing a small hypercube domain around the optimal point in the offline dataset. These baselines then uniformly sample 256 random points in the cube as the proposed candidates; this can be seen as a offline dataset oracle w/ noise model. In Tab. 28, we show the results for different widths of this hypercube ranging from 0 to 0.1. 0 width means returning the best point in the offline dataset. We can see that the best result (226.10) from these noisy oracle models is significantly lower than the result from our model (304.36). We can also see from the trend in max, mean and std. values that simply extending the width of cube for random search is hopeless for solving the offline BBO task. This is very likely due to the high-dimensionality and highly-multimodal nature of the black-box function input space. The comparison between these random baselines and our method suggests that our method does find an informative latent space for effectively extrapolating beyond the offline dataset best designs. The prior model in our formulation provides meaningful gradient for exploring the underlying high-function-value manifold.

Table 28: **Random baseline results on the D'Kitty dataset.** Results from 5 runs.

| CUBE WIDTH | 0.00 | 0.005 | 0.01 | 0.05 | 0.1 | OURS |
|---|---|---|---|---|---|---|
| MAX$^\uparrow$ | 199.23 | 212.66 | 222.44 | 226.10 | 209.00 | **304.36** |
| MEAN$^\uparrow$ | 199.23 | 190.68 | 182.13 | $-169.62$ | $-368.71$ | **206.76** |
| STD.$^\downarrow$ | 0.00 | 9.28 | 12.21 | 331.00 | 261.37 | **63.26** |

## USE OF LARGE LANGUAGE MODELS

During the preparation of this manuscript, we used Large Language Models (LLMs) to assist with language polishing, organization of sections, and improving clarity of exposition. All technical contributions, including theoretical results, experimental design, and analysis, were conceived and carried out by the authors. The LLMs were not used to generate novel research content, proofs, or experimental results.

