# OpenReview forum: "``Noisier'’ Noise Contrastive Estimation is (Almost) Maximum Likelihood"
_ICLR.cc/2026/Conference — ICLR 2026 Poster_

### Official Review · Reviewer_LHEq · 2025-10-27

**Soundness:** 3
**Presentation:** 3
**Contribution:** 2
**Rating:** 4
**Confidence:** 4

**Summary:**

This paper proposes “Noisier” Noise Contrastive Estimation (N2CE), a simple modification of the NCE objective that increases the noise magnitude by a factor $M$. The central claim is that as $M\to\infty$, the gradient of the N2CE objective approaches the maximum-likelihood (MLE) gradient. The paper proves a gradient-limit result and a convergence guarantee for exponential families, and analyzes the finite-M, finite-sample error. Empirically, N2CE improves over NCE/MLE-style baselines across a wide range of different tasks.

**Strengths:**

**1. Clear Convergence guarantees.** Proposition 3.1 gives a clean expression showing $\nabla L_M \to \nabla J_{\text{MLE}}$ as $M\to\infty$, and Theorem 3.2 yields polynomial iteration complexity when $M$ is large.

**2. Simple, drop-in objective.** N2CE just reweights the negative class via $M$; it’s easy to integrate into existing pipelines.

**3. Extensive experiments.** Experiments cover diverse datasets and baselines. It demonstrates versatility across tasks: improving FID scores in generative modeling, boosting AUPRC in anomaly detection, and enhancing diffusion distillation.

**Weaknesses:**

The method proposed in this work is straightforward: its core idea is to assign greater weight to the loss on negative samples.

First, I find the paper’s theoretical results quite confusing. The theory generally requires a fairly large $M$, and in Proposition 3.1, $M$ even tends to infinity. We know that in NCE, the model is expected to give high scores to positive samples and low scores to negative ones. If the loss function assigns an extremely large weight to the negative part, the model is likely to largely ignore the positive part and focus solely on the negatives. In that case, the model could simply assign very low scores to all outputs, which is clearly detrimental to learning. I am therefore not surprised that the experiments indicate $M$ should not be too large. However, the theoretical analysis in the paper suggests that a larger $M$ is preferable. This is not only puzzling but also inconsistent with the experimental results.

Second, the fundamental idea of the method is to reweight positives and negatives. How is this essentially different from simply multiplying the negative part by a weight $k$? I believe the authors need to discuss the distinction between the two. Do the theoretical results in the paper also hold for such a simple weighted baseline? Empirically, with a straightforward weighting scheme and careful tuning of $k$, can the results match those of state-of-the-art methods?

Moreover, if the choice of $M$ is also important, the paper does not provide a theoretical analysis of how $M$ should be selected. If hyperparameter search over $M$ is required in experiments, the computational cost will inevitably be high.

Finally, since the authors have already conducted image generation of people, why not include generated image results in the appendix, as is commonly done in the experiments of other generative methods?

**Questions:**

See the Weakness part.

---

> ### Author Response · Authors · 2025-11-20
> **Response to Reviewer LHEq (1/N)**
>
> We would like to sincerely thank reviewer LHEq for the time devoted into reviewing our submission. We appreciate that reviewer LHEq raised detailed and interesting questions, to which we will provide point-to-point responses below and hopefully help to clarify the issues.
>
> **The role of M**
>
> > First, I find the paper’s theoretical results quite confusing. (...) If the loss function assigns an extremely large weight to the negative part, the model is likely to largely ignore the positive part and focus solely on the negatives. In that case, the model could simply assign very low scores to all outputs, which is clearly detrimental to learning. (...)
>
> > Moreover, if the choice of $M$ is also important, the paper does not provide a theoretical analysis of how $M$ should be selected
>
> We appreciate that reviewer LHEq explicitly raised this question. This points to the core idea of our method: instead of simply studying the limiting case of the $L_M$ objective, we indeed proposes a *family* of distributions indexed by $M$, where $M=1$ leads to vanilla NCE, and $M \to \infty$ recovers MLE. In fact, if we have **infinite sample size**, i.e., $n \to \infty$ meaning that we can recover the **true** expectation with Monte Carlo estimation and no variance, then the larger $M$, the better indeed. This is because of our results in Prop 3.1 and especially in Theorem 3.2. In this case, the model **will not** ignore the positive part, intuitively speaking because we also have infinitely many positive samples for a single estimate.
>
> We actually have a more precise quantitative description of the relationship between these two ``infinitely large'' values $n$ and $M$, **as we stated in Prop. 3.3**. If we can increase the sample size $n$, then according to Prop 3.3, as long as $M \sim o(\sqrt{n})$,  meaning that $M$ increases but at a rate slower than $\sqrt{n}$, then we can still enjoy increasingly more accurate approximation of the MLE gradient because the variance is controlled. But in practice and particularly in our setting, we only have a (fixed) **finite sample size**, since we don't want to have computational overhead. This is when the variance of estimation starts to kick in: if we increase $M$ but with a fixed $n$. Indeed, this is the key motivation of our result in Prop. 3.3 as we want to enjoy the reduced bias by our results as much as possible, while not exaggerating too much about the variance by increasing $M$. In short, the root reason that increasing $M$ can lead to worse results is the variance caused by finite (and fixed!) sample size. And we would like to clarify that Prop 3.1 only addresses the mean value. We hope this would help with further understanding of our work.
>
> We also agree that choosing the optimal $M$ is an interesting research question, as we observed U-shaped patterns in different scenarios when sweeping through $M$ (tab. 14, 15 & 24). Indeed, again based on our Prop 3.3 results and as we discussed above, although finding an exact optimal $M$ could be case-by-case as it also depends heavily on the ratio estimator $r_\alpha$, **our result does suggest that with a given sample size (batch size) of $n$, $M$ in general may not exceed $C \sqrt{n}$ where $C$ is a constant**, C=10 for example. We can see in tab. 14, 15 & 24 that this value fits very well with our empirical observations. Due to the scope of this paper, we will have to leave studying an exact optimal $M$ for future works.

---

> > ### Author Response · Authors · 2025-11-20
> > **Response to Reviewer LHEq (2/N)**
> >
> > **What about re-weighting the negative part**
> >
> > > (...) How is this essentially different from simply multiplying the negative part by a weight $k$?
> >
> > We are not very clear about how reviewer LHEq would like to re-weight the negative part. Therefore, we take the liberty of proposing the following two variants: (i) reweighting the negative part of vanilla NCE directly:
> > $L_{\rm neg-reweight-nce}(\alpha) = E_{q_*} \left[\log \frac{r_{{\alpha}}(x)}{1 + r_{{\alpha}}(x)}\right] + ME_{q_{0}} \left[\log \frac{1}{1+ r_{{\alpha}}(x)}\right]$, and (ii) reweighting the negative part only with N2CE-style weighting:
> > $L_{\rm neg-reweight-n2ce}(\alpha) = E_{q_*} \left[\log \frac{r_{{\alpha}}(x)}{1 + r_{{\alpha}}(x)}\right] + ME_{q_{0}} \left[\log \frac{M}{M + r_{{\alpha}}(x)}\right]$.
> >
> > For (i) we did a pilot study on 5-d gaussian models as we did in Fig. 1 and D. 1. We set $M=1000$ and compare against the proposed variant with MLE and our N2CE objective. The (i) objective **fail to converge**. It actually diverges after the first tens of steps. For (ii), under the same settings it starts to converge but the estimated results are far off the optimal ones. **We provide a visualization of the optimization process with objective (ii) in the newly added Fig.5(a) as in Fig. 1 in the updated D.1 section**. We can see that without the N2CE-style weighting on the positive part, the bias can be excessively large. Detailed settings can be found in the `Further comparison with Nguyen-Wainwright-Jordan (NWJ) and simple reweighting.` paragraph in D.1 section.
> >
> > We further move on to conduct a light-weight experiment on LEBM on cifar-10 dataset, using exactly the same setting as we did in Sec. 4.1. **We observe that the LEBM learned with the (ii) variant can obtain a FID score of 93.62, much higher than that of the LEBM learned by N2CE (FID: 77.35)**.
> >
> > We believe these empirical evidences further confirm the effectiveness of our formulation.
> >
> > **Presentation**
> >
> > > (...) why not include generated image results in the appendix (...)
> >
> > We would like to thank the reviewer for bringing this matter to our attention. We have added a Fig. 6 on page 31 to present uncurated 1-step generation results of our distilled models on cifar-10 and imagenet64 datasets.
> >
> > We sincerely hope our further clarification could help to address the raised issues. We are open to any further questions regarding our work. We truly appreciate the time devoted by reviewer Qdxk on our submission and kindly ask for raising the rating of our submission if possible.

---

### Official Review · Reviewer_PhJz · 2025-10-27

**Soundness:** 3
**Presentation:** 3
**Contribution:** 1
**Rating:** 2
**Confidence:** 4

**Summary:**

The authors explore a simple modification to the Noise Contrastive Estimation (NCE) density ratio estimator: as proposed by Gutmann et al. (2012), a positive factor $M$ is used to reweigh the noise distribution, yielding N$^2$CE. It is claimed that this simple modification results in loss function gradients that get closer to the maximum-likelihood loss gradients as $M$ grows, providing practical improvements over classical maximum-likelihood and standard NCE (e.g., yielding better samplers).

**Strengths:**

The experimental setups are diverse and truly extensive, and the provided results are rich and comprehensive. The writing is also clear and easy to follow.

Additionally, the authors provide theoretical results, proving their main claim and supplementary convergence results.

**Weaknesses:**

**Connection with MLE**

The work is rather weak when it comes to the main claim: the connection between N$^2$CE and MLE, and the novelty behind it. Firstly, this connection is outright obvious from the loss function alone: note that
$$
\log M + \log \frac{r}{M + r} = \log \frac{r}{1 + r/M} \underset{M \to +\infty}{\longrightarrow} \log r
$$
$$
M \log \frac{M}{M + r} = -\log \left( 1 + \frac{r}{M} \right)^M =-\log \left( 1 + \frac{1}{M/r} \right)^{r \cdot M/r} \underset{M \to +\infty}{\longrightarrow} -\log e^r = -r
$$
Now, combining everything and denoting $T = \log r$, for high $M$ we get
$$
\mathcal{L}\_M \sim \mathbb{E}\_{q^*} T - \mathbb{E}\_{q_0} e^T + \text{const},
$$
which is the **well-known** Nguyen-Wainwright-Jordan (NWJ) [1] or KL-$f$-GAN [2] or "self-normalizing DRE" [3] variational lower bound on the Kullback-Leibler divergence. Maximizing this bound **is** the maximum-likelihood learning (recall the connection between the KL-divergence and cross-entropy).

[1] X. Nguyen et al, "Estimating divergence functionals and the likelihood ratio by convex risk minimization". Proc. of NeurIPS 2008.

[2] S. Nowozin et al, "f-GAN: Training Generative Neural Samplers using Variational Divergence Minimization". Proc. of NeurIPS 2016.

[3] B. Poole et al, "On Variational Bounds of Mutual Information". Proc. of ICML 2019

Next, the work lacks an important information-theoretic background. Currently, without it, the connection between N$^2$CE and MLE appears as some sort of miracle (*"As we show below, increasing M has a striking effect..."*, line 146). However, by showing that N$^2$CE is just a lower bound on some divergence which interpolates between the Jensen-Shannon and Kullback-Leibler divergences, one resolves this mystery.

From the theory of variational representations of $f$-divergences (Section 7.13 in [4]),
$$
\text{KL}(p \Vert q) = 1 + \sup\_T [\mathbb{E}\_p T - \mathbb{E}\_q e^T] = 1 + \sup\_r [\mathbb{E}\_p \log r - \mathbb{E}\_q r] \qquad \text{(MLE, NWJ)}
$$
$$
\text{JS}(p \Vert q)  \overset{\text{def}}{=} \frac{1}{2} \text{KL}(p \Vert q/2 + p/2) + \frac{1}{2} \text{KL}(q \Vert q/2 + p/2)  =  \log 2 + \sup\_h [\mathbb{E}\_p \log h - \mathbb{E}\_q \log (1 - h)] = \log 2 + \sup\_r \left[\mathbb{E}\_p \log \frac{r}{1+r} + \mathbb{E}\_q \log \frac{1}{1+r}\right] \qquad \text{(NCE)}
$$
Now, one can prove that
$$
D\_\alpha(p \Vert q) \overset{\text{def}}{=} (1 - \alpha) \cdot \text{KL}(p \Vert \alpha q + (1 - \alpha) p) + \alpha \cdot \text{KL}(q \Vert \alpha q + (1 - \alpha) p) = h(\alpha) + \sup\_r \left[\mathbb{E}\_p \log \frac{r}{M+r} + M \mathbb{E}\_q \log \frac{M}{M+r}\right] \qquad \text{(N$^2$CE)},
$$
where $\alpha = M / (1 + M)$, $h(x)$ is the binary entropy function, and the optimal $r^*(x) = p(x) / q(x)$.

That said, of course we get $\left. \text{(N$^2$CE)} \right|\_{M=1} = \text{NCE}$ and $\lim\_{M \to +\infty} \text{N$^2$CE} = \text{MLE}$.

[4] Polyanskiy, Y., and Wu, Y. "Information Theory: From Coding to Learning". Cambridge University Press, 2024.

**Is there really a need for N$^2$CE?**

Now that we have established that N$^2$CE is just NWJ for sufficiently large $M$, why can't we just use the latter? In the authors' text, I find no justification for using moderate $M$. Therefore, it is logical to jump straight to NWJ, thus avoiding numerical instabilities and recovering the *exact* MLE gradients.

**The "Density-chasm"**

While the authors address the _density-chasm_ problem from the perspective of the "magnitude" of the noise distribution, it is a well-known fact that NWJ (an $M \to \infty$ limit of N$^2$CE) still suffers from this exact problem (for instance, please, refer to Figure 2 in [3], that corresponds to the density-chasm case for high mutual information). Therefore, I do not find it convincing that N$^2$CE offers something new to resolve this issue.

Furthermore, while the authors cite the telescoping density ratio estimation technique (TRE by Rhodes et al. (2020)), they do not recognise more recent classifier-based DRE approaches: DRE-$\infty$ [5] and Classification Diffusion Models [6]. These methods proved to be better at dealing with the density-chasm, while also providing tractable density ratios (and decent generation results in [6]).

[5] K. Choi et al., "Density Ratio Estimation via Infinitesimal Classification". Proc. of AISTATS 2022.

[6] S. Yadin et al., "Classification Diffusion Models: Revitalizing Density Ratio Estimation". Proc. of NeurIPS 2024.

**Questions:**

1. In the text, you conclude that sufficiently large $M$ (e.g., $50$ or $100$) are necessary for achieving better results: _"This
again supports our analysis that a sufficiently large M is critical for accurate gradient approximation."_, line 411. However, with $M$ of this magnitude, N$^2$CE is essentially NWJ. Is there any reason to prefer N$^2$CE with moderate $M$ over NWJ?
2. Why did you focus on proving the convergence of the gradients, but not of the loss function itself?
3. Can you compare your method with plain NWJ, DRE-$\infty$ and Classification Diffusion Models (please, see the weaknesses for the references)?

---

> ### Author Response · Authors · 2025-11-21
> **Response to Reviewer PhJz (1/N)**
>
> We would like to thank reviewer PhJz for the time devoted into reviewing our submission. We appreciate that reviewer raised some intriguing questions, to which we will provide point-to-point responses below and hopefully help to address the issues.
>
> **The need of N$^2$CE and its connection with MLE/NWJ**
>
> > Firstly, this connection is outright obvious from the loss function alone (...)
>
> > (...) we have established that N$^2$CE is just NWJ for sufficiently large $M$. In the authors' text, I find no justification for using moderate $M$. Therefore, it is logical to jump straight to NWJ, thus avoiding numerical instabilities and recovering the exact MLE gradients.
>
> > In the text, you conclude that sufficiently large $M$ (e.g., 50 or 100) are necessary for achieving better results. However, with $M$ of this magnitude, N$^2$CE is essentially NWJ (...)
>
> We appreciate that reviewer PhJz raises these questions and bring NWJ to our attention. This points to the core idea of our method: instead of simply studying the limiting case of the $L_M$ objective, we indeed proposes a *family* of distributions indexed by $M$, where $M=1$ leads to vanilla NCE, and $M \to \infty$ recovers MLE, or equivalently in reviewer PhJz's word, NWJ. In fact, if we have **infinite sample size**, meaning that we can recover the true expectation with Monte Carlo estimation, then we can safely push the limit towards infinity and obtain MLE, NWJ or our method N$^2$CE. **But is that the case in practice?**
>
> As we stated in Prop. 3.3, and it's true in most if not all machine learning problems that we only have **finite sample size**, where the variance of estimation plays an important role. The reviewer stated that `In the authors' text, I find no justification for using moderate $M$`. But variance is the key motivation of our result in Prop. 3.3 as we want to enjoy the reduced bias by our results as much as possible, while not exaggerating too much about the variance by increasing $M$. **We explicitly mention this in Sec. 3.3, particularly in L207-214.** We would like to kindly remind that we also have sufficient empirical evidence for this bias-variance trade-off, as well as the existence of a finite, practically optimal $M$ in different scenarios, please check **tab. 14, 15 & 24 in which U-shaped patterns clearly emerge**.
>
> The reviewer also states that `it is logical to jump straight to NWJ, thus avoiding numerical instabilities and recovering the exact MLE gradients` and `with $M$ of this magnitude (50 - 100), N$^2$CE is essentially NWJ`. We respectfully disagree with both of these statements, with additional empirical evidence presented below.
>
> **With M of 50 - 100, N$^2$CE is NEVER NWJ. In fact, N$^2$CE is NOT NWJ even with $M = 1000000000$ in a very clean if not the simplest 5-d Gaussian case.** If anything, **NWJ is the one we observe obvious numerical instabilities but not N$^2$CE**. To be specific, similar to our settings in Fig. 1, we first consider the 5-d gaussian cases. We set the target as $\alpha^* = ([-1.5 , -0.75,  0.  ,  0.75,  1.5 ])$ and initialize the model at $\alpha = -\alpha^* $, i.e., on the opposite side. We use different number of Monte Carlo samples to approximate different regimes and for better visual comparison: (i) Gradients are approximated by Monte Carlo with $n=500$ samples per iteration, corresponding with low gradient estimation variance case in Prop 3.3, and (ii) with $n=2$ samples per iteration, corresponding with high estimation variance. Optimization is performed with step size $\eta=0.2$ for $T=150$ iterations. We plot again the distance to the optimum $\|\alpha_t-\alpha^*\|_2$ to illustrate the training trajectories. Results are shown in Fig. 5(a) and (b) correspondingly. In (a), we set $M=1000$, and observe decent approximation of the MLE trajectory with N$^2$CE. On the contrary, we observe severe fluncutations of the NWJ trajectory in this case, which indicates excessively large estimation variance even with large sample size. We tried to replicate the NWJ pattern with N$^2$CE by greatly increasing $M$ (actually to $1000000000$ indeed) in high variance setting in (b). However, although we did observe increased flunctuations of N$^2$CE curves, it's still better than NWJ. **We kindly refer to the updated D.1 section and Fig. 5(a)(b) in appendix for more details.**
>
> We further move on to conduct a light-weight experiment on LEBM on cifar-10 dataset, using exactly the same setting as we did in Sec. 4.1. We observe that the **LEBM learned with the NWJ objective can obtain a FID score of 90.97, much higher than that from the LEBM learned by N2CE** w/o TRE (FID: 77.35 in tab. 1).
>
> We believe these empirical evidences together serve as strong evidence of why we need N$^2$CE, instead of directly jumping to NWJ.

---

> ### Author Response · Authors · 2025-11-21
> **Response to Reviewer PhJz (2/N)**
>
> **Comparison with DRE-$\infty$ and CDM**
>
> > Furthermore, while the authors cite the telescoping density ratio estimation technique, they do not recognise more recent classifier-based DRE approaches: DRE-$\infty$ [5] and Classification Diffusion Models [6]. These methods proved to be better at dealing with the density-chasm, while also providing tractable density ratios (and decent generation results in [6]).
>
> We thank the reviewer for pointing us to these very interesting works. We would like to first clarify that **in our method, multi-stage ratio estimation or TRE serves only as a regularizer**, we observe in **tab. 1, 2 and 8** that although using this regularizer indeed improves performances, our method alone also shows decent results. Therefore, DRE-$\infty$ or CDM (which is a discrete-time diffusion model implemented as a classifier over time) is **orthogonal or complementary to our work**, as we study the objective itself.
>
> That being said, we conduct pilot studies again on LEBM on cifar-10 dataset for these two methods. We use exactly the same setting as we did in Sec. 4.1 particularly **with the same LEBM architecture**, except that we replace our N$^2$CE objective with (a) DRE-$\infty$ vanilla time loss (Equ. (5) in the DRE-$\infty$ paper) or (b) CDM loss.
>
> For DRE-$\infty$, we note that it (a) is admittedly slow to converge, which is also mentioned in the corresponding DRE-$\infty$ paper in F.4 section in the appendix, and (b) **requires more than 100x number of function evaluations (NFEs) compared with our method for 1 density query**, since one density query of DRE-$\infty$ models requires an integral over time, which is also mentioned in F.4 in the appendix. **Models trained with N$^2$CE requires only 1 NFE**. For reference, DRE-$\infty$ models require on average $120-270$ NFEs to complete 1 density query on MNIST dataset (see F.4 in the DRE-$\infty$ appendix). And (c) DRE-$\infty$ performs worse than our method. Particularly we implemented a Trapezoidal rule for integral over t and **allow for 100 NFEs for 1 density query**. We observe that the LEBM learned with the DRE-$\infty$ objective can obtain a FID score of 95.42, **much worse than that from the LEBM learned by N2CE** w/o TRE (FID: 77.35 in tab. 1).
>
> For CDM, we consider two different settings. (i) We set T=3 to be consistent with our usage of TRE, and (ii) T=1000 to respect the original default setting. We note that the LEBM learned with (i) can obtain a FID score of 80.67, while with (ii) can obtain a FID score of 83.15. We **observe no clear advantages of using the CDM loss alone**, while in order to accommodate for the classification head of CDM we need to further increase the output dimensions from 1 to $T+2$.
>
> Additionally, **it remains unclear how to adapt CDM to experiment settings where $q_0$ is non-gaussian, evolving and implicitly represented by samplers.** It seems non-trivial to re-purpose the CDM formulation to obtain a reliable and indeed effective reward/critic in these situations, for example settings we used in Section 4.2 of our paper. On the contrary, **our method have already demonstrated decent results on learning rewards or critics on high-dimensional datasets like ImageNet64. Please see tabs. 3,4 and 5**
>
> Moreover, **training CDM requires backpropagating twice in one training iteration**. Specifically, during training CDM needs to **first take the gradient w.r.t. its input** to get the $\epsilon$ prediction, and then **take the gradient w.r.t. to its parameters to backpropagate twice** to update its parameters, as explicitly mentioned in, for example, Algorithm 1 in the CDM paper. This renders it prohibitively expensive to scale this method to large scale datasets like ImageNet64, which has $> 1.2M$ high-dimensional samples. As reference datapoints, in the CDM paper appendix C.2.2 and C.3.2, training a CDM **on the cifar-10 dataset requires 35 hrs for 500K training iterations, with 4xA6000 GPUs**. While a forward-only training method like DDPM **requires on the cifar-10 dataset <6.7 hrs for 500K training iterations, with the same architecture (as mentioned in the CDM appendix C.1), with 8xV100 GPUs (even less compute!)**. Please refer to the DDPM paper appendix B section for detailed training efficiency report. To be clear, 4xA6000 GPUs have a peak compute of $\sim 154.8$ TFLOPs, while 8xV100 GPUs have a peak compute of only $\sim 125.6$ TFLOPS. These results are on cifar-10 dataset, and scaling up to ImageNet64 dataset will only enlarge the gap as the backpropagation through its input requires quadratically more compute.
>
> Our method is a forward-only training method, and as we stated above, has already demonstrated practical and decent results on large scale datasets like ImageNet64.

---

> ### Author Response · Authors · 2025-11-21
> **Response to Reviewer PhJz (3/N)**
>
> **Density chasm**
>
> > it is a well-known fact that NWJ (an $M \to \infty$ limit of N$^2$CE) still suffers from this exact problem (...)
>
> We believe that **we have made it clear that NWJ is only a special case of our model familiy. And these two are equivalent only in prohibitively expensive scenarios, where $n \to \infty$ and $M \to \infty$.**
>
> **Motivation**
>
> > Why did you focus on proving the convergence of the gradients, but not of the loss function itself?
>
> We believe **we have made it clear in both abstract (L16-20), introduction (L51-75), our theorem 3.2 and related remarks (L162-186) that we care about the optimization of this objective.**
>
> Finally, we appreciate that the reviewer enriches the connection of our N$^2$CE objective by bringing NWJ to our attention. We agree that with further discussions regarding NWJ and other related works like DRE-$\infty$ and CDM we could further improve the quality of our manuscript, and have better positioning of our paper. However, with our further clarification and empirical evidence, we fail to see why our submission should be rejected because of either NWJ or DRE-$\infty$ or CDM. We therefore kindly ask for raising the rating of our submission.

---

> ### Comment · Reviewer_PhJz · 2025-11-23
> **Thank you**
>
> Dear Authors,
>
> Thank you for your enormous and detailed response! It will take me some time to provide point-to-point replies. However, I can already say that you have eased some of my concerns and pinpointed the exact places in the Appendix that I had missed.
>
> In this short and preliminary response, I would like to briefly address the most important parts of your reply.
>
> 1. **Background and connection with NWJ**
>
>    While the paper is overall good and well-structured (as reflected in my initial review and scores for soundness and presentation), the neglect of previous works is almost criminal. NWJ is employed in one of the most fundamental pre-diffusion generative frameworks ($f$-GAN [2], 2100+ citations) and lies at the core of the most popular neural Mutual Information estimator (MINE [7], 2600+ citations). It also lies at the other end of the N$^2$CE spectrum, which makes me believe that NWJ deserves an equal (to those of NCE) portion of recognition in the main text.
>
>    Additionally, in its current form, the manuscript lacks a fundamental understanding of these objectives and the connection between them. This connection is derived through the means of "crude calculus," while an information-theoretic explanation is much more important, as it also explains *why* the gradients approach those of MLE.
>
> 2. **Is N$^2$CE better than NWJ?**
>
>    I thank the authors for pointing out the exact locations in the Appendix that show the existence of an optimal $M$. I strongly suggest moving these parts to the main text, as Proposition 3.3 is not sufficient here (after all, it is an upper bound, which may become loose at high $M$).
>
>    I also thank you for the direct comparison between NWJ and N$^2$CE. However, **I am not able to reproduce the results from Figure 5**. My implementation of NWJ appears to be orders of magnitude more stable than N$^2$CE for $M=10^9$ (please use scientific notation hereinafter so the reader won't have to count zeros). Below are my results for $n=500$:
>
>    | Step | 0 | 10 | 20 | 40 | 80 | 149 |
>    | :--- | :--- | :--- | :--- | :--- | :--- | :--- |
>    | $\|\| \theta\_t - \theta^* \|\|\_2$ | 4.7434 | 0.4908 | 0.2783 | 0.0823 | 0.0835 | 0.0829 |
>
>    Here is the code that I used:
>
> ```python3
> import torch
>
> class GCritic(torch.nn.Module):
>     def __init__(self, initial_alpha):
>         super().__init__()
>
>         self.alpha = torch.nn.Parameter(initial_alpha)
>
>     def forward(self, x):
>         return torch.sum(x * self.alpha[None,:], axis=-1) - 0.5 * torch.sum(self.alpha * self.alpha)
>
> true_alpha = torch.tensor([-1.5, -0.75, 0.0, 0.75, 1.5])
> initial_alpha = -true_alpha
>
> model = GCritic(initial_alpha)
> optim = torch.optim.SGD(model.parameters(), lr=0.2)
>
> batch_size = 500
>
> errors = []
>
> for step in range(150):
>     batch = torch.randn((batch_size, 5)) + true_alpha
>     noise_batch = torch.randn((batch_size, 5))
>
>     with torch.no_grad():
>         errors.append(torch.sqrt(torch.sum((true_alpha - model.alpha)**2)))
>
>     optim.zero_grad()
>
>     (-model(batch).mean() + torch.exp(model(noise_batch) - 1.0).mean()).backward()
>
>     optim.step()
> ```
>
> The results are also better than those of N$^2$CE if I sample a single batch at the beginning.
>
> I am still open for discussion (with a possibility of increasing the score) and still going to answer the other points of your rebuttal. However, my opinion is also strong: while the contribution is decent, the paper requires a major revision.
>
> [7] M. Belghazi et al., "MINE: Mutual Information Neural Estimation." Proc. of ICML 2018

---

> ### Author Response · Authors · 2025-11-24
> **Thank you! (1/N)**
>
> We would like to thank reviewer PhJz for going through our responses and involving in the discussion with us. We are glad and actually grateful that reviewer PhJz now acknowledge that our contribution is decent, and will consider increasing the score. Below we will provide point-to-point (and try-to-be concise) responses to respect reviewer's response style and reduce the possible workload, which hopefully help to address the issues.
>
> **Background and connection with NWJ**
>
> We indeed greatly appreciate reviewer PhJz's comments and derivation in the original review to establish further a connection between our method and previous objectives such as NWJ. We agree with and highly respect the information-theoretic explanation, and consider this as a valuable addition to our current manuscript. In fact, we would like to rephrase and organize the corresponding part of reviewer PhJz's review as a new and separate section named as "An Information-Theoretic Point of View" or other appropriate names, right after our current Sec 3.3, if reviewer PhJz would not mind extending a permission for this. We would also like reviewer PhJz and other program committee members to advise on how to appropriately acknowledge reviewer PhJz's contribution to this part, in addition to explicitly mentioning it in the final acknowledgement section.
>
> **Is N$^2$CE better than NWJ?**
>
> > (...) in the Appendix that show the existence of an optimal $M$. I strongly suggest moving these parts to the main text, as Proposition 3.3 is not sufficient here (...)
>
> We totally agree and fully accept this suggestion. We will add another paragraph in Sec. 3.3 named as "N$^2$CE as a family and U-shaped patterns" to more explicitly explain this observation.
>
> > (...) However, I am not able to reproduce the results from Figure 5. (...)
>
> We appreciate that reviewer PhJz spends extra time to check the details. We provide responses below in two parts: (a) we would like to clarify the experiment settings used in Fig. 5, and (b) we respect and further experiment on reviewer PhJz's code to provide consistent, clear and scientific empirical evidence. To respect the transparency of reviewer PhJz's response, we also attach all the self-contained code pieces used in these experiments at the end of our responses.
>
> First we would like to briefly address the concerns of reproducing our Fig. 5. **We noted several inconsistencies between the reviewer's implementation and ours**. (i) For plotting Fig. 5 we used the *explicit population-wise gradient* equation of N$^2$CE in D.1, $E_{q_*}\left[\tfrac{M}{M+r_{\alpha}(x)}\,\left(x - \alpha\right)\right] - E_{p_{\alpha}}\left[\tfrac{M}{M+r_{\alpha}(x)}\,\left(x - \alpha\right)\right] $, to calculate the gradient. This is because we care about the gradient approximation itself in this simulation. Specifically, we draw $n$ MC samples in each iteration to construct a finite-sample estimation of this population-wise gradient. For NWJ, we used also the grad. equation in D.1 to estimate it's empirical grad. (ii) For producing Fig. 5(b), where $M=1e9$, as mentioned in our previous response, we used $n=2$ to mimic the high-variance regime, instead of $n=500$. (iii) The NWJ objective we used, as suggested in reviewer PhJz's previous response, uses the paramterization of $T = \log r$, instead of $T = \log r - 1$. Although in terms of expectation this does not change the optimum of the objective, in practice with finite samples it leads to different estimation variance. (iv) We used numpy for implementation, which uses float64 by default, while pytorch uses float32. We kindly refer to our code for more details.

---

> ### Author Response · Authors · 2025-11-24
> **Thank you! (2/N)**
>
> Despite these inconsistencies, we respect and further experiment on reviewer PhJz's implementation to provide side-by-side comparable supporting evidence. To account for the variability of different settings, we experiment on (a) both $n=2$ and $n=500$ sample sizes, and (b) compare our N$^2$CE implementation with both NWJ parameterizations, and we calculate the objective and use pytorch's autograd and optimizer to update the parameters. Further, to make sure that the comparison is systematic and scientific, ruling out wins/losses by chance (which we believe is one reason why the reviewer did not observe instability in NWJ but instead in ours, i.e., due to different random seeds), we consider running multiple independent runs using these settings, and report the mean and std. of the results.
>
> Specifically, we use the average MSE of the whole trajectory as the metric: $\frac{1}{T} \sum_{t=0}^{T-1} \| \theta_t - \theta^* \|_2^2$ for one run. This accounts for the trajectory-wise errors, i.e., summary of one run. We repeat independent experiments and calculate the mean and std of 100 runs to account for randomness caused by different seeds.
>
> With sample $n=2$, the optimal $M$ suggested by Proposition 3.3 is at the scale of $C\sqrt{2} \sim C * 1.414$. We sweep over $M$s:
>
> | M | NWJ ($T=\log r - 1$) |NWJ ($T=\log r$)| 1 | 1.5 | 2 | 5| 10 | 100 | 1000| 1e9|
> |---------|----------|----------|---------|----------|----------|---------|----------|----------|----------|----------|
> | avg. runs mse | $ \underline{757.612 \pm 7525.349} $ | $\color{gray}54.548 \pm 237.528$ | $0.904 \pm 0.084$ | $\mathbf{0.884 \pm 0.083}$ | $0.888 \pm 0.084$ | $0.983 \pm 0.107$ | $1.139 \pm 0.168$ | $3.158 \pm 2.683 $| $17.564 \pm 42.751$| $\underline{\color{gray}61.291 \pm 277.336}$ |
>
> With $n=500$, the optimal $M$ suggested by Proposition 3.3 is at the scale of $C\sqrt{500} \sim C * 22.361$. We sweep over $M$s:
>
> | M | NWJ ($T = \log r - 1$) | NWJ ($T=\log r$) | 1 | 10 | 50 | 100 | 1000| 1e4| 2e4| 1e9|
> |---------|----------|----------|---------|----------|----------|---------|----------|----------|----------|----------|
> | avg. runs mse | $\underline{0.531 \pm 0.761}$ | $\color{gray}1.359 \pm 5.454$ | $0.678 \pm 0.004$ | $0.489 \pm 0.004$ | $0.456 \pm 0.006$ | $\mathbf{0.453 \pm 0.007}$ | $0.489 \pm 0.020 $| $0.641 \pm 0.324$ | $\underline{0.750 \pm 0.889}$ | $\color{gray}1.909 \pm 11.201$ |
>
> For each table, we highlight the corresponding cells where the scale of stds match those of NWJ's stds. We can see that (i) with $n=2/500$, the optimal-$M$ N$^2$CEs is indeed much better than those of NWJs, demonstrated by both lower mean values and lower stds. **This indicate that N$^2$CEs are indeed better than NWJs, oftentimes even with $M=1e9$**. (ii) The U-shaped pattern emerges again in both tabs, and the optimal-$M$ suggested by Proposition 3.3 fits well in these simulations.
>
> We also remain open for discussion and willing to answer any further questions from reviewer PhJz. Hopefully our further responses can help to address the concerns.
>
> As promised, we attach below first the code for plotting Fig. 5(b) with our setting, and the code for implementing these simulations based on reviewer's code. Please modify L104-108 to simulate for different $M$s and sample size $n$. Please comment/uncomment L53-54 to use either parameterization of NWJ. Please note that the first code piece requires matplotlib and seaborn as plotting packages to make it work.

---

> > ### Author Response · Authors · 2025-11-24
> > **Our code for plotting Fig. 5(b) (1/N)**
> >
> > ~~~
> > import numpy as np
> > import matplotlib.pyplot as plt
> > import seaborn as sns
> > from cycler import cycler
> >
> > # =========================
> > #  Seaborn style & palette
> > # =========================
> > sns.set_theme(style="darkgrid", context="talk")
> > palette = sns.color_palette("mako_r", 10)
> > plt.rcParams["axes.prop_cycle"] = cycler(color=palette)
> > LEGEND_FS = 10
> > plt.rcParams.update({
> >     "legend.fontsize": LEGEND_FS,
> >     "grid.alpha": 0.35,
> >     "grid.linestyle": "--",
> >     "grid.linewidth": 0.8,
> >     "legend.frameon": True,
> >     "legend.framealpha": 0.9,
> >     "legend.fancybox": True,
> >     "lines.linewidth": 2.2,
> >     "axes.titlesize": 12,
> >     "axes.labelsize": 12,
> >     "xtick.labelsize": 10,
> >     "ytick.labelsize": 10,
> > })
> >
> > def enable_minor_grid(ax):
> >     ax.minorticks_on()
> >     ax.grid(which="minor", linestyle=":", linewidth=0.6, alpha=0.25)
> >
> > # =========================
> > #  Setup & utilities
> > # =========================
> > rng = np.random.default_rng(0)
> >
> > d = 5                      # higher dimension to make variance differences clearer
> > theta_star = np.linspace(1.5, -1.5, d)
> > theta0 = -theta_star       # start on the opposite side
> > lr = 0.2
> > T = 150
> > n_samples = 2              # MC samples for NCE/N²CE/NWJ estimates at each step
> >
> > M = 1000000000
> >
> > # q0 ~ N(0, I); p_theta ~ N(theta, I); q* ~ N(theta_star, I)
> > def sample_normal(mean, n, rng):
> >     return mean + rng.standard_normal(size=(n, mean.size))
> >
> > def log_ratio(theta, x):
> >     # log r_theta(x) = theta^T x - 0.5 ||theta||^2  (since p_theta / q0)
> >     return x @ theta - 0.5 * np.dot(theta, theta)
> >
> > def mle_grad(theta):
> >     # ∇ J_MLE = E_{q*}[x] - E_{p_theta}[x] = theta_star - theta (analytic here)
> >     return theta_star - theta
> >
> > def n2ce_grad(theta, M, n_samples, rng):
> >     """
> >     ∇ L_M = E_{q*}[ w_M(x) (x - a) ] - E_{p_theta}[ w_M(x) (x - a) ],
> >     w_M(x) = M / (M + r_theta(x)), r_theta(x) = exp(log_ratio)
> >     Approximated via Monte Carlo.
> >     """
> >     x_star = sample_normal(theta_star,  n_samples, rng)
> >     x_mod  = sample_normal(theta,       n_samples, rng)
> >
> >     r_star = np.exp(log_ratio(theta, x_star))
> >     r_mod  = np.exp(log_ratio(theta, x_mod))
> >
> >     w_star = M / (M + r_star)
> >     w_mod  = M / (M + r_mod)
> >
> >     g_star = (w_star[:, None] * (x_star - theta)).mean(axis=0)
> >     g_mod  = (w_mod[:, None]  * (x_mod - theta)).mean(axis=0)
> >     return g_star - g_mod
> >
> > def nwj_grad(theta, n_samples, rng):
> >     """
> >     NWJ gradient with T(x) = log r_theta(x).
> >
> >     Objective (up to a constant):
> >         L_NWJ(θ) = E_{q*}[log r_θ(x)] - E_{q0}[r_θ(x)].
> >
> >     Gradient:
> >         ∇_θ L = E_{q*}[∇_θ log r_θ(x)] - E_{q0}[∇_θ r_θ(x)]
> >                = E_{q*}[x - θ] - E_{q0}[r_θ(x) (x - θ)].
> >     """
> >     # Data samples from q* = N(theta_star, I)
> >     x_star = sample_normal(theta_star, n_samples, rng)
> >     # Noise samples from q0 = N(0, I)
> >     x0 = sample_normal(np.zeros_like(theta), n_samples, rng)
> >
> >     grad_log_r_star = x_star - theta         # ∇_θ log r_θ(x) under q*
> >     grad_log_r_0    = x0 - theta             # ∇_θ log r_θ(x) under q0
> >     r0 = np.exp(log_ratio(theta, x0))        # r_θ(x) under q0
> >
> >     g_star  = grad_log_r_star.mean(axis=0)
> >     g_noise = (r0[:, None] * grad_log_r_0).mean(axis=0)
> >
> >     return g_star - g_noise
> >
> > # =========================
> > #  Runner
> > # =========================
> > def run_method(name, update_grad_fn, **kwargs):
> >     """
> >     Tracks:
> >       - thetas[t]
> >       - distances[t] = ||theta_t - theta*||
> >       - cos_to_mle[t] = cosine( g_method(theta_t), g_mle(theta_t) )
> >       - g_mle_norms[t] = ||g_mle(theta_t)||_2
> >       - abs_err[t]     = ||g_method - g_mle||_2
> >     """
> >     theta = theta0.copy()
> >     thetas = [theta.copy()]
> >     dists = [np.linalg.norm(theta - theta_star)]
> >     cos_mle = []
> >     g_mle_norms = []
> >     abs_errs = []
> >
> >     for _ in range(T):
> >         # compute method grad and MLE grad at current theta
> >         g_method = update_grad_fn(theta, **kwargs)
> >         g_mle_at_theta = mle_grad(theta)
> >
> >         # update
> >         theta = theta + lr * g_method
> >         dists.append(np.linalg.norm(theta - theta_star))
> >
> >     return {
> >         "name": name,
> >         "distances": np.array(dists),           # (T+1,)
> >     }

---

> > > ### Author Response · Authors · 2025-11-24
> > > **Our code for plotting Fig. 5(b) (2/N)**
> > >
> > > ~~~
> > > # =========================
> > > #  Define methods & run
> > > # =========================
> > > runs = []
> > >
> > > # 1) MLE (analytic gradient)
> > > runs.append(
> > >     run_method(
> > >         "MLE (limit M→∞)",
> > >         update_grad_fn=lambda th: mle_grad(th)
> > >     )
> > > )
> > >
> > > # 2) NWJ with T = log r_theta
> > > local_rng_nwj = np.random.default_rng(777)
> > > runs.append(
> > >     run_method(
> > >         "NWJ (T = log r)",
> > >         update_grad_fn=lambda th, r=local_rng_nwj: nwj_grad(th, n_samples=n_samples, rng=r)
> > >     )
> > > )
> > >
> > > # 3) N²CE
> > > Ms = [M]
> > > for M in Ms:
> > >     local_rng = np.random.default_rng(42 + M)
> > >     r = run_method(
> > >         "NCE / N²CE (M=1)" if M == 1 else "N²CE (M={})".format(M),
> > >         update_grad_fn=lambda th, M=M, r=local_rng: n2ce_grad(th, M, n_samples, r)
> > >     )
> > >     runs.append(r)
> > >
> > > # =========================
> > > #  Plot 1: distance vs iter
> > > # =========================
> > > fig, ax = plt.subplots(figsize=(7.2, 4.6))
> > > light_blue = sns.color_palette("Blues", 6)[1]
> > > for r in runs:
> > >     if "MLE" in r["name"]:
> > >         ax.plot(r["distances"], label=r["name"], color='blue', linewidth=2.8, alpha=0.95)
> > >     elif "NWJ" in r["name"]:
> > >         ax.plot(r["distances"], label=r["name"], color='gray')
> > >     else:
> > >         ax.plot(r["distances"], label=r["name"])
> > > ax.set_xlabel("Iteration")
> > > ax.set_ylabel(r"$\|\theta_t - \theta^\ast\|_2$")
> > > ax.set_title("Distance to target parameter")
> > > enable_minor_grid(ax)
> > > ax.legend(ncol=2, fontsize=8)
> > > ax.set_ylim(-0.24, 5.00)
> > > fig.tight_layout()
> > > fig.savefig("./distance_vs_iter_seaborn.pdf", bbox_inches="tight")
> > > plt.close(fig)

---

> > > > ### Author Response · Authors · 2025-11-24
> > > > **Our code for further results**
> > > >
> > > > ~~~
> > > > import torch
> > > > import torch.nn.functional as F
> > > > import numpy as np
> > > >
> > > > # -------------------------
> > > > #  Model (same as before)
> > > > # -------------------------
> > > > class GCritic(torch.nn.Module):
> > > >     def __init__(self, initial_alpha):
> > > >         super().__init__()
> > > >         self.alpha = torch.nn.Parameter(initial_alpha)
> > > >
> > > >     def forward(self, x):
> > > >         # f_alpha(x) = alpha^T x - 0.5 ||alpha||^2
> > > >         return torch.sum(x * self.alpha[None, :], axis=-1) - 0.5 * torch.sum(self.alpha * self.alpha)
> > > >
> > > >
> > > > def run_single_experiment(batch_size=500, lr=0.2, n_steps=150, M_n2ce=1e9, seed=0):
> > > >     """
> > > >     Run one NWJ vs N2CE experiment and return:
> > > >       - traj_mse_nwj:  trajectory-averaged MSE for NWJ
> > > >       - traj_mse_n2ce: trajectory-averaged MSE for N2CE
> > > >     where MSE is mean_t ||alpha_t - alpha*||_2^2 over steps.
> > > >     """
> > > >     # Make runs reproducible
> > > >     torch.manual_seed(seed)
> > > >     np.random.seed(seed)
> > > >
> > > >     true_alpha = torch.tensor([-1.5, -0.75, 0.0, 0.75, 1.5])
> > > >
> > > >     # -------------------------
> > > >     #  NWJ run
> > > >     # -------------------------
> > > >     initial_alpha = -true_alpha
> > > >     model = GCritic(initial_alpha.clone())
> > > >     optim = torch.optim.SGD(model.parameters(), lr=lr)
> > > >
> > > >     errors_nwj = []  # store ||alpha_t - alpha*||_2 at each step
> > > >
> > > >     for step in range(n_steps):
> > > >         batch = torch.randn((batch_size, 5)) + true_alpha     # q* = N(true_alpha, I)
> > > >         noise_batch = torch.randn((batch_size, 5))            # q0 = N(0, I)
> > > >
> > > >         with torch.no_grad():
> > > >             err = torch.linalg.norm(true_alpha - model.alpha)
> > > >             errors_nwj.append(err.item())
> > > >
> > > >         optim.zero_grad()
> > > >
> > > >         # NWJ loss: -E_q*[f] + E_q0[exp(f - 1)] or
> > > >         #           -E_q*[f] + E_q0[exp(f)]
> > > >         loss_nwj = -model(batch).mean() + torch.exp(model(noise_batch) - 1).mean()
> > > >         # loss_nwj = -model(batch).mean() + torch.exp(model(noise_batch)).mean()
> > > >         loss_nwj.backward()
> > > >         optim.step()
> > > >
> > > >     errors_nwj = np.array(errors_nwj)
> > > >     traj_mse_nwj = float(np.mean(errors_nwj ** 2))  # trajectory-averaged MSE
> > > >
> > > >     # -------------------------
> > > >     #  N2CE run
> > > >     # -------------------------
> > > >     def N2CE(model, x_p, x_n, M):
> > > >         logr_pos, logr_neg = model(x_p) - np.log(M), model(x_n) - np.log(M)
> > > >
> > > >         loss_pos = F.binary_cross_entropy_with_logits(
> > > >             logr_pos, torch.ones_like(logr_pos), reduction="none"
> > > >         )
> > > >         loss_neg = M * F.binary_cross_entropy_with_logits(
> > > >             logr_neg, torch.zeros_like(logr_neg), reduction="none"
> > > >         )
> > > >         return loss_pos.mean() + loss_neg.mean()
> > > >
> > > >     # Re-init model & optimizer
> > > >     model = GCritic(initial_alpha.clone())
> > > >     optim = torch.optim.SGD(model.parameters(), lr=lr)
> > > >
> > > >     errors_n2ce = []
> > > >
> > > >     for step in range(n_steps):
> > > >         batch = torch.randn((batch_size, 5)) + true_alpha
> > > >         noise_batch = torch.randn((batch_size, 5))
> > > >
> > > >         with torch.no_grad():
> > > >             err = torch.linalg.norm(true_alpha - model.alpha)
> > > >             errors_n2ce.append(err.item())
> > > >
> > > >         optim.zero_grad()
> > > >         loss_n2ce = N2CE(model, batch, noise_batch, M_n2ce)
> > > >         loss_n2ce.backward()
> > > >         optim.step()
> > > >
> > > >     errors_n2ce = np.array(errors_n2ce)
> > > >     traj_mse_n2ce = float(np.mean(errors_n2ce ** 2))
> > > >
> > > >     return traj_mse_nwj, traj_mse_n2ce
> > > >
> > > >
> > > > if __name__ == "__main__":
> > > >     # -------------------------
> > > >     #  Hyperparameters
> > > >     # -------------------------
> > > >     N_RUNS = 100
> > > >     batch_size = 500 # 2
> > > >     lr = 0.2
> > > >     n_steps = 150
> > > >     M_n2ce = 1
> > > >
> > > >     traj_mse_nwj_runs = []
> > > >     traj_mse_n2ce_runs = []
> > > >
> > > >     for run in range(N_RUNS):
> > > >         traj_mse_nwj, traj_mse_n2ce = run_single_experiment(
> > > >             batch_size=batch_size,
> > > >             lr=lr,
> > > >             n_steps=n_steps,
> > > >             M_n2ce=M_n2ce,
> > > >             seed=run,  # different seed per run
> > > >         )
> > > >         traj_mse_nwj_runs.append(traj_mse_nwj)
> > > >         traj_mse_n2ce_runs.append(traj_mse_n2ce)
> > > >
> > > >         # print(
> > > >         #     f"Run {run:02d}: "
> > > >         #     f"traj MSE NWJ = {traj_mse_nwj:.6f}, "
> > > >         #     f"traj MSE N2CE = {traj_mse_n2ce:.6f}"
> > > >         # )
> > > >
> > > >     traj_mse_nwj_runs = np.array(traj_mse_nwj_runs)
> > > >     traj_mse_n2ce_runs = np.array(traj_mse_n2ce_runs)
> > > >
> > > >     print("\n=== Summary over runs ===")
> > > >     print(f"N runs = {N_RUNS}")
> > > >     print(
> > > >         f"NWJ:  mean traj-MSE = {traj_mse_nwj_runs.mean():.6f}, "
> > > >         f"std = {traj_mse_nwj_runs.std(ddof=1):.6f}"
> > > >     )
> > > >     print(
> > > >         f"N2CE: mean traj-MSE = {traj_mse_n2ce_runs.mean():.6f}, "
> > > >         f"std = {traj_mse_n2ce_runs.std(ddof=1):.6f}"
> > > >     )

---

> ### Comment · Reviewer_PhJz · 2025-11-26
>
> Dear Authors,
>
> Thank you for engaging in the discussion and providing further details.
>
> **Experiments**
>
> I have updated my code to adopt both $n=2$ and $n=500$, as well as seed/trajectory-wise averaging. It now also uses different parametrizations of NWJ (the $\log r - 1$ parametrization is from the original works [1-2], while the $\log r$ parametrization is a direct limit of N$^2$CE). The results are provided below, and the new code is in a separate comment.
>
> $n=2$
>
> | M | NWJ ($T = \log r$) | NWJ ($T = \log r - 1$) | NWJ ($T = \log r - 4$) | $1$ | $1.5$ | $2$ | $5$ | $10$ | $10^2$ | $10^3$ | $10^9$ |
> |--|--|--|--|--|--|--|--|--|--|--|--|
> |avg. runs MSE| $304 \pm 2815$ | $26 \pm 230$ | $0.73 \pm 0.35$ | $0.922 \pm 0.083$ | $0.904 \pm 0.086$ | $0.908 \pm 0.092$ | $0.999 \pm 0.133$ | $1.150 \pm 0.197$ | $3.24 \pm 3.31$ | $22 \pm 115$ | $304 \pm 2815$ |
>
> $n=500$
>
> | M | NWJ ($T = \log r$) | NWJ ($T = \log r - 1$) | NWJ ($T = \log r - 4$) | $1$ | $1.5$ | $2$ | $5$ | $10$ | $10^2$ | $10^3$ | $10^9$ |
> |--|--|--|--|--|--|--|--|--|--|--|--|
> |avg. runs MSE| $1.07 \pm 3.18$ | $0.47 \pm 0.25$ | $0.418 \pm 0.002$ | $0.678 \pm 0.004$ | $0.618 \pm 0.004$ | $0.585 \pm 0.004$ | $0.518 \pm 0.004$ | $0.489 \pm 0.004$ | $0.452 \pm 0.006$ | $0.487 \pm 0.024$ | $1.075 \pm 3.18$ |
>
> The results are now more consistent with yours and also self-consistent (N$^2$CE now indeed approaches NWJ when $T = \log r$ and $M \to \infty$; *however, strangely, I do not observe this behaviour in the authors' table*).
>
> I consider this a satisfactory outcome. Please, include your enhanced (100 seeds + averaging) results in the manuscript.
>
> **Revision and final score**
>
> Provided the authors revise their manuscript in accordance with our discussion (which includes a discussion of NWJ in the main text that is roughly equal to that of NCE, providing an information-theoretic perspective, and more compelling arguments for the existence of an optimal $M$), I am ready to raise my score to at least a 6 (weak accept).
>
> **Legal stuff**
>
> According to [OpenReview terms of use](https://openreview.net/legal/terms), public comments are under the Creative Commons Attribution 4.0 International (CC BY 4.0) license. Therefore, the authors are free to use the results of this discussion under these terms.

---

> ### Comment · Reviewer_PhJz · 2025-11-26
>
> ```python3
> import math
> import numpy
> import torch
>
> class GCritic(torch.nn.Module):
>     def __init__(self, initial_alpha):
>         super().__init__()
>
>         self.alpha = torch.nn.Parameter(initial_alpha)
>
>     def forward(self, x):
>         return torch.sum(x * self.alpha[None,:], axis=-1) - 0.5 * torch.sum(self.alpha * self.alpha)
>
> def NWJ_loss(model, batch, noise_batch, shift: float=1.0):
>     return -model(batch).mean() + torch.exp(model(noise_batch) - shift).mean()
>
> def N2CE_loss(model, batch, noise_batch, M: float=1.0):
>     log_M = math.log(M)
>
>     log_r_batch = model(batch)
>     log_r_noise_batch = model(noise_batch)
>
>     first_term  = log_r_batch.mean() - torch.log(M + log_r_batch.exp()).mean()
>     second_term = M * (log_M - torch.log(M + log_r_noise_batch.exp()).mean())
>
>     return -(first_term + second_term)
>
> def perform_run(
>     true_alpha=torch.tensor([-1.5, -0.75, 0.0, 0.75, 1.5]),
>     lr=0.2,
>     batch_size=500,
>     n_steps=150,
>     loss=NWJ_loss
> ):
>     initial_alpha = -true_alpha
>
>     model = GCritic(initial_alpha)
>     optim = torch.optim.SGD(model.parameters(), lr=lr)
>
>     errors = numpy.zeros(n_steps)
>     for step in range(n_steps):
>         batch = torch.randn((batch_size, initial_alpha.shape[0])) + true_alpha
>         noise_batch = torch.randn((batch_size, initial_alpha.shape[0]))
>
>         with torch.no_grad():
>             errors[step] = torch.sum((true_alpha - model.alpha)**2).item()
>
>         optim.zero_grad()
>         loss(model, batch, noise_batch).backward()
>         optim.step()
>
>     return errors.mean(), errors[-1]
>
> from functools import partial
>
> loss_grid = {
>     "NWJ-s-1":   partial(NWJ_loss, shift=-1.0),
>     "NWJ-s0":    partial(NWJ_loss, shift=0.0),
>     "NWJ-s1":    partial(NWJ_loss, shift=1.0),
>     "NWJ-s2":    partial(NWJ_loss, shift=2.0),
>     "NWJ-s3":    partial(NWJ_loss, shift=3.0),
>     "NWJ-s4":    partial(NWJ_loss, shift=4.0),
>     "M=1e0":     N2CE_loss,
>     "M=1.5":     partial(N2CE_loss, M=1.5),
>     "M=2.0":     partial(N2CE_loss, M=2),
>     "M=5.0":     partial(N2CE_loss, M=5),
>     "M=1e1":     partial(N2CE_loss, M=10),
>     "M=1e2":     partial(N2CE_loss, M=100),
>     "M=1e3":     partial(N2CE_loss, M=1000),
>     "M=1e4":     partial(N2CE_loss, M=1e4),
>     "M=1e5":     partial(N2CE_loss, M=1e5),
>     "M=1e6":     partial(N2CE_loss, M=1e6),
>     "M=1e9":     partial(N2CE_loss, M=1e9),
> }
>
> batch_size_grid = [2, 500]
>
> n_runs = 100
>
> from tqdm import tqdm
>
> rows = []
>
> for loss_name, loss in tqdm(loss_grid.items()):
>     for batch_size in batch_size_grid:
>         trajectory_errors = numpy.empty(n_runs)
>         final_errors      = numpy.empty(n_runs)
>
>         # Seed everything
>         seed = 42
>
>         torch.manual_seed(seed)
>         torch.cuda.manual_seed(seed)
>         torch.backends.cudnn.deterministic = True
>         torch.backends.cudnn.benchmark = False
>
>         numpy.random.seed(seed)
>
>         for run in range(n_runs):
>             trajectory_errors[run], final_errors[run] = perform_run(loss=loss, batch_size=batch_size)
>
>         rows.append({
>             "loss": loss_name,
>             "batch_size": batch_size,
>             "trajectory_mean": trajectory_errors.mean(),
>             "trajectory_std": trajectory_errors.std(),
>             "final_mean": final_errors.mean(),
>             "final_std": final_errors.std(),
>         })
>
> import pandas
>
> results = pandas.DataFrame(rows)
> ```

---

> ### Author Response · Authors · 2025-11-29
> **Thank you!**
>
> We would like to sincerely thank reviewer PhJz again for being **(i) responsive:** kept engaging in the discussion with us **since the very early stage of the rebuttal session**, **(ii) professional:** providing constructive, concrete and fruitful feedback that greatly enhances the quality of our manuscript, and **(iii) reasonable:** quoted **``ready to raise the score to at least a 6 (weak accept).''** after going through our point-to-point responses and clarification. We enjoy and appreciate the opportunity to discuss in-depth with reviewer PhJz.
>
> We have revised and updated the manuscript accordingly (highlighted in red) following reviewer PhJz's final suggestions:
>
> > (...) includes a discussion of NWJ in the main text that is roughly equal to that of NCE, (...)
>
> We have update Sec. 2 at L93 and L112-124 to further introduce the NWJ framework as our background.
>
> > (...) providing an information-theoretic perspective, (...)
>
> As promised, we have added a new Sec. 3.4 `An Information-Theoretic Perspective` L252-285 to incorporate the discussion and reviewing comments from reviewer PhJz.
>
> > (...) and more compelling arguments for the existence of an optimal $M$
>
> As promised, we have revised Sec. 3.3 as `Practical Error Analysis, N$^2$CE Family and Regularization`, revised L219-223 and added L226-235 `The N$^2$CE family and U-shaped patterns` to make the optimal-$M$ characterization more explicit and precise.
>
> > I consider this a satisfactory outcome. Please, include your enhanced (100 seeds + averaging) results in the manuscript.
>
> We have included the results from our code in our appendix D.1 section, and provided explicit reference in the main text at L282-283. Also we would like to clarify that our results with $M=1e9$ indeed approximates the NWJ results with $T=\log r$. We think the slight gap may come from the pytorch implementation of `F.binary_cross_entropy_with_logits`.
>
> We would like to thank reviewer PhJz again for the time devoted into reviewing our manuscript and discussing with us. We will explicitly mention the contribution of reviewer PhJz in the acknowledgement section in our final manuscript.

---

### Official Review · Reviewer_Qgtk · 2025-10-31

**Soundness:** 3
**Presentation:** 2
**Contribution:** 3
**Rating:** 6
**Confidence:** 3

**Summary:**

This paper proposes a more general Noise Contrastive Estimation (NCE).
Recalling the classical sigmoid function $\sigma(x)$ and the binary-cross entropy is used to compute the NCE loss as:
$\sigma(x)=1/(1+e^{-x})$ and

$$\mathcal{L}_\text{NCE}(\theta) = \mathbb{E}_{x\sim X}[log(\sigma(f(x;\theta)))] + \mathbb{E}_{\epsilon}[log(1-\sigma(f(\epsilon;\theta)))]$$

The method proposed by the paper called Noiser NCE (N$^2$CE), it introduces a new user-provided hyper-parameter $M$ that increases the impact of noise.
The sigmoid function and losses are modified as follows:
$\sigma_M(x)=M/(M+e^{-x})$ and

$$\mathcal{L}_{\text{N}^2\text{CE}}(\theta,M) = \mathbb{E}_{x\sim X}[log(\sigma_M(f(x;\theta)))] + M\mathbb{E}_{\epsilon}[log(1-\sigma_M(f(\epsilon;\theta)))]$$

And as noted in the paper $\text{N}^2\text{CE}(.,M=1)=\text{NCE}(.)$

This surprisingly simple modification makes the gradient of the N$^2$CE loss approximate the gradient of Maximum Likelihood Estimation (MLE) when $M\to +\infty$.
In turns this allows N$^2$CE to converge better and in higher than NCE can as demonstrated by the various experiments.

**Strengths:**

1. The method is simple and easy to implement.
2. It addresses an important limitation of vanilla NCE: its difficulty to deal with high-dimension data, in which it typically revert to learning the noise directly, not the ratio data-noise.
3. The connection with MLE is proven.
4. There's no overhead training cost compared to vanilla NCE.
3. Experimental results are very convincing, showing large gains over NCE and other methods.

**Weaknesses:**

1. The paper packs too much information, notably in the experimental settings, and consequently a lot of it has been pushed to the appendix (the paper is 37 pages in total). I tend to prefer paper that focus on one idea and line of experimentation in depth which are then built-on by follow up paper.
2. Possibly as a consequence of the previous point, background information is severely lacking. For example LEBM and DAMC which are heavily used in experiments are not provided as background, making the paper not self-contained.
3. As a whole, inference is left out of the paper (apart from saying we use methods A and B for inference, again reinforcing the previous point), only training is covered.
3. Lack of experimental consistency: sometimes telescoping ratios is used while other time direct ratio regularization is used.
4. Lack of terminology consistency: telescoping ratios are also referred to as multi-stage ratio and ratio decomposition. This made me doubt my understanding, are they the same thing as I assumed or not?
5. From the above, I get the feeling it's not one method but two depending on how the ratio is regularized, taking away from the elegance of the idea.

**Questions:**

1. I gather that the loss can be seen as a modified sigmoid and a weighted binary cross entropy. But how does inference, do we use a vanilla sigmoid?

---

> ### Author Response · Authors · 2025-11-20
> **Response to Reviewer Qgtk**
>
> We would like to sincerely thank reviewer Qgtk for the time devoted into reviewing our submission. We appreciate that reviewer Qgtk raised detailed questions, to which we will provide point-to-point responses below and hopefully help to clarify the issues.
>
> **Regarding inference**
>
> > As a whole, inference is left out of the paper (apart from saying we use methods A and B for inference, again reinforcing the previous point), only training is covered. (...) a lot of it has been pushed to the appendix (the paper is 37 pages in total).
> > For example LEBM and DAMC which are heavily used in experiments are not provided as background, making the paper not self-contained.
>
> We thank reviewer Qgtk for bringing this matter to our attention. We agree that we focus primarily on the training objective in the main text, and defer most of the detailed experiment settings and implementation to appendix C. and D. Specifically, we mentioned in the main text L264-286 that experiment and inference settings of LEBM can be found in C.1 and C.2; in the main text L299-306 we mentioned that the DAMC implementation and inference settings can be found in C.2 and D.3. We admit that this may lead to extra cognitive load for readers unfamiliar with this topic. We sincerely apologize for the inconvenience and will try to incorporate as much information regarding inference as possible in revision with an additional page.
>
> > I gather that the loss can be seen as a modified sigmoid and a weighted binary cross entropy. But how does inference, do we use a vanilla sigmoid?
>
> We would like to clarify that after training with our objective, oftentimes we will obtain a density model or a reward/critic that outputs the scalar. Therefore, we can take this scalar, for example, as the unnormalized density score of p_\alpha(z) in DAMC (see L1602-1603) for inference. Specifically for the anomaly detection, we use the sum of this scalar output together with the reconstruction error ($\log p(x | z)$) as the final score for the decision function.
>
> **Regarding writing**
>
> >  (...) telescoping ratios are also referred to as multi-stage ratio and ratio decomposition. This made me doubt my understanding, are they the same thing as I assumed or not?
>
> We confirm that they are the same.
>
> > (...) sometimes telescoping ratios is used while other time direct ratio regularization is used.
> > (...) I get the feeling it's not one method but two depending on how the ratio is regularized, taking away from the elegance of the idea.
>
> We would like to clarify that in L215-229, we have explicitly stated the correspondingly suitable scenarios for these two different regularizers. They essentially serve for different purposes: ratio-decomposition may be used for cases where actual densities really matter, or when the data dimension is not of extra-high dimension; otherwise the regularizer may induce compute overhead. this regularizer also requires a pre-specified noise density and interpolation method. Direct ratio regularization on the hand, does not impose these requirements, and works well in high dimensional spaces. But it comes at a cost of possibly biasing the gradient estimation. We therefore list these two regularizers as reference for the readers.
>
> That being said, we feel sorry if the way we organize and present our work brings extra loads for the reviewer. We sincerely hope our further clarification could help to address issues. We will fully commit to further revising the draft and improving the readability in the updated version. We would like to thank the reviewer again for the time.

---

### Official Review · Reviewer_Qdxk · 2025-11-05

**Soundness:** 3
**Presentation:** 3
**Contribution:** 3
**Rating:** 6
**Confidence:** 3

**Summary:**

This paper points out the problem that Noise Contrastive Estimation (NCE), which estimates a density of the data distribution by estimating the ratio between the target density and noise density, underperforms when the ratio is large. This phenomenon commonly occurs when the data is high dimensional or multimodal. The authors propose to rescale the log odds that appear in the NCE objective by constant $M$. When $M > 1$. this corresponds to artificially increasing the magnitude of the noise. This "Noisier NCE" or N2CE reduces to the conventional NCE when $M=1$. They show that as $M$ increases the gradient of the N2CE loss with respect to the parameter, approaches to that of the log-likelihood (score function), thus N2CE amounts to maximum likelihood estimation (MLE), both theoretically and empirically. Then it is further shown that this simple remedy to NCE works effectively well in high-dimensional or multimodal settings, and generative modeling without additional cost or structural change of the models.

**Strengths:**

1. Theoretical results indicate that that gradient of the population version of the N2CE objective converges to the score function as $M$ increases, under mild regulatory conditions. Under the exponential family assumption, the iteration complexity is bounded by the cube of the condition number of the Fisher information matrix, for $M$ sufficiently large. For a finite $M$, the difference between the score function and the gradient of the empirical N2CE loss exhibits a bias-variance trade-off, where the bias diminishes at a rate $O(1/M^2)$.

2. The proposed method is simple, easy-to-understand, and requires only a modest change in the NCE objective function so there is little additional cost in applying it. Yet it shows an impressive performance boost in image generation, anomaly detection, and offline black-box optimization. For example, in CIFAR-10 and ImageNet64x64, combined with 10-step or 1-step sampling scheme, a SOTA-level result is achieved with half the training iterations.

**Weaknesses:**

1. The key assumption for the main result that the gradient of the N2CE loss approaches to the score function as $M\to\infty$ relies on the assumption that $p_{\alpha}$ is dominated by $q_0$ uniformly over $\alpha$. Density chasm is most severe when this assumption is violated, and simply increasing $M$ does not completely resolve this.

2. The choice of the noise density $q_0$ may be much more critical than its scale. Some sort of comparison based on a matrix of $M$ and $q_0$ is warranted.

3. Proposition 3.3 suggest bias-variance trade-off in $M$, but no guide for choosing $M$ is provided.

4. Large-scale benchmarks in limited to visual datasets like CIFAR-10 and ImageNet64x64. It is desirable to test N2CE on the other domains, such s LLM or speech data.

**Questions:**

1. In Proposition 3.3, isn't the second term of $V_n$ $\min\\{ E_{q_0}\\|\nabla_{\alpha} \log r_{\alpha}\\|^2, M^2 \\|\nabla_{\alpha} r_{\alpha}\\|^2\\}$?

2. Also in Proposition 3.3, the result suggests that there is an optimal, finite, choice of $M$, unless $\\|\nabla_{\alpha} r_{\alpha}\\|^2 \gg M^2 \\|\nabla_{\alpha} \log r_{\alpha}\\|^2$.for all $M$. But Proposition 3.1 states that the larger $M$, the better. How can this apparent conflict be resolved?

---

> ### Author Response · Authors · 2025-11-20
> **Response to Reviewer Qdxk (1/N)**
>
> We would like to sincerely thank reviewer Qdxk for the time devoted into reviewing our submission. We appreciate that reviewer Qdxk raised detailed and insightful questions, to which we will provide point-to-point responses below and hopefully help to ease your concerns.
>
> **Discussions about the noise density**
>
> We take the liberty of discussing the subtlety of the interplay between noise densities and our objective to respond to the raised 1. and 2. weaknesses.
> In 1., reviewer mentioned that
> > (the result ...) relies on the assumption that $p_\alpha$ is dominated by $q_0$ uniformly over $\alpha$. Density chasm is most severe when this assumption is violated, and simply increasing M does not completely resolve this.
>
> We agree that if the support of $q_0$ does not entirely contain $p_*$, there exists the risk that the optimal ratio cannot be learned with *any* ratio-estimation-based method. However, it is not unnatural that we can use Gaussian-like distributions as the noise density for continuous cases, which spans the entire space; we may similarly use uniform distribution for discrete cases as well. In fact, a large portion of our experiments including synthetic gaussian, latent space modeling for image generation and anomaly detection, and offline black-box optimization all rely on the standard normal distributions to obtain decent results. The key property that distinguishes our objective from the vanilla NCE objective is: our objective **can still behave** when the noise density is at its tail while the target density is not, i.e., the ratio of ground-truth $p_*/q_0$ can be very large but still exists. The positive part of our objective and also NCE naturally fulfills this: $E_{q_*}
>    \left[\log \frac{r_{{\alpha}}(x)}{M + r_{{\alpha}}(x)}\right]$. But only by increasing $M$, which is the case in our objective, we can also achieve this for the negative part $E_{q_{0}} \left[\log \frac{M}{M + r_{{\alpha}}(x)}\right]$, i.e., the objective values remains moderate. In fact, we can use $M$ to modulate the peakyness of the N2CE objective in these ill-behaved regions. We consider this as a handy solution at least for these pathological cases.
>
> > The choice of the noise density may be much more critical than its scale. Some sort of comparison based on a matrix of
>  and is warranted.
>
> We agree that the choice of the noise density could be important, especially with finite $M$ and finite sample sizes for estimation. In our experiments, we mainly used two categories of noise densities: (i) for cases where we need explicit densities, such as synthetic gaussian, latent space modeling for image generation and anomaly detection, and offline black-box optimization experiments, we use standard Gaussian as our noise density. (ii) for reward and/or critic learning in high-dimensional spaces as we did in sec. 4.2, we consider the implicit distribution defined by the learned sampler as our noise density, which is evolving along the training process. In both cases, we observe that our method shows decent results. That being said, we will explicitly discuss this in our updated draft and provide outlooks to other choices of noise density.

---

> > ### Author Response · Authors · 2025-11-20
> > **Response to Reviewer Qdxk (2/N)**
> >
> > **Clarifications about Prop 3.3 and its connection with Prop 3.1**
> >
> > We appreciate that the reviewer dives into details about our results.
> >
> > > In Proposition 3.3, isn't the second term of $V_n$ ...
> >
> > We confirm that the second term *is* our original result in Prop 3.3. After checking again our proof in B.2, we suspect that it is due to our typo at L1095 and L1100 that leads to reviewer's concern. We clarify that the second term in these lines should be $\nabla_\alpha r_\alpha$ instead of $\nabla_\alpha \log r_\alpha$ and have revised it accordingly. We apologize for the inconvenience and truly thank reviewer Qdxk for bringing this matter to our attention. Again, this typo does not affect our final result.
> >
> > > Also in Proposition 3.3, the result suggests that there is an optimal, finite, choice of $M$ ... But Proposition 3.1 states that the larger $M$, the better. How can this apparent conflict be resolved?
> >
> > > Proposition 3.3 suggest bias-variance trade-off in $M$, but no guide for choosing is provided.
> >
> > We appreciate that reviewer Qdxk raises this question. This points to the core idea of our method: instead of simply studying the limiting case of the $L_M$ objective, we indeed proposes a *family* of distributions indexed by $M$, where $M=1$ leads to vanilla NCE, and $M \to \infty$ recovers MLE. In fact, if we have **infinite sample size**, meaning that we can recover the true expectation with Monte Carlo estimation, then the larger $M$, the better because of our results in Prop 3.1 and especially in Theorem 3.2. But in practice as we stated in Prop. 3.3, we only have a (fixed) **finite sample size**, where the variance of estimation kicks in. This is the key motivation of our result in Prop. 3.3 as we want to enjoy the reduced bias by our results as much as possible, while not exaggerating too much about the variance by increasing $M$. In fact, if we allow the sample size $n$ to increase as well, then according to Prop 3.3, as long as we have $M \sim o(\sqrt{n})$, meaning that M increases at a rate slower than $\sqrt{n}$, we can still enjoy the bias reduction with a controlled variance. In short, the root reason is the variance caused by a fixed and finite sample size in our setting. And we would like to clarify that Prop 3.1 only addresses the mean value. We hope this would help with further understanding of our work.
> >
> > We agree that choosing the optimal $M$ is an interesting research question, as we observed U-shaped patterns in different scenarios when sweeping through $M$ (tab. 14, 15 & 24). Indeed, again based on our Prop 3.3 results and as we discussed above, although finding an exact optimal $M$ could be case-by-case as it also depends heavily on the ratio estimator $r_\alpha$, **our result does suggest that with a given sample size (batch size) of $n$, $M$ in general may not exceed $C \sqrt{n}$ where $C$ is a constant**, C=10 for example. We can see in tab. 14, 15 & 24 that this value fits very well with our empirical observations. Due to the scope of this paper, we will have to leave studying an exact optimal $M$ for future works.
> >
> > **Further applications**
> >
> > > It is desirable to test N2CE on the other domains, such s LLM or speech data.
> >
> > We fully agree with this suggestion and looking forward to applying our method to other domains.
> >
> > We sincerely hope our further clarification could help to address the raised issues. We are open to any further questions regarding our work. We truly appreciate the time devoted by reviewer Qdxk on our submission.

---

### Author Response · Authors · 2025-11-21
**Rebuttal Summary**

We want to thank all the reviewers again for their valuable comments. Their feedback has been constructive. We revised and uploaded the manuscript and summarize the key changes below:

- We update the Appendix D.1 section to include further comparison with NWJ objective and a variant of our objective that only reweights the negative part.
- We add 1-step image generation qualitative results from our distilled models in Fig. 6 in the Appendix D.4.
- We revised typos in lines 1095 and 1100, which do not affect our results.

Further, after discussing with reviewer PhJz (highlighted in red in our updated manuscript) :

- We have update Sec. 2 at L93 and L112-124 to further introduce the NWJ framework as our background.
- As promised, we have added a new Sec. 3.4 `An Information-Theoretic Perspective` L252-285 to incorporate the discussion and reviewing comments from reviewer PhJz.
- As promised, we have revised Sec. 3.3 as `Practical Error Analysis, N$^2$CE Family and Regularization`, revised L219-223 and added L226-235 `The N$^2$CE family and U-shaped patterns` to make the optimal-$M$ characterization more explicit and precise.
- We have included the new simulation results in our appendix D.1 section, and provided explicit reference in the main text at L282-283.

In our rebuttal responses, we:

- Further clarify that we propose a family of objectives $L_M$ indexed by $M$, establish that NCE, MLE and NWJ are special cases of our objective, and emphasize that we focus on the finite-$M$ and finite sample size $n$ scenarios because it faithfully reflects common machine learning problem settings.
- Further clarify that Proposition 3.3 exactly indicates if the sample size (batch size) $n$ is fixed, there exists an optimal $M$ due to the derived bias-variance tradeoff. In addition, Proposition 3.3 also provides guidance on how to select appropriate $M$s: it may not exceed $C\sqrt{n}$, where $C$ is a constant.
- Further clarify that Proposition 3.3 is not conflicting with Proposition 3.1. Proposition 3.1 is always true as it addresses only the mean estimation of the gradient, while in practice the variance starts to kick in. Proposition 3.3 also does not indicate that larger $M$ is worse, it indicates that $M$ should grow at a rate of $o(\sqrt{n})$, meaning that if we can increase the sample size, then we can follow this regime to safely increase $M$ at a rate slower than $C \sqrt{n}$ to obtain better results.
- Further provide empirical results that our method is clearly better than NWJ, simply reweighting the negative part of either vanilla NCE or our objective. We further discuss and compare with other related methods like DRE-$\infty$ and CDM, clearly stating the methodological and empirical advantages of our method.
- Further clarify the experiment settings. We also work on further improve the organization of our results to further enhance the readability.

---

### Meta-Review · Area_Chair_FyXy · 2025-12-29

**Summary:**

The paper proposes a simple modification of Noise Contrastive Estimation (NCE), which significantly improves its ability to learn (unnormalized) density models in practical settings. In the initial round of reviews, two reviewers evaluated the paper as above the acceptance threshold (scores 6,6) and two evaluated it below the threshold (score 2,4). The main concerns were: (1) lack of clarity regarding whether and how the proposed method overcomes the density chasm problem (Qdxk, PhJz); (2) lack of guidelines for the optimal choice of the hyperparameter M (Qdxk, PhJz, LHEq) and particularly why the optimum is not at infinity (PhJz) or not much smaller than that used in the experiments (LHEq); (3) lack of clarity regarding the use of the method at inference (Qgtk); (4) failure to recognize that for M tending to infinity, the objective coincides with known methods, including that of Nguyen Wainwright and Jordan (NWJ), the formulation from the f-GAN paper, and that from the self-normalizing DRE approach of Poole et al. (PhJz).

After the rebuttal, reviewer PhJz engaged in several rounds of discussions, and eventually indicated they would like to raise the score from 2 to 6 given that the authors incorporate the modifications in the discussion. The AC views the authors’ modifications in response to PhJz as mostly satisfactory, except for the omission of the discussion on DRE-$\infty$ and CDM, which the authors provided in a response but did not incorporate in the paper. The authors are highly encouraged to include those discussions in the paper.

The AC views the responses to all other reviewers as satisfactory as well. Based on this, the AC sees the paper as appropriate for publication in this ICLR.

**Reviewer Concerns:**

Most major concerns have been properly addressed and clarified by the rebuttal, including the addition of a discussion about the relation of the method to NWJ and clarifications about the optimal choice of M for a given finite number of samples n. As mentioned above, the AC believes there is still room for a discussion about recent attempts to tackle the density chasm problem, like DRE-$\infty$ and CDM, and about the advantages of the proposed method over those approaches.

**Reviewer Scores:**

**Qdxk: score 6.**

This is also the original score. The rebuttal seems to have addressed all the reviewer’s concerns.

**Qgtk: score 6.**

This is also the original score. The AC encourages the authors to incorporate more information regarding inference in the final version, as suggested by the reviewer.

**PhJz: score 6.**

The reviewer’s original score was 2. After several rounds of discussions, which improved the manuscript (as the authors acknowledge), the reviewer indicated they would like to raise the score from 2 to 6 if all the discussed modifications are incorporated in the paper. The AC confirms that the authors made the modifications asked by the reviewer.

**LHEq: score 6.**

The initial score was 4. The rebuttal convincingly clarified the reviewer’s major concern that the proposed method outperforms a simple baseline that reweighs the loss of the negative examples. The AC predicts that the reviewer would have increased the score to 6.

---

### Decision · Program_Chairs · 2026-01-26

Accept (Poster)